EMBO
Molecular Medicine

# Bi-allelic variants in *WDR47* cause a complex neurodevelopmental syndrome

Efil Bayam [ID][1,2,3,4✉], Peggy Tilly[1,2,3,4], Stephan C Collins [ID][1,2,3,4,5], José Rivera Alvarez[1,2,3,4], Meghna Kannan[1,2,3,4], Lucile Tonneau [ID][5], Elena Brivio[1,2,3,4], Bruno Rinaldi[6,23], Romain Lecat [ID][1,2,3,4], Noémie Schwaller[1,2,3,4], Ludovica Cotellessa [ID][7], Sateesh Maddirevula [ID][8,9], Fabiola Monteiro[10], Carlos M Guardia[11], João Paulo Kitajima [ID][10], Fernando Kok[10,12], Mitsuhiro Kato[13], Ahlam A A Hamed[14], Mustafa A Salih [ID][15], Saeed Al Tala[16], Mais O Hashem[8], Hiroko Tada[17,18], Hirotomo Saitsu[19], Mariano Stabile [ID][20], Paolo Giacobini [ID][7], Sylvie Friant[6,24], Zafer Yüksel[21], Mitsuko Nakashima[19], Fowzan S Alkuraya[8,22], Binnaz Yalcin [ID][1,2,3,4,25,26✉] & Juliette D Godin [ID][1,2,3,4,26✉]

## Abstract

**Brain development requires the coordinated growth of structures and cues that are essential for forming neural circuits and cognitive functions. The corpus callosum, the largest interhemispheric connection, is formed by the axons of callosal projection neurons through a series of tightly regulated cellular events, including neuronal specification, migration, axon extension and branching. Defects in any of those steps can lead to a range of disorders known as syndromic corpus callosum dysgenesis (CCD). We report five unrelated families carrying bi-allelic variants in *WDR47* presenting with CCD together with other neuroanatomical phenotypes such as microcephaly and enlarged ventricles. Using in vitro and in vivo mouse models and complementation assays, we show that WDR47 is required for survival of callosal neurons by contributing to the maintenance of mitochondrial and microtubule homeostasis. We further propose that severity of the CCD phenotype is determined by the degree of the loss of function caused by the human variants. Taken together, we identify *WDR47* as a causative gene of a new neurodevelopmental syndrome characterized by corpus callosum abnormalities and other neuroanatomical malformations.**

**Keywords** WDR47; Corpus Callosum Dysgenesis; Neurodevelopmental Disorder; Callosal Neurons; Microtubule and Mitochondrial Homeostasis

**Subject Categories** Genetics, Gene Therapy & Genetic Disease; Neuroscience

## Introduction

Brain development is a dynamic process that involves the coordinated growth of structures at multiple scales and relies on a system of cues that together contribute to the formation of circuits and cognitive functions. Dysfunctional brain development can impact many biological functions, including learning and memory, motor control, emotion processing, and social skills. These functions are mediated by various brain structures, such as the cortex, the hippocampus, the cerebellum, and the striatum. Proper anatomy, including the integrity of structures that connect brain regions like the corpus callosum, is crucial for these functions.

The corpus callosum consists of 190 million axons in an adult human brain making it the largest commissure in mammals. This interhemispheric connection is largely formed by the axons of layer 2/3 callosal projection neurons (CPNs) through a series of tightly regulated cellular events involving commitment of newborn

[1]Institut de Génétique et de Biologie Moléculaire et Cellulaire, IGBMC, Illkirch F-67404, France. [2]Centre National de la Recherche Scientifique, CNRS, UMR7104, Illkirch F-67404, France. [3]Institut National de la Santé et de la Recherche Médicale, INSERM, U1258, Illkirch F-67404, France. [4]Université de Strasbourg, Strasbourg F-67000, France. [5]Université de Bourgogne, INSERM UMR1231, 21000 Dijon, France. [6]Université de Strasbourg, CNRS, GMGM UMR7156, F-67000 Strasbourg, France. [7]Université de Lille, INSERM, CHU Lille, Laboratory of Development and Plasticity of the Neuroendocrine Brain, Lille Neuroscience & Cognition UMR-S 1172, Lille, France. [8]Department of Translational Genomics, Center for Genomic Medicine, King Faisal Specialist Hospital and Research Center, Riyadh, Saudi Arabia. [9]College of Medicine, Alfaisal University, Riyadh, Saudi Arabia. [10]Mendelics Análise Genomica SA, CEP 02511-000, Sao Paulo, Brazil. [11]Placental Cell Biology Group, National Institute of Environmental Health Sciences, National Institutes of Health, Research Triangle Park, NC 27709, USA. [12]Department of Neurology, University of Sao Paulo School of Medicine, 01246-903 Sao Paulo, Brazil. [13]Department of Pediatrics, Showa University School of Medicine, Tokyo 142-8555, Japan. [14]Department of Pediatric and Child Health, Faculty of Medicine University of Khartoum, Khartoum, Sudan. [15]Health Sector, King Abdulaziz City for Science and Technology, Riyadh 11442, Saudi Arabia. [16]Department of Pediatrics, Genetic Unit, Armed Forces Hospital, Khamis Mushayt, Saudi Arabia. [17]Department of Brain and Neurosciences, Tokyo Metropolitan Institute of Medical Science, Tokyo 156-0057, Japan. [18]Division of Pediatrics, Chibaken Saiseikai Narashino Hospital, Chiba 275-8580, Japan. [19]Department of Biochemistry, Hamamatsu University School of Medicine, 1-20-1 Handayama, Chuuo-ku, Hamamatsu 431-3192, Japan. [20]Center of Genetics and Prenatal Diagnosis "Zygote", 84131 Salerno, Italy. [21]Human Genetics, Bioscientia GmbH, Ingelheim, Germany. [22]Department of Anatomy and Cell Biology, College of Medicine, Alfaisal University, Riyadh, Saudi Arabia. [23]Present address: INSERM, U1112, CRBS (Centre de recherche en biomédecine de Strasbourg), Université de Strasbourg, Strasbourg F-67000, France. [24]Present address: PCBIS-IMPReSs, Plateforme de Chimie Biologique Intégrative de Strasbourg, UAR 3286 CNRS/Université de Strasbourg, 67400 Illkirch, France. [25]Present address: INSERM UMR1231, Université de Bourgogne, 21000 Dijon, France. [26]These authors contributed equally: Binnaz Yalcin, Juliette D Godin. ✉E-mail: bayame@igbmc.fr; binnaz.yalcin@inserm.fr; godin@igbmc.fr

neurons to CPN identity, migration to the proper cortical layer, axonal extension and guidance towards the midline with the aid of secreted and contact-mediated axon-guidance cues (Fenlon and Richards, 2015; Leyva-Diaz and Lopez-Bendito, 2013; Lowery and Van Vactor, 2009). Defects in any of those steps can lead to a spectrum of disorders collectively referred to as syndromic corpus callosum dysgenesis (CCD). Syndromic CCD occurs together with other neuroanatomical phenotypes (NAPs (Collins et al, 2019)) such as microcephaly presenting a high risk of cognitive and neuropsychiatric disorders (Edwards et al, 2014; Fenlon and Richards, 2015). To date, there are more than 300 known genetic syndromes associated with CCD, however, the majority remain undiagnosed at the molecular level and their underlying pathophysiological mechanisms under characterized (Edwards et al, 2014).

WDR47 is a poorly-studied microtubule associated protein particularly enriched in neurons in humans (Cardoso-Moreira et al, 2019) and mice (Kannan et al, 2017; Wang et al, 2012) that we previously identified as a novel regulator of mouse CC development using a large-scale knock-out (KO) screen (Collins et al, 2019). *Wdr47* is essential for mouse survival and brain morphogenesis in a dose dependent manner (Chen et al, 2020; Kannan et al, 2017). Total loss of *Wdr47* leads to lethality immediately after birth (*Wdr47*tm1b/tm1b). Yet, a fraction (nearly 6%) of mice expressing 30% of *Wdr47* (*Wdr47*tm1a/tm1a) survive and display severe neuroanatomical abnormalities including primary microcephaly and severe loss of commissural fibers, including CC, that manifest as hyperactivity and sensory motor gating abnormalities (Kannan et al, 2017). At the cellular level, WDR47 has been shown to regulate the proliferation of late progenitors and neuronal migration in vivo in mouse developing brains as well as axonal and dendritic development in vitro in primary culture of neurons (Buijs et al, 2021; Chen et al, 2020; Kannan et al, 2017). While WDR47 is known to regulate cellular processes that are critical for proper brain development such as microtubule stability (Buijs et al, 2021; Chen et al, 2020), intracellular transport (Guardia et al, 2021), and autophagy (Kannan et al, 2017), it is still unclear whether these regulatory functions are required for proper development of callosal projection neurons.

Despite the clear role of Wdr47 in mouse brain development and function, there is yet no clear association of *WDR47* variants with neurological disorders in humans. The only exception is a genome-wide association study (GWAS) that identified *WDR47* as a hub gene in Alzheimer's disease (Zhang et al, 2020). Here, we provide the first evidence of a causal relationship between variants in *WDR47* and human neuroanatomical disorders. We report five bi-allelic variants in patients presenting with severe brain malformations including CCD, microcephaly, thin cortex, hydrocephalus and cerebellar atrophy. Using a combination of in vitro and in vivo mouse models and complementation assays, we show that Wdr47 is required for survival of callosal neurons. Although independent from its previously identified roles in neuronal migration and axonal extension, WDR47's role in neuronal survival partly relies on its cell-autonomous function on microtubule cytoskeleton. We propose that the degree of the loss of function imposed by the different human *WDR47* bi-allelic variants varies and dictates the severity of the CC phenotype. Taken together, our data imply *WDR47* as an important gene for neuroanatomical disorders in humans and that *WDR47* variants identified in exome and genome sequencing in unexplained syndromic cases involving corpus callosum dysgenesis, microcephaly and other neuroanatomical phenotypes should be considered as probably pathogenic.

## Results

### Human *WDR47* bi-allelic missense variants delineate complex neurodevelopmental phenotypes with corpus callosum dysgenesis (CCD), microcephaly, intellectual disability, and epilepsy

We identified, through the GeneMatcher platform (Sobreira et al, 2015) and data sharing, a series of seven cases (M01 to M07; five males, 2 females; M01-M02 and M05-M06 are siblings) from five unrelated families worldwide (Families 1 to 5 living in Sudan, Japan, Saudi Arabia, Italia and Brazil, respectively) with bi-allelic missense variants (Fig. 1A–E). Variants were inherited from healthy consanguineous parents in an autosomal recessive manner at the exception of the Japanese Family 2 that showed a compound heterozygous mode of inheritance where the parents are healthy non-consanguineous. The median age at the time of the last evaluation was 2,65 years (ranging from 1 day old to 11 years old).

The clinical features are summarized in Dataset EV1 and fully described in Appendix Supplementary Information. The brain is the most affected organ. Indeed, all cases had neuroanatomical phenotypes (NAPs) with the most frequent phenotypes being corpus callosum dysgenesis (7/7) and microcephaly (7/7), followed by enlarged ventricles with no other multicilia-related pathological symptoms including ciliary dyskinesia, situs invertus or chronic coughing (5/7), a reduced size of the hindbrain (5/7), simplified gyral pattern (2/7) and periventricular nodular heterotopia (1/7) (Fig. 1F–J). Other features included mild to severe intellectual disability (7/7), epilepsy (7/7), developmental delays (4/5), muscle tone defects (5/5), and craniofacial features (3/3). Craniofacial characteristics included widely spaced eyes (M03), epicanthus (M03, M05, and M06), a short philtrum (M03), hypertelorism (M04, M05, and M06), medial flaring of eyebrows (M04), thick upper and lower lips (M04, M05, and M06), elevated ear lobules, a low auricle (M04) and a tented upper lip (M03 and M04) (Fig. 1G–I) (Dataset EV1, Appendix Supplementary Information). The brain-specific clinical manifestation of the identified variants in *WDR47* correlates with the enriched expression of human *WDR47* transcripts in central nervous system structures compared to other organs throughout life (Fig. 1K,L) (Cardoso-Moreira et al, 2019). WDR47 immunostaining performed on sagittal and coronal brain sections of two Gestational Week (GW) 14 fetuses revealed a high expression of the WDR47 protein in the developing cortex. Within the cortex, WDR47 is expressed in germinal zones and cortical plate as well as in the intermediate zone that is enriched in growing axons (Fig. 1M,N). Analysis of single-cell transcriptomic data (Abdulla et al, 2023) confirmed the enriched expression of human *WDR47* in all subtypes of cortical projection neurons, including layer 2/3 callosal neurons, in both developing and adult cortices (Fig. 1O). Overall, these analyses reveal a strong association between expression of WDR47 and the specific neuroanatomical phenotypes observed in patients.

We identified five missense *WDR47* variants that occur within conserved residues located in the N-terminal domain (NTD (Ren et al, 2022)) (NM_001142551.2, c.(578 G > A), p.(Arg193His), M01

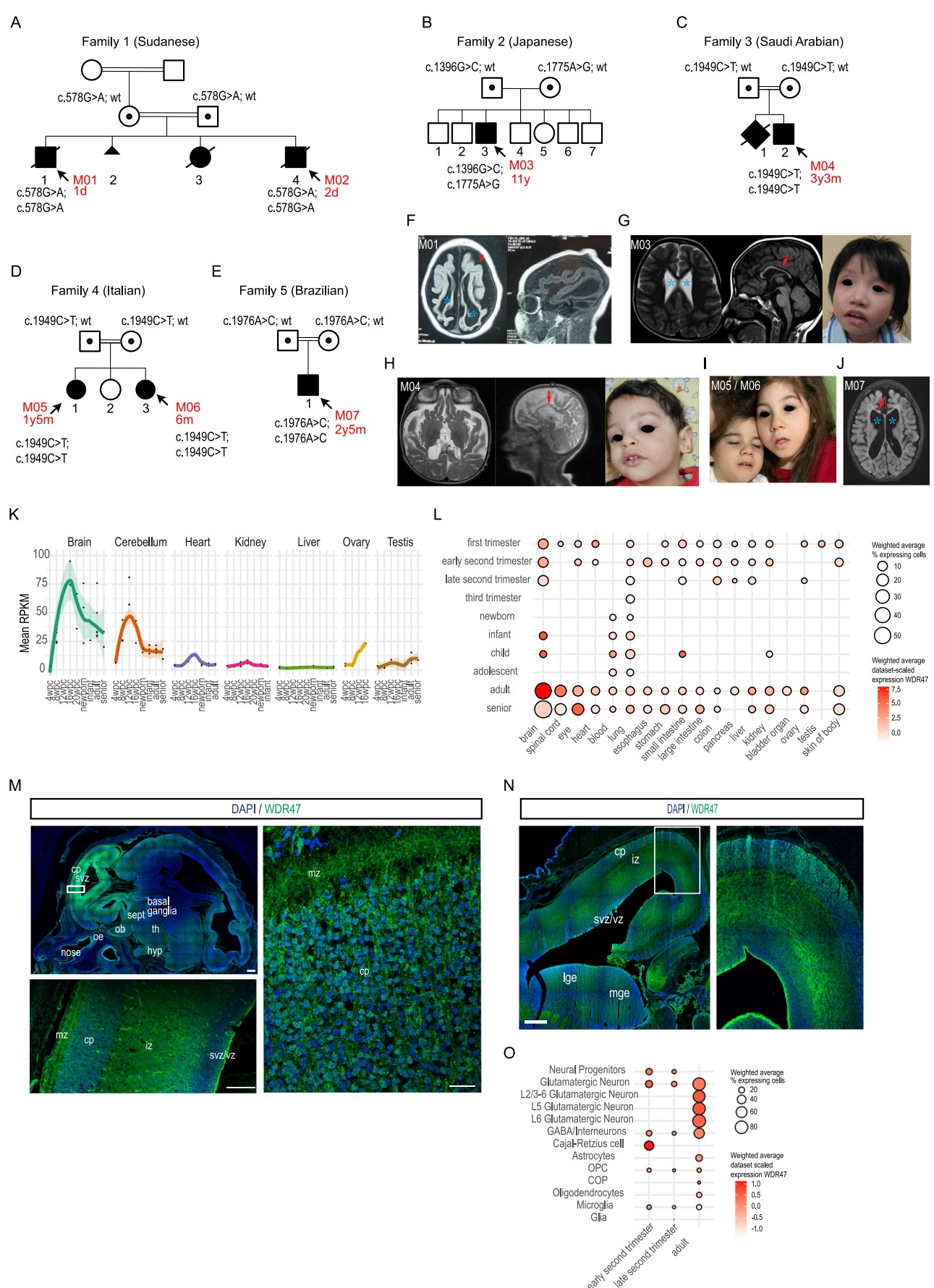

**Figure 1. Clinical and brain imaging data from seven patients with *WDR47* recessive loss-of-function variants.**

(A–E) Pedigrees showing the segregation of the identified *WDR47* variants with the syndrome in consanguineous families (exception of Family 2). (F–J) Axial and sagittal T1 and T2-weighted brain MRI and facial features of patients with *WDR47* variants. The red arrowhead indicates the simplified gyral pattern of the cortex, the blue stars show ventricular dilatations, and the red arrows point to thin corpus callosum. (K) Expression values of WDR47 as RPKM (Reads per kilo base per million mapped reads) throughout life for different organs in human. Dots represent average values for each replicate. Shaded regions represent standard deviation of distribution. Data from (Cardoso-Moreira et al, 2019). (L) Expression pattern of *WDR47* in human across organs (x axis) and ages (y axis) showing prominent expression in the brain. Data obtained by multiple published dataset through https://cellxgene.cziscience.com (Abdulla et al, 2023). Color scale represents the weighted average expression across datasets and dot size represents the weighted average percentage of cells expressing *WDR47*. (M–N) Images of (M) sagittal and (N) coronal human fetal brain sections immunostained with WDR47 antibody at GW14. Scale bars: 1 mm (upper left), 500 μm (lower left) and 40 μm (right). vz: ventricular zone, svz: subventricular zone, iz: intermediate zone, cp: cortical plate, mz: marginal zone, oe: olfactory epithelium, ob: olfactory bulb, lge: lateral ganglionic eminence, mge : medial ganglionic eminence, th: thalamus, hyp: hypothalamus, sept: septum. (O) Expression pattern of *WDR47* across ages and selected cell types of the human cerebral cortex. Data obtained by multiple published dataset through https://cellxgene.cziscience.com (Abdulla et al, 2023). Color scale is as in (L). Source data are available online for this figure.

and M02), the central region (NM_001142551.2, c.(1396 G > C), p.(Asp466His) paternal, c.(1775 A > G), p.(Lys592Arg) maternal, M03), and the C-terminal WD40 domain (NM_001142551.2, c.(1949C > T), p.(Pro650Leu), M04, M05 and M06 and c.(1976A > C), p.(His659Pro), M07) (Fig. 2A and Appendix Fig. S1A–D). Various predictions scores, such as CADD, SIFT, Polyphen2, and PhastCon, indicated deleteriousness at conserved residues (Dataset EV1). None of the five variants are reported in public databases, including dbSNP, 1000 genomes and gnomAD at the homozygous state. Of note, the other variants of unknown significance found in patients were excluded based on their predicted weak pathogenic score and absence of overlap with the clinical features observed in patient carrying homozygous variants (Table EV1). To evaluate the impact of identified variants in the full-length WDR47, we employed structural modeling with an Alphafold2-derived atomic model. The variants within the central domain (p.(Asp466His)) or close or within the C-terminal WD40 (p.(Lys592Arg), p.(Pro650-Leu), p.(His659Pro)), localize to solvent-exposed regions and, although their specific effects remain uncertain, are unlikely to disrupt the correct fold of the protein (Fig. 2B). In contrast, the p.(Arg193His) substitution, that maps to WDR47-NTD involved in homodimerization (Ren et al, 2022), likely interferes with the formation of the intertwined WDR47-NTD dimer. This disruption is likely due to the alteration of the salt bridge that Arginine 193 forms with Glutamate 205, a corresponding residue in the other monomer (Fig. 2B). This weakened WDR47 homodimerization interface was also confirmed with the PISA server (http://www.ebi.ac.uk/pdbe/prot_int/pistart.html) (Krissinel and Henrick, 2007).

To interrogate the molecular effect of the *WDR47* variants, we first assessed WDR47 protein expression levels after overexpression of HA-tagged WT and mutant constructs in N2A neuroblastoma cell line. The variants have various effect on protein expression level ranging from no effect for p.(Asp466His), p.(Lys592Arg), and p.(His659Pro) variants, to moderate (−32%, p.(Pro650Leu)) and severe (−72%, p.(Arg193His)) decrease compared to the wild-type (WT) human protein (Fig. 2C). Decreased expression level of Arg193His mutant protein was rescued after treatment with the proteasome inhibitor MG132, suggesting instability of the WDR47 mutant Arg193His protein (Fig. 2D). Such rescue was not observed with the Pro650Leu, indicating that the Pro650Leu WDR47 protein is not targeted for proteasomal degradation (Fig. 2D). Next, we examined WDR47 protein levels in patient's samples when available (M01 and M03). Though the levels of *WDR47* transcripts remained stable (Appendix Fig. S1E), immunoblotting revealed a

nearly complete loss of p.(Arg193His) WDR47 protein levels in fibroblasts derived from M01 (Fig. 2E), further indicating that the mutant Arg193His WDR47 protein is unstable. Western-Blot analysis in lymphoblastoid cell line obtained from the Patient M03 carrying the compound heterozygous variant (p.(Asp466His), p.(Lys592Arg)) confirmed the absence of phenotype at the protein expression level (Fig. 2F). Overall, we identified five missense substitutions in the *WDR47* gene, that differently impact on WDR47 protein abundance, in seven patients presenting with severe neurodevelopmental delay associated with corpus callosum dysgenesis (CCD), microcephaly and other NAPs.

## Neuron-specific *Wdr47* knock-out in mice recapitulates NAPs observed in human patients

Interestingly, the clinical manifestations of the patients are recapitulated in previously studied constitutive *Wdr47* KO mouse models (Chen et al, 2020; Kannan et al, 2017; Liu et al, 2021) that display severe neuroanatomical phenotypes including CCD and microcephaly among other NAPs (Dataset EV1, Table 1 and Appendix fig. S2A). Together with the fact that the pattern of expression of the mouse WDR47 is comparable to the one of the human WDR47 (Fig. EV1) (Abdulla et al, 2023; Cardoso-Moreira et al, 2019), this indicates that *Wdr47*-deficient mice are valuable models to understand the pathogenicity of *WDR47* loss-of-function variants in humans. As NAPs could arise from defects in progenitors and neurons or both, we investigated the extent to which NAPs induced by *WDR47* deficiency are intrinsic to neurons. We first conditionally deleted *Wdr47* specifically in early glutamatergic projection neurons in the dorsal telencephalon using the neuronal basic helix-loop-helix Nex^Cre mouse line (Goebbels et al, 2006) (Appendix Fig. S2A). Loss of *Wdr47* in E18.5 Nex^Cre;*Wdr47*^fl/fl embryos was validated by immunoblotting (Appendix Fig. S2B). To ensure high reproducibility, we acquired neuroanatomical data in both Nex^Cre;*Wdr47*^fl/fl and *Wdr47*^tm1a/tm1a simultaneously using our recently established neuroanatomical quantification approach for embryonic brains (Nguyen et al, 2022). Our histology pipeline was designed based on two coronal sections at Bregma +2.19 mm and Bregma +3.51 mm, spanning 17 developmentally distinct brain regions for 32 quantifiable brain parameters of area, height, and width measurements (Fig. 3A–D and Dataset EV2). Each brain image was carefully checked for quality control before analysis taking into consideration the correct stereotaxic position and the symmetry both along the dorsoventral and rostrocaudal orientations. In line with our previous report

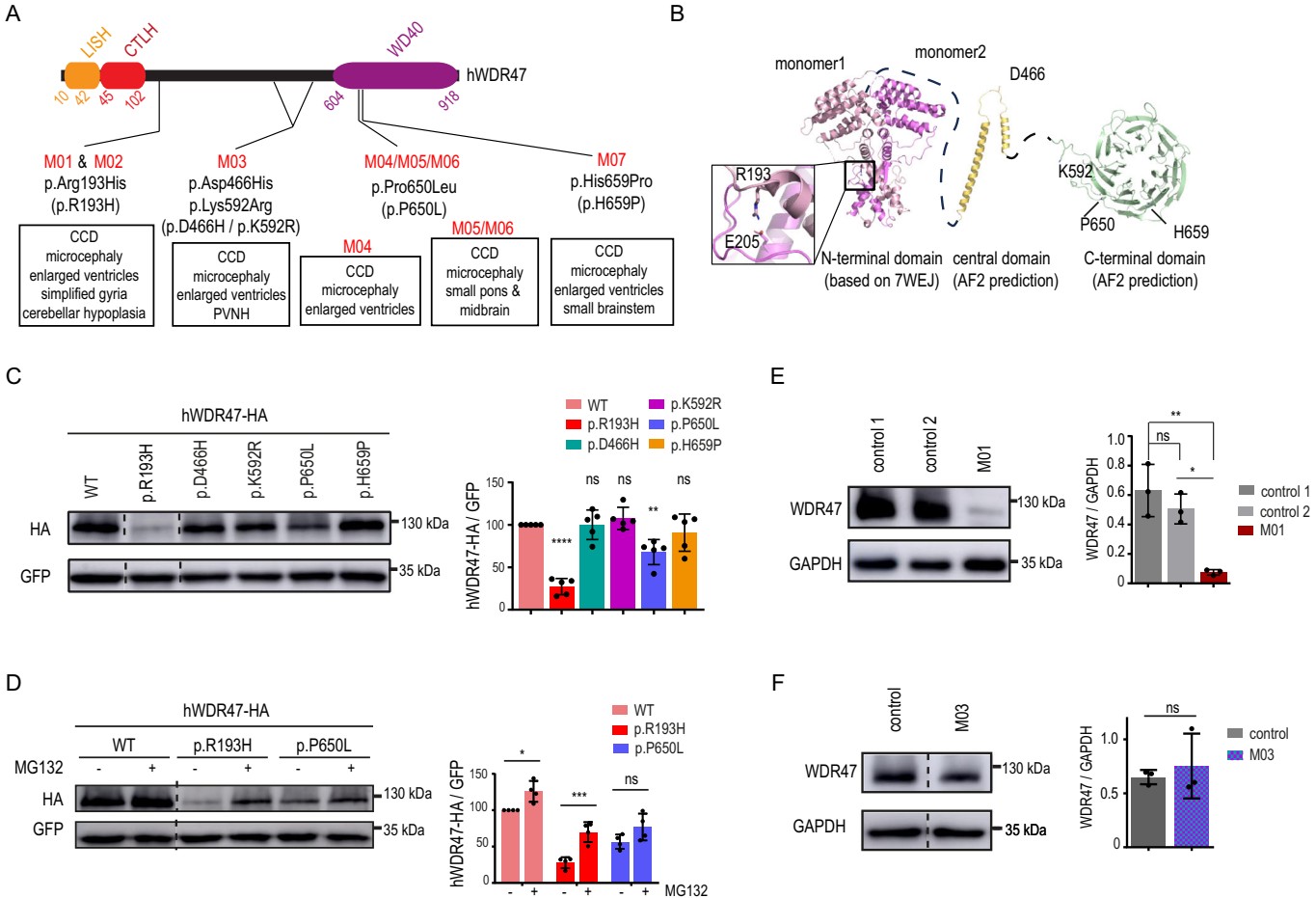

**Figure 2. WDR47 missense variants affect the structure and stability of the WDR47 protein to various extent.**

(A) Schematic representation of the human WDR47 (hWDR47) protein depicting the LISH, CTLH, and WD40 domains and the position of the mutated amino acids for M01&M02 (p.(Arg193His) (R193H)), M03 (p.(Asp466His)/p.(Lys592Arg) (D466H/L592R)), M04, M05, and M06 (p.(Pro650Leu) (P650L)) and M07 (p.(His659Pro) (H659P)). (B) Structural modeling of the human WDR47 (hWDR47) protein with an Alphafold2-derived atomic model. Homodimerization of the N-terminal featuring the LISH and CTLH domains is shown based on the crystal structure of mouse WDR47-NTD intertwined dimer. The position of the mutated amino acids for M01 & M02 (p.R193), M03 (p.D466) and (p.K592), M04, M05 and M06 (p.P650) and M07 (p.H659) and E205 residue that makes a salt bridge with the p.R193 are indicated. (C) Western blot analysis of extracts from N2A cells transfected with the indicated HA tagged hWDR47 constructs showed variable effect of variants on WDR47 protein levels. GFP was used as a transfection control. Dotted lines indicate where the membrane was cut. Data (means ± s.d.) from 5 independent cultures was analyzed by one-way ANOVA, with Bonferroni's multiple comparison test. (D) MG132 treatment rescued the decreased levels of p.R193H variant suggesting proteasomal degradation of the mutant protein. Data (means ± s.d.) from 4 independent cultures was analyzed by two-way ANOVA, with Bonferroni's multiple comparison test. (E, F) Western blot analysis showing that endogenous levels of WDR47 is (E) decreased in fibroblast extracts from patient M01 and (F) unchanged in lymphoblastoid cell extracts from patient M03. (E) Fibroblast and (F) lymphoblastoid cell lines from two and one healthy subjects were used, respectively, as controls. Data (means ± s.d.) from 3 cultures was analyzed by (E) one-way ANOVA, with Bonferroni's multiple comparison test and (F) unpaired t-test. ns, non-significant; *$P < 0.05$; **$P < 0.01$; ***$P < 0.001$; ****$P < 0.0001$. Dotted lines indicate the position where the membrane was cut. Exact $P$ values are listed in Dataset EV4. Source data are available online for this figure.

**Table 1. Comparison of neuroanatomical phenotypes between constitutive and brain-specific Wdr47 knockout mouse models.**

| | Wdr47^tm1a/tm1a | Nex^Cre;Wdr47^fl/fl | Wdr47^tm1a/tm1a | CaMKIIα^Cre;Wdr47^fl/fl |
|---|---|---|---|---|
| Wdr47 expression levels | 30% | 0% in dorsal cortex | 30% | 0% in cortical projection neurons from E18.5 on |
| Lethality at birth | 0% | 100% | 0% | 0% |
| NAPs | Breg. +2.19 mm (E18.5/16 wks) | Breg. +2.19 mm (E18.5/16 wks) | Lateral +0.72 mm (E18.5/16 wks) | Lateral +0.72 mm (E18.5/16 wks) |
| Total brain area | ns/–18% | ns/na | nd/–28% | na/–34% |
| Corpus callosum area | –24%/–52% | ns/na | nd/–55% | na/–61% |
| Anterior commissure area | –42%/–23% | –80%/na | nd/–81% | na/–51% |
| Cortical thickness | –14%/–17% | ns/na | nd/–14% | na/–44% |

*ns* not significant, *na* not applicable, *nd* not done.

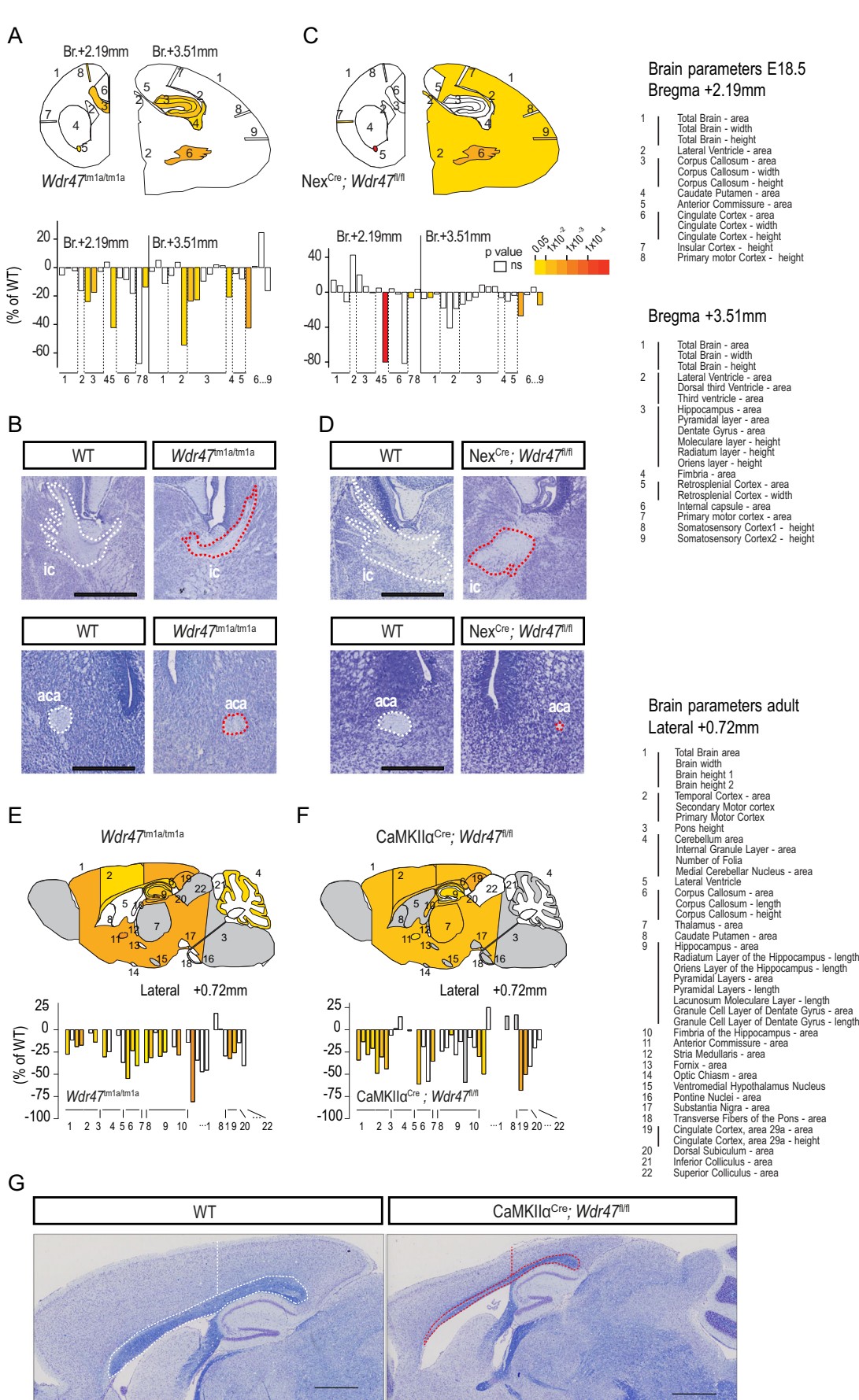

**Figure 3. Neuroanatomical studies conducted in various mouse models reveal the neuronal origin of the observed defects.**

(A) Top: Schematic representation of the 17 brain regions assessed at Bregma +2.19 mm and +3.51 mm in E18.5 $Wdr47^{tm1a/tm1a}$ mice ($n = 5$ WT $vs$ $n = 4$ $Wdr47^{tm1a/tm1a}$). Colored regions indicate the presence of at least one significant parameter within the brain region at the 0.05 level. White indicates a $p$-value > 0.05, gray shows not enough data to calculate a $p$-value. Bottom: Histograms of percentage change relative to matched WT animals (set as 0) for each of the measured parameters (listed in Dataset EV2 and on the right-hand side of the Figure). (B) Representative brain images stained with Nissl of embryonic WT and $Wdr47^{tm1a/tm1a}$ mice showing the internal capsule (ic) at Bregma +3.51 mm and the anterior part of the anterior commissure (aca) at Bregma +2.19 mm. Scale bars: 500 µm (top) and 250 µm (bottom). (C) Top: Schematic representation of the 17 brain regions assessed at Bregma +2.19 mm and +3.51 mm in E18.5 Nex$^{Cre}$;$Wdr47^{fl/fl}$ mice ($n = 6$ WT $vs$ $n = 6$ cKO). Colored regions indicate the presence of at least one significant parameter within the brain region at the 0.05 level. White indicates a $p$-value > 0.05, gray shows not enough data to calculate a $p$-value. Bottom: Barplots of percentage change relative to matched WT animals (set as 0) for each of the measured parameters. (D) Representative brain images stained with Nissl of embryonic WT and Nex$^{Cre}$;$Wdr47^{fl/fl}$ embryonic mice showing the internal capsule (ic) at Bregma +3.51 mm and the anterior part of the anterior commissure (aca) at Bregma +2.19 mm. Scale bars: 500 µm (top) and 250 µm (bottom). (E) Top: Schematic representation of the 22 brain regions quantified at Lateral +0.72 mm on parasagittal section from adult $Wdr47^{tm1a/tm1a}$ mice, aged 16 weeks mice ($n = 3$ WT $vs$ $n = 3$ $Wdr47^{tm1a/tm1a}$). Colored regions indicate the presence of at least one significant parameter within the brain region at the 0.05 level. White indicates a $p$-value > 0.05, gray shows not enough data to calculate a $p$-value. Bottom: Histograms of percentage change relative to WT mice (set as 0) for each of the measured parameters (listed in Dataset EV2 and on the right-hand side of the Figure). (F) Top: Schematic representation of the 22 brain regions quantified at Lateral +0.72 mm on parasagittal section from adult CaMKIIα$^{Cre}$;$Wdr47^{fl/fl}$ at 16 weeks of age ($n = 3$ WT $vs$ $n = 2$ cKO). Colored regions indicate the presence of at least one significant parameter within the brain region at the 0.05 level. White indicates a $p$-value > 0.05, gray shows not enough data to calculate a $p$-value. Bottom: Histograms of percentage change relative to WT mice (set as 0) for each of the measured parameters. (G) Representative brain images stained with Nissl-luxol of adult WT (left) and CaMKIIα$^{Cre}$;$Wdr47^{fl/fl}$ (right) mice showing the area of the corpus callosum and the height of the cortex at Lateral +0.72 mm. Scale bar: 1 mm. Source data are available online for this figure.

(Kannan et al, 2017), we identified NAPs in $Wdr47^{tm1a/tm1a}$ constitutive E18.5 KO embryonic brains (Fig. 3A) including CCD ($-24\%$, $p = 0.04$), hypoplasia of the hippocampus, thinning of the motor cortex, and reduced size of the internal capsule (ic) and anterior part of the anterior commissure (aca) (Fig. 3B and Table 1). However, unlike in $Wdr47^{tm1a/tm1a}$ constitutive E18.5 KO, Nex$^{Cre}$;$Wdr47^{fl/fl}$ embryonic brains did not reveal any change in the CC area, width or height despite the strong reduction in the size of other commissures including the internal capsule ($-27\%$, $p = 0.0019$) and the anterior commissure ($-80\%$, $p = 4.56E-07$) (Fig. 3C,D and Table 1). This suggests that the 24% decrease in the size of the CC observed in the constitutive KO mice stem from defects likely involving non-neuronal cells (such as glial cells or radial glia progenitors) at early stages of CC development. Similar to constitutive KO mice (Kannan et al, 2017) and a previous report (Chen et al, 2020), Nex$^{Cre}$;$Wdr47^{fl/fl}$ pups died soon after birth confirming that the neonatal death of the $Wdr47$-deficient mice is attributable to brain defects.

The neonatal death of the Nex$^{Cre}$;$Wdr47^{fl/fl}$ pups did not allow to study the neuron-specific involvement of $Wdr47$ at later stages of CC development. To circumvent this problem, we used the calcium/calmodulin-dependent protein kinase II alpha subunit CaMKIIα$^{Cre}$ line (Dragatsis and Zeitlin, 2000) to delete $Wdr47$ specifically in post-mitotic neurons in the forebrain starting from E18.5 (Cook et al, 2018) (Appendix Fig. S2A). With respect to the 3Rs (replacement, reduction, and refinement) principles to limit animal use, we studied brain anatomy using one time point in adult CaMKIIα$^{Cre}$; $Wdr47^{fl/fl}$ (16 weeks of age) as the mice were viable. RT-qPCR analysis confirmed the decreased expression of $Wdr47$ transcripts in adult cortices (Appendix Fig. S2C). Using our established pipeline for the neuroanatomical quantification of 22 brain structures in the adult mouse totaling 40 measurements at Lateral +0.72 mm (Collins et al, 2018), we identified a similar profile of NAPs in CaMKIIα$^{Cre}$; $Wdr47^{fl/fl}$ and $Wdr47^{tm1a/tm1a}$ adult mice with the exception of the cerebellar anomalies which were absent in the CaMKIIα$^{Cre}$; $Wdr47^{fl/fl}$ mice (Fig. 3E,F). The corpus callosum was smaller in size by 61% ($p = 0.008$) and 55% ($p = 0.027$) and the total brain area by 34% ($p = 0.0095$) and 28% ($p = 0.014$) in CaMKIIα$^{Cre}$; $Wdr47^{fl/fl}$ and $Wdr47^{tm1a/tm1a}$ adult mice, respectively (Fig. 3G and Table 1). These combined findings in conditional and

constitutive mouse models showed that, although non-neuronal cells likely contribute to prenatal CC abnormalities, there is a major involvement of post-mitotic neurons to neuroanatomical phenotypes in young adult (Table 1).

## $Wdr47$ deficiency leads to perinatal loss of callosal interhemispheric connections by impairing neuronal survival

Considering the presence of CCD in all the patients (Dataset EV1) and severe CC anomalies in $Wdr47$-deficient mice (Table 1), we next focused on the development of callosal projection neurons (CPNs) that extend axons toward the midline that in fine innervate the contralateral cortex through extensive branching. We performed acute depletion of $Wdr47$ specifically in callosal neurons using in utero electroporation (IUE) of vectors allowing the expression of Cre-GFP or GFP under the control of the NeuroD promoter together with the pCAG2-mScarlet reporter plasmid in $Wdr47^{fl/fl}$ or $Wdr47^{fl/WT}$ embryonic cortices at E15.5, when the CPNs are born. At postnatal day 2 (P2), soon after the axons cross the midline, the proportion of neurons able to project their axon was decreased upon $Wdr47$ depletion as indicated by the reduced density (-55%) of scarlet-positive axons in the white matter (WM) of $Wdr47^{fl/fl}$ (IUE NeuroD-Cre-GFP) cortices compared to controls (NeuroD-GFP in $Wdr47^{fl/fl}$ or $Wdr47^{fl/WT}$) (Fig. 4A–C). In addition, almost one third of the $Wdr47$-deficient axons failed to reach the midline (31% of axons arriving to the midline in controls compared to 11% in $Wdr47^{fl/fl}$ pups after IUE with Cre) (Fig. 4A,B,D). However, once they reached the midline, axons depleted for $Wdr47$ showed similar ability to cross the midline than control axons (Fig. 4A,B,E). Of note, no aberrant axonal projection or midline crossing were observed in $Wdr47^{fl/WT}$ pups electroporated with NeuroD-Cre-GFP (Fig. 4A–E) suggesting that $Wdr47$ is haplosufficient for axon extension. Strikingly, at P4, while axons extended further into the contralateral hemisphere in control and heterozygous (NeuroD-Cre-GFP in $Wdr47^{fl/WT}$) condition, the proportion of scarlet-positive axons at the midline was severely decreased upon $Wdr47$ deletion (IUE Neuro-Cre-GFP in $Wdr47^{fl/fl}$) (Fig. 4F). A closer examination of $Wdr47$-deficient neurons in the upper cortical plate (uCP) revealed aberrant cellular appearance with loss of bipolar organization and fragmented neurites

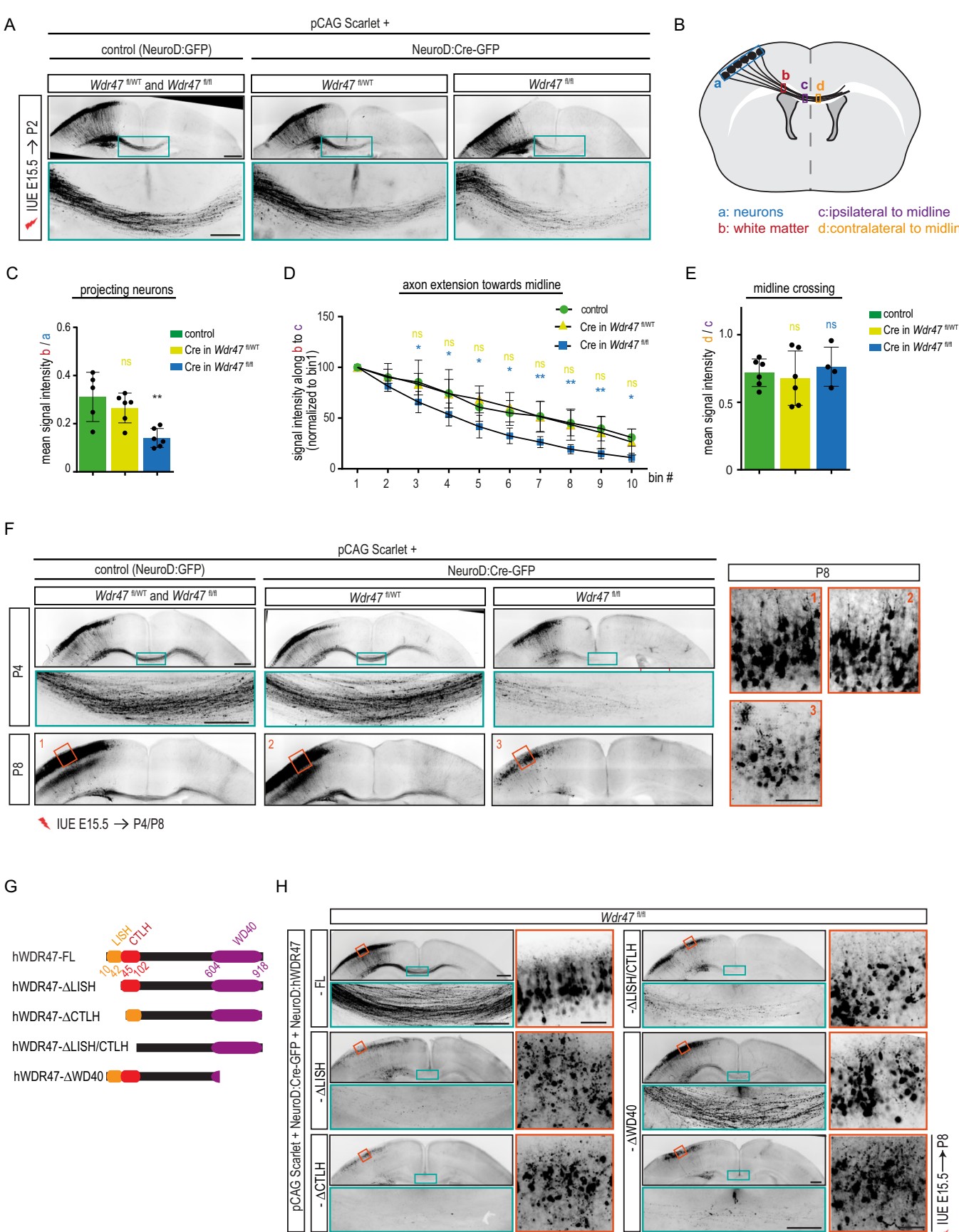

**Figure 4. Deletion of *Wdr47* impedes interhemispheric connectivity through impaired neuronal survival.**

(A) Coronal sections of P2 mouse brains electroporated at E15.5 with pCAG:Scarlet plasmid together with either a control (NeuroD:Ires:GFP) or a NeuroD:Cre-GFP vector. Scarlet positive electroporated neurons are depicted in black. Close-up views of the green boxed area show axon extension defects in *Wdr47*fl/fl pups but not in *Wdr47*fl/WT pups or control conditions. Scale bars: 500 μm and 200 μm (green boxed inset). (B) Schematic describing the methods used to quantify (C) the percentage of projecting neurons (mean intensity of the scarlet signal in the white matter (red box) normalized on the mean intensity in the cortical plate (blue box)), (D) axon extension towards midline (intensity was plotted along a line that was divided into 10 equal bins from the immediate start of CC (red box) till the midline (purple box); intensity in bin1 was considered 100% and intensity in each bin was normalized to bin1) and (E) midline crossing (mean intensity of the scarlet signal just after the midline (yellow box) is normalized on the mean intensity just before midline crossing (purple box)). (C–E) Quantification of (C) projecting neurons, (D) axon extension towards midline, and (E) midline crossing. Data (means ± s.d.) from at least 4 pups from 2 to 3 different litters per condition were analyzed by (C, E) one-way ANOVA, with Bonferroni's multiple comparison test, or (D) two-way ANOVA, with Bonferroni's multiple comparison test. (F) Coronal sections of P4 and P8 mouse brains electroporated at E15.5 with pCAG:Scarlet plasmid together with either a control (NeuroD:Ires:GFP) or a NeuroD:Cre-GFP vector. Scarlet positive electroporated neurons are depicted in black. Close-up views of the green and orange boxed area show loss of CC at P4 and loss of neuronal morphology at P8, respectively, in *Wdr47*fl/fl pups but not in *Wdr47*fl/WT pups or control conditions. At least 4 pups from 2 to 3 different litters per condition was analyzed. Scale bars: 500 μm, 200 μm (blue boxed inset) and 100 μm (red boxed inset). (G) Schematic representation of the WDR47 protein depicting different domains and the truncated Wdr47 constructs used in (H) to rescue the phenotype induced by the loss of *Wdr47*. (H) Coronal sections of P4 and P8 mouse brains electroporated at E15.5 with pCAG:Scarlet plasmid and NeuroD:Cre-GFP vector together with a truncated hWDR47 construct. Scarlet positive electroporated neurons are depicted in black. Close-up views of the green and orange boxed area show that at P4 ΔLISH, ΔCTLH, and ΔLISH/CTLH fail to rescue the loss of CC and neuronal morphology while ΔWD40 construct could partially restore the survival defects. Scale bars: 500 μm, 200 μm (blue boxed inset) and 50 μm (red boxed inset). ns, non-significant, *P < 0.05, **P < 0.01. Exact P values are listed in Dataset EV4. Source data are available online for this figure.

(Fig. EV2A) as well as presence of several pyknotic nuclei and apoptotic cells (positive for activated Caspase 3) at P3 and P4 but not at P2 (Fig. EV2A,B). As a consequence, at P8, most of the *Wdr47*-depleted neurons have disappeared and the remaining neurons failed to maintain their bipolar morphology and axonal projections as shown by the absence of scarlet-positive axons at the midline (Fig. 4F, insets). Altogether, these results suggested that WDR47 has a pro-survival function in callosal neurons at perinatal stages.

WDR47 is a multidomain protein with N-terminally located LISH and CTLH domains and seven C-terminally located WD40 domains (Fig. 4G). Therefore, to study which of those domains could mediate the pro-survival function of WDR47, we tested the ability of truncated WDR47 constructs to rescue the phenotype induced by the loss of *Wdr47*. We performed IUE of NeuroD-Cre-GFP and pCAG2-mScarlet plasmids together with either a NeuroD-driven full length (hWDR47-FL) or truncated human WDR47 construct lacking different domains in E15.5 *Wdr47*fl/fl or *Wdr47*fl/WT embryos and analyzed the scarlet-positive neurons and their axons at P4. In accordance with a cell autonomous function of *Wdr47*, introduction of WDR47-FL construct rescued the survival defects (Fig. 4H). Expression of all N-terminal deletion proteins (hWDR47-ΔLISH, -ΔCTLH, and -ΔLISH/CTLH) failed to rescue CC phenotype, as assessed by the absence of Scarlet positive axons in the midline (Fig. 4H). Although *Wdr47*-deficient neurons expressing truncated WDR47 proteins that lack the WD40 domain (WDR47-ΔWD40) were able to project their axons toward the contralateral cortex as the WDR47-FL expressing neurons, they started displaying signs of degeneration at P4 (Fig. 4H, insets). At P8, most of the axons at the midline disappeared indicating that expression of WDR47-ΔWD40 likely delays the onset of the CC phenotype (Fig. 4H). Of note, 10 times less of WD40 plasmid was used to reach similar expression level between all the constructs (Fig. EV2C). Given the absence of phenotypes after IUE of Cre in *Wdr47*fl/WT pups at P4 (Figs. 4F and EV2A,B), we used them as controls and showed that none of those constructs impair neuronal survival in control condition (Fig. EV2D). Altogether these results indicate that WDR47 is required cell autonomously to protect CPNs from sudden neuronal death in vivo and that both N- and

C-terminal domains of Wdr47 are indispensable for its pro-survival function.

## Loss of *Wdr47* causes neuronal death independently of its role in migration and axonal extension

As Wdr47 was previously shown to regulate radial migration (Chen et al, 2020; Kannan et al, 2017) and as wound healing assay performed using fibroblasts derived from M01 confirmed impaired collective cell migration in the disease context (Fig. EV3A,B), we further investigated whether faulty migration at early stages could drive cell death of callosal neurons at later stages. First, we investigated whether the roles of Wdr47 in cell survival and neuronal migration involve the same domains. We tested for the restoration of migration phenotype induced by the loss of *Wdr47* (Chen et al, 2020; Kannan et al, 2017) by introducing FL or truncated hWDR47 constructs together with NeuroD-Cre-GFP using IUE in E14.5 *Wdr47*fl/fl embryos and quantified the distribution of GFP-positive neurons in the cortical plate 4 days later. As we previously did not observe any difference in migration between control and *Wdr47* heterozygous neurons (Kannan et al, 2017), we used IUE of NeuroD-Cre-GFP in *Wdr47*fl/WT embryos as control. Consistent with previous reports (Chen et al, 2020; Kannan et al, 2017), acute depletion of *Wdr47* resulted in severe migration defects with 54% of cell reaching the upper cortical plate (uCP) compared to 77% in control condition (Fig. 5A,B). While introduction of FL-hWDR47 construct fully restored the migration (73% GFP+ cells in uCP), none of the N-terminal truncated constructs could rescue the phenotype (53%, 48%, 51%, GFP+ cells in the uCP for the WDR47-ΔLISH, -ΔCTLH, and -ΔLISH/CTLH constructs, respectively) (Fig. 5A,B). At the opposite, Wdr47 protein lacking the C-terminal WD40 domain partially rescued the migration phenotype (70% GFP+ cells in uCP) (Fig. 5A,B). Of note, expression of none of those constructs affected migration in *Wdr47*fl/WT embryos (Fig. EV3C,D). These results suggested that N-terminally located LISH and CTLH domains likely mediate functions of WDR47 in both neuronal migration and survival whereas WD40 repeats are at least partially dispensable for migration.

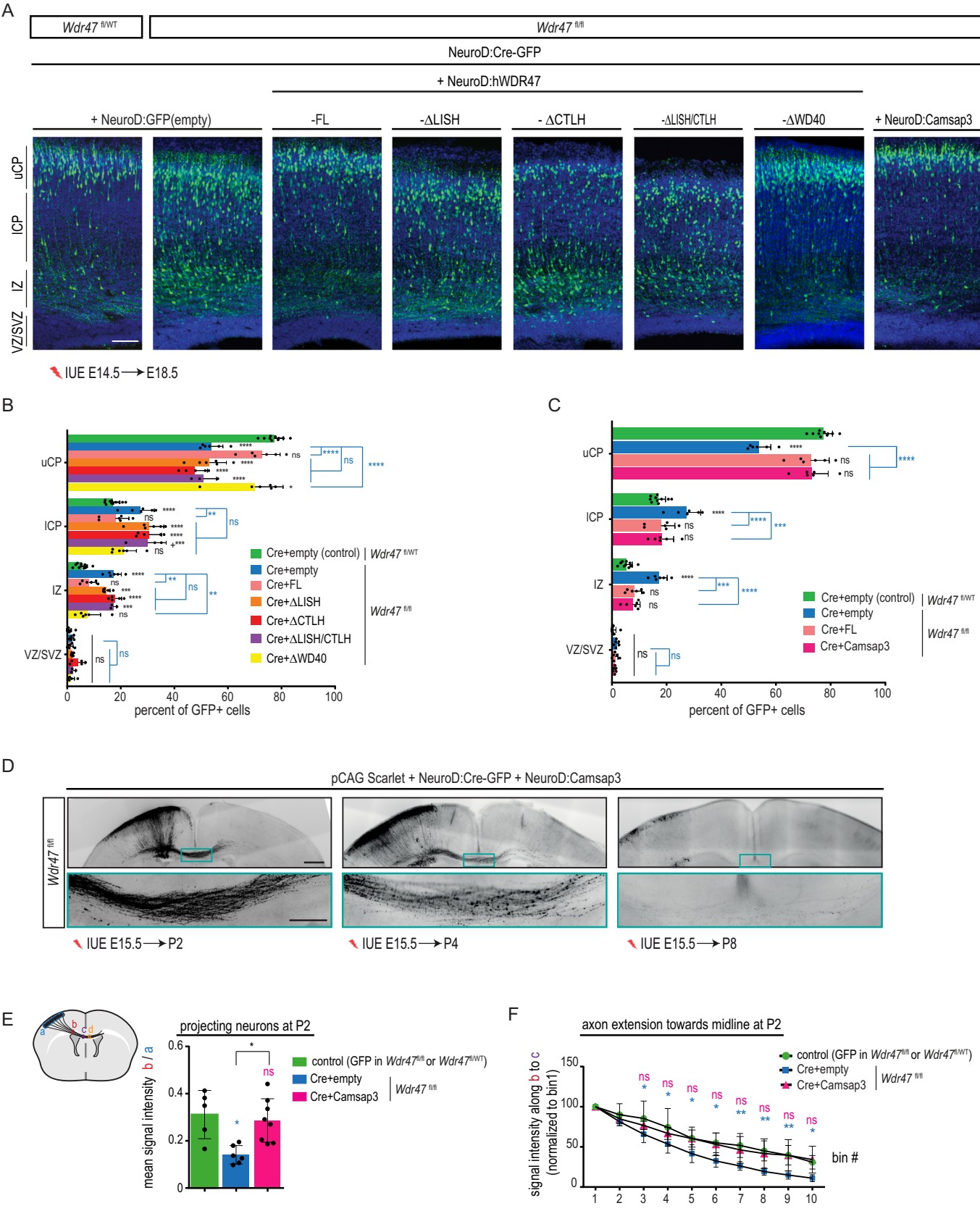

**Figure 5.   Function of WDR47 in neuronal survival is independent from its function in neuronal migration and axon extension.**

(A) Coronal sections of E18.5 mouse cortices, four days after in utero electroporation with NeuroD-Cre-GFP together with either NeuroD:empty vector or a different rescue vector (either NeuroD:hWDR47-FL or a truncated NeuroD:hWDR47 construct or NeuroD:Camsap3). GFP-positive electroporated cells are depicted in green. Nuclei are stained with DAPI. Scale bar: 100 μm. (B, C) Analysis of the percentage of electroporated GFP-cells in different regions (uCP, lCP, IZ, and VZ/SVZ) showing effect of introducing (B) different truncated Wdr47 constructs and (C) Camsap3 on neuronal migration. Data (means ± s.d.) from at least three embryos from 2 to 3 different litters per condition was analyzed by two-way ANOVA, with Bonferroni's multiple comparisons test. uCP, Upper cortical plate; lCP, Lower cortical plate; IZ, intermediate zone; VZ, ventricular zone; SVZ, subventricular zone. (D) Coronal sections of P2, P4, and P8 mouse brains electroporated at E15.5 with pCAG:Scarlet, NeuroD:Cre-GFP and NeuroD:Camsap3 plasmids. Scarlet positive electroporated neurons are depicted in black. Close-up views of the green boxed area show that axon extension defects are rescued upon introduction of Camsap3 at P2, however, CC is still lost at P8. Scale bars: 500 μm and 200 μm (green boxed inset). (E, F) Quantification of (E) projecting neurons and (F) axon extension towards midline. Data (means ± s.d.) from at least 5 pups per condition were analyzed by (E) one-way ANOVA, with Bonferroni's multiple comparison test, (F) two-way ANOVA, with Bonferroni's multiple comparison test. ns, non-significant; *$P < 0.05$; **$P < 0.01$; ***$P < 0.001$; ****$P < 0.0001$. Exact $P$ values are listed in Dataset EV4. Source data are available online for this figure.

Next, as Camsap3, a microtubule (MT) minus end binding protein that recruits WDR47 to MT minus ends, had been shown to rescue the migration and neuronal polarization defects of *Wdr47*-deficient neurons (Chen et al, 2020), we tested whether rescuing the migration and axonal extension defects by introducing Camsap3 in *Wdr47*-deficient neurons would also rescue neuronal survival defects. We expressed mouse Camsap3 together with NeuroD-Cre-GFP in *Wdr47*[fl/fl] embryos by IUE at E14.5 and analyzed the distribution of GFP+ cells at E18.5. In parallel, we performed IUE at E15.5 and assessed axonal projection at the midline at P2 and neuronal survival at P4 and P8. Interestingly, although over-expression of Camsap3 fully restored the migration defects at E18.5 (Fig. 5A,C) (Chen et al, 2020) as well as axon extension defects at P2 and P4 (Fig. 5D–F), it only slightly delayed neuronal death as shown by the absence of CC and severe degeneration of neurons at P8 (Fig. 5D). Moreover, the analysis of CC of Camsap3 adult KO mice revealed a thinner CC, likely due to the specific loss of medial fibers as no notable differences were observed in the lateral fibers connecting the somatosensory cortex (Fig. EV3E and Dataset EV2). This is in contrast with the agenesis of the corpus callosum observed in *Wdr47*[tm1a/tm1a] KO mice (Kannan et al, 2017), supporting that only some of the WDR47-related functions during brain development are mediated by Camsap3.

To sum up, our results showed that although they rely on the same domains, pro-survival function of WDR47 is independent of its roles in migration and axonal growth and involves downstream effectors other than Camsap3.

## Wdr47 protects callosal neurons from apoptosis

We next sought to understand the mode of cell death induced upon loss of *Wdr47*. As we previously reported a negative correlation between WDR47 expression and activation of autophagy (Kannan et al, 2017), we tested whether autophagic cell death could cause the survival defects induced by loss of *Wdr47*. We turned to the yeast model and addressed first the requirement of LISH/CTLH domains for yeast growth as a proxy of survival. While expression of human WDR47-FL allowed normal growth (Kannan et al, 2017), expression of hWDR47-ΔLISH or -ΔCTLH constructs strongly impaired growth in WT yeast (Appendix Fig. S3A–C), reminiscent of their requirement for the survival of cortical neurons. Then, we investigated the involvement of those domains in autophagy inducing conditions upon nitrogen starvation, by following the yeast Atg8 (homologue to mammalian LC3) reporter. In line with a role of WDR47 in inhibiting autophagy (Kannan et al, 2017), mCherry-Atg8 did not accumulate in

the lumen of the vacuole and instead remained cytoplasmic upon expression of full length WDR47 (Appendix Fig. S3D). Surprisingly, expression of hWDR47-ΔLISH and -ΔCTLH inhibited accumulation of mCherry-Atg8 in the vacuolar lumen similarly to the FL construct suggesting that those 2 domains, that are indispensable for the function of WDR47 in survival of cortical neurons (Fig. 4H) and yeast (Appendix Fig. S3B,C), are not required for inhibiting autophagy (Appendix Fig. S3D). According to an autophagy independent mechanism, expression of Wdr47-ΔLISH and -ΔCTLH constructs in the Δoxa1 yeast strain that is deficient in respiration and that lacks mitochondrial DNA (Bonnefoy et al, 1994) did not lead to any growth defects (Appendix Fig. S3B,C) suggesting that growth defects induced by those constructs likely involve mitochondrial- rather than autophagy-dependent mechanisms.

To further identify which mode of cell death is triggered upon *Wdr47*-deficiency, we used a simplified in vitro system that allows to study dynamics of neuronal death at the single cell resolution. We transfected primary neuronal cultures established from cortices of E15.5 *Wdr47*[WT/WT] and constitutive *Wdr47*[tm1b/tm1b] knock-out embryos (hereafter named WT and HOM, respectively) with a pCAG2-mScarlet plasmid. Monitoring of individual scarlet-positive neuron from day in vitro (DIV) 3 to DIV10 by time lapse imaging (Fig. 6A), revealed a burst of neuronal death starting at DIV8 in HOM cultures leading to a decrease by half of the number of surviving neurons in HOM cultures compared to WT at DIV10 (Fig. 6A–C). Treatment of HOM primary neurons with Qvd-Oph, a pan-Caspase inhibitor, at DIV2, fully rescued the death of *Wdr47* knock-out neurons at DIV10 (62% survival in Qvd-Oph-treated HOM cultures compared to 69% survival in WT cultures) (Fig. 6B,C), whereas treatment with Necrostatin and Ferrostatin, necroptosis and ferroptosis inhibitors, respectively, did not have any effect on survival (34%, 36%, and 49% survival in untreated, necrostatin- and ferrostatin-treated HOM cultures, respectively) (Fig. 6B,C). Of note, none of the drugs affected survival of WT neurons (Appendix Fig. S3E,F). Taken together, these results demonstrated that loss of *Wdr47* leads to a severe, Caspase-dependent, early-onset burst of neuronal death that manifests as CC dysgenesis in vivo.

## *Wdr47*-deficient neurons present a neurodegenerative signature at the molecular level

To gain molecular insights into the mechanisms underlying neuronal death induced by the loss of *Wdr47*, we compared the global transcriptome of *Wdr47*[WT/WT] (WT) and *Wdr47*[tm1b/tm1b]

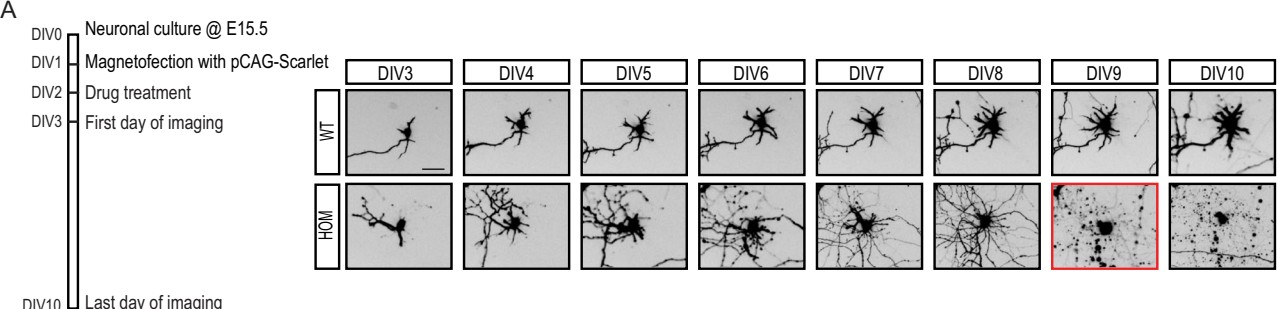

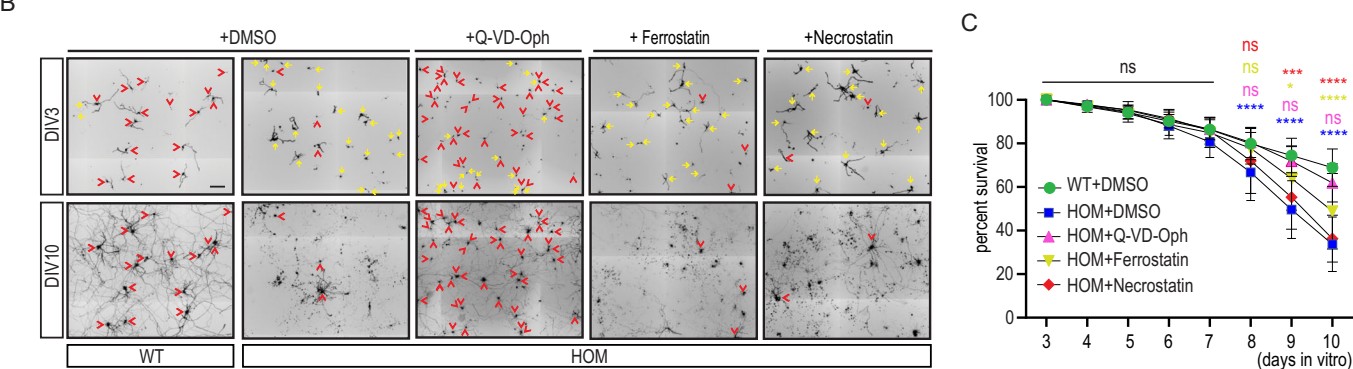

**Figure 6. Wdr47 deficient neurons die via apoptosis.**

(A) Left panel: Schematic of the viability analysis in primary neuronal cultures. Right panel: Fluorescence imaging depicting single scarlet positive neurons (in black) that were followed from DIV3 to DIV10 in primary cultures obtained from WT and HOM (*Wdr47*<sup>tm1b/tm1b</sup>) embryos. Red rectangle indicates the time of neuronal death. Scale bar: 50 µm. (B, C) Neuronal survival in vitro in WT and HOM (*Wdr47*<sup>tm1b/tm1b</sup>) primary neuronal cultures upon treatment with different drugs. (B) Representative fields, at DIV3 and DIV10, of neuronal cultures treated with DMSO, Qvd-OPh (Caspase inhibitor), Ferrostatin (ferroptosis inhibitor), and Necrostatin (Necroptosis inhibitor) at DIV2. Scarlet positive electroporated neurons are depicted in black. Yellow arrows correspond to neurons that died, red arrowheads correspond to neurons that are alive and could be followed from DIV3 to DIV10. Scale bar: 200 µm. (C) Survival of neurons from DIV3 to DIV10 upon different drug treatments. Data (means ± s.d.) from at least 3 independent cultures per condition was analyzed by two-way ANOVA, with Bonferroni's multiple comparison test, ns, non-significant, *$P < 0.05$, ***$P < 0.001$, ****$P < 0.0001$. Exact $P$ values are listed in Dataset EV4. Source data are available online for this figure.

knock-out (HOM) cortical neurons at DIV6, 2 days before the initiation of neuronal death using mRNA sequencing. Differential gene expression analysis identified 46 upregulated and 159 downregulated transcripts in HOM neurons (adjusted $p$ value < 0.05) (Fig. 7A and Dataset EV3). Gene Ontology (GO) enrichment analysis on differentially expressed genes (DEGs) revealed that, while upregulated genes were not enriched for any terms, downregulated genes were enriched in several biological process terms including regulation of synaptic plasticity (qvalue = 0.005), regulation of neuron apoptotic process (qvalue = 0.049), neuron projection extension (qvalue = 0.042), postsynaptic specialization organization/assembly (qvalue = 0.0003) and regulation of microtubule polymerization or depolymerization (qvalue = 0.047) (Fig. 7B). Further enrichment analysis of disease ontology terms indicated on one hand that upregulated genes were mainly associated to neurodevelopmental deficits including various forms of intellectual disability and on the other hand that downregulated genes were linked to neurodegenerative diseases (e.g., Alzheimer's and Parkinson's disease) (Fig. 7C), in line with a degenerative phenotype occurring at early stage.

In accordance, among the deregulated genes we found genes encoding for proteins involved in pathways critical to maintain neuronal homeostasis including intracellular trafficking (Cytoskeleton subunits, Tuba4A, Tubb4A, Tubb4B and Nefl; molecular

motors, Dyncl1l and Klc2), mitochondrial function (Uqcrq, Ndufs7 and Trap1) and response to stress (Sod1 and Atf4) (Fig. 7A). Downregulation of all those genes but Sod1 was confirmed by quantitative polymerase chain reaction (RT-qPCR) (Fig. 7D) in *Wdr47*-deficient neurons at DIV6. Interestingly, in accordance with a sudden burst of cell death, none of those genes were differentially expressed in HOM cultures at DIV4 (Fig. EV4A). Collectively, these data highlighted several key cellular mechanisms whose combined deregulation upon loss of *Wdr47* likely contributes to rapid neuronal death.

## Loss of Wdr47 impairs mitochondrial homeostasis in neurons

Given the presence of *Ndufs7* and *Uqcrq*, members of electron transport chain and *Trap1*, a mitochondrial chaperone that regulates mitochondrial homeostasis and bioenergetics among the downregulated genes in *Wdr47* KO neurons (Fig. 7A,D) and the causative link between mitochondrial dysfunction and neuronal death (Flippo and Strack, 2017; Mattson et al, 2008), we tested whether mitochondrial impairment could be one the mechanisms leading to the early death of *Wdr47*-deficient neurons. Live imaging of DIV6 WT and *Wdr47* KO neurons after transfection of pCAG-GFP and pCAG-mito-dsRed constructs to label axons and mitochondria, respectively,

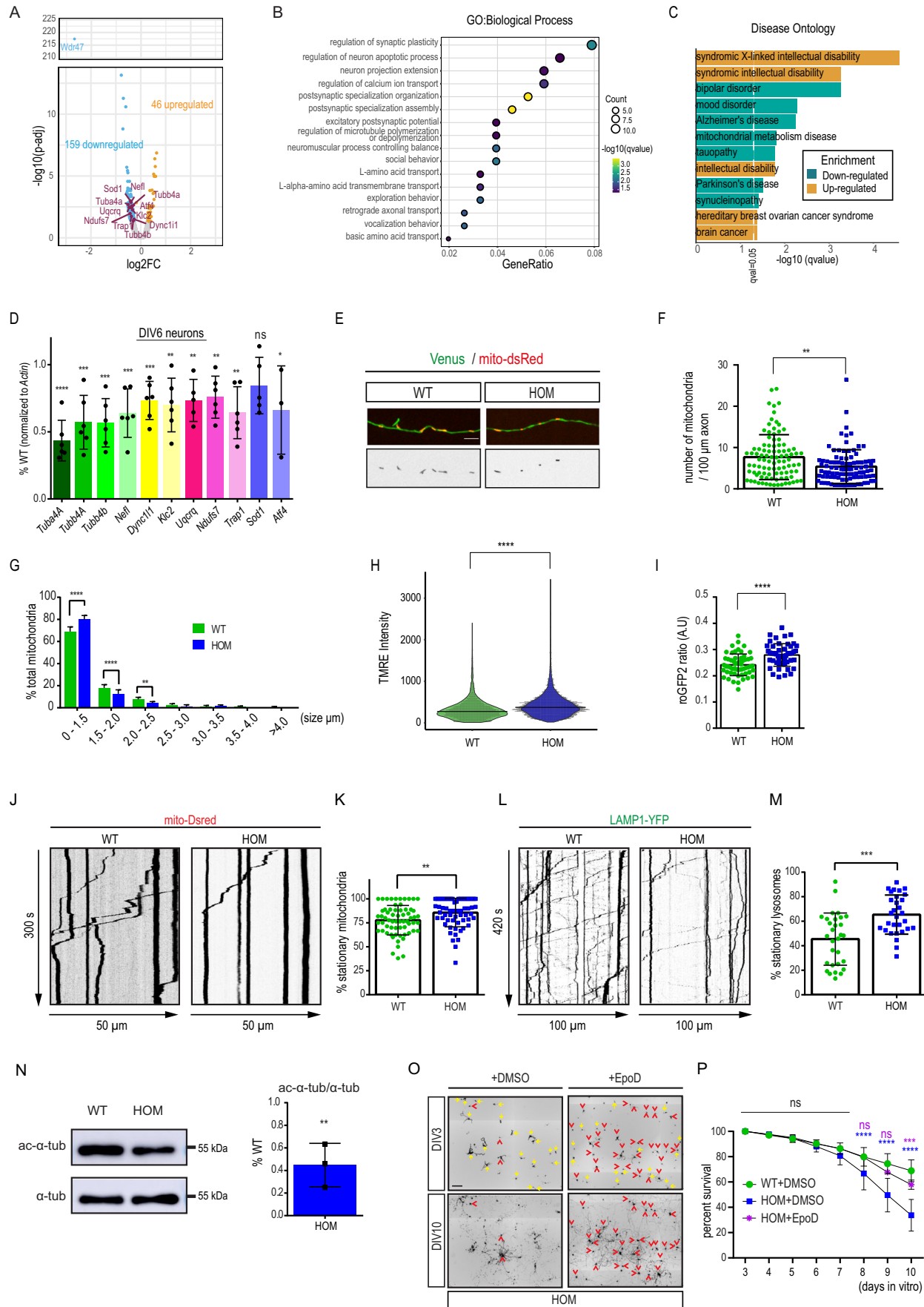

◀ **Figure 7. *Wdr47* deficient neurons present neurodegenerative hallmarks with altered mitochondrial and microtubule homeostasis.**

(A) Volcano plot showing the negative $\log_{10}$ adjusted *P*-value (p-adj) of all genes against their $\log_2$ fold change ($\log_2$FC) (*Wdr47*^tm1b/tm1b^ versus WT neurons). Upregulated and downregulated genes (p-adj < 0.05) are in orange and blue, respectively, and the neurodegeneration-related genes selected for validation of RNA-seq results are labeled in red. Data from DIV6 cultures of 2 WT and 3 HOM E15.5 embryos were analyzed using Wald test with *p*-values adjusted for multiple testing using the Benjamini and Hochberg method. (B) GO term analysis of downregulated genes in DIV6 neurons in *Wdr47*^tm1b/tm1b^ versus WT. The size of the circle represents the number of genes enriched in the GO term and the color of the circles represents the $-\log_{10}$ (qvalue). (C) Disease Ontology analysis of dysregulated genes in DIV6 neurons in *Wdr47*^tm1b/tm1b^ versus WT. Diseases enriched among upregulated and downregulated genes are depicted in yellow and blue, respectively. Dashed line indicates qvalue equal to 0.05. (D) RT-qPCR validation of RNA-seq results for 11 downregulated genes associated to neurodegenerative diseases. Data (means ± s.d.) from at least 3 independent cultures per condition was analyzed by unpaired t-test. (E–G) Assessment of mitochondria in axons of WT and HOM (*Wdr47*^tm1b/tm1b^) DIV6 cortical neurons magnetofected with pCAG-GFP and mito-dsred. (E) Live-cell confocal imaging of individual mitochondria (red) in single axons (green) of WT and HOM (*Wdr47*^tm1b/tm1b^) DIV6 cortical neurons. Scale bar: 5 μm. (F) Number of mitochondria per 100 μm of axon in DIV6 cortical neurons (*n* > 90 axons per condition). Data (means ± s.d.) from 5 independent cultures per condition was analyzed by unpaired t-test with Welch's correction. (G) Quantitative assessment of mitochondrial morphology (Feret's diameter) in axons of DIV6 cortical neurons (*n* > 400 mitochondria per condition). Data (means ± s.d.) from 5 independent cultures per condition was analyzed by two-way ANOVA. (H) Mitochondrial membrane potential (ΔΨm) (TMRE intensity) of individual mitochondria in the neurites of WT and HOM (*Wdr47*^tm1b/tm1b^) DIV6 cortical neurons (*n* > 3500 mitochondria per condition). Data from at least 3 independent cultures was analyzed by Mann Whitney test. (I) Redox potential (mito-Grx1-roGFP2) of mitochondria in the neurites of WT and HOM (*Wdr47*^tm1b/tm1b^) DIV6 cortical neurons (*n* > 50 neurites per condition). Data (means ± s.d.) from 3 independent cultures per condition was analyzed by unpaired t-test with Welch correction. (J–M) Neurons were magnetofected at DIV4 with (J, K) mito-Dsred and (L, M) Lamp1-YFP to analyze mitochondria and lysosome motility, respectively, by videomicroscopy. Kymographs illustrate the motility of (J) mitochondria and (L) lysosomes in time (y, sec) and space (x, μm). Histograms represent the percentage of stationary (K) mitochondria (*n* > 60 axons per condition from 5 independent cultures) and (M) lysosomes (*n* = 30 axons per condition from 4 independent cultures). Data (means ± s.d.) was analyzed by unpaired t-test. (N) Western blot analysis shows decreased levels of acetylated tubulin in DIV6 HOM (*Wdr47*^tm1b/tm1b^) primary neurons compared to WT. Data (means ± s.d.) from 3 independent cultures was analyzed by unpaired t-test. (O, P) Neuronal survival in vitro in WT and HOM (*Wdr47*^tm1b/tm1b^) primary neuronal cultures upon treatment with MT stabilizer EpoD. (O) Representative fields, at DIV3 and DIV10, of neuronal cultures treated with DMSO EpoD at DIV2. Scarlet positive electroporated neurons are depicted in black. Yellow arrows correspond to neurons that died, red arrowheads correspond to neurons that are alive and could be followed from DIV3 to DIV10. The images shown for the DMSO condition are identical to those presented in Fig. 6B. Scale bar: 200 μm. (P) Survival of neurons from DIV3 to DIV10 upon treatment with DMSO and EpoD (10 nM). Data (means ± s.d.) from at least 3 independent cultures per condition was analyzed by two-way ANOVA, with Bonferroni's multiple comparison test. ns, non-significant, *$P < 0.05$, **$P < 0.01$, ***$P < 0.001$, ****$P < 0.0001$. Exact *P* values are listed in Dataset EV4. Source data are available online for this figure.

revealed that *Wdr47* deficient axons contain fewer mitochondria (Fig. 7E,F). In addition, the relative distribution of mitochondrial size is shifted towards smaller mitochondria in *Wdr47*-lacking neurons (Fig. 7G), indicating an increased prevalence of fragmented mitochondria, a hallmark of damaged mitochondria. Then, to further assess mitochondrial integrity, we measured the mitochondrial inner membrane potential (ΔΨm) in WT and *Wdr47* KO neurons by performing dual imaging of TMRE (tetramethylrhodamine, ethyl ester) and mitotracker green, two dyes that allow labeling of mitochondria in a potential-dependent and independent manner, respectively. Surprisingly, we found that in neurites of DIV6 *Wdr47*-deficient neurons, mitochondria showed 36% increased TMRE signal (mean TMRE intensity of 274.3 in WT and 373.4 in *Wdr47* deficient neurons) (Fig. 7H) reflecting higher membrane potential. As mitochondrial hyperpolarization can lead to an increase in reactive oxygen species (ROS) production (Esteras et al, 2017), we employed mitochondrial-targeted ratiometric roGFP2 probe (mito-roGFP2) (Gutscher et al, 2008) to measure the mitochondrial redox potential and observed a 15% increase of the roGFP2 ratio in *Wdr47*-deficient neurites compared to WT neurites indicating increased ROS production in mutant neurons (Fig. 7I). Reinforcing a function of WDR47 as a regulator of mitochondrial homeostasis in human, FACS analysis performed in *WDR47*-KO HeLa cells confirmed the loss of mitochondrial mass and impaired membrane potential upon complete *WDR47* deletion (Fig. EV4B,C). Collectively, these results raise the possibility that the mammalian WDR47 exerts its pro-survival activity through regulation of mitochondrial function.

## Compromised axonal transport and MT stability precede degeneration of *Wdr47*-deficient neurons

Given the critical role of axonal trafficking in neuronal homeostasis (Brady and Morfini, 2017) and the deregulation of genes associated to axonal transport in *Wdr47*-KO neurons (Fig. 7A,B,D), we

hypothesized that Wdr47 survival-related function could also be mediated through regulation of axonal transport. We therefore analyzed axonal transport of mitochondria and lysosomes in WT and *Wdr47* deficient neurons after transfection of pCAG-mito-dsRed and Lamp1-GFP constructs, respectively. Similar to previous reports (Farfel-Becker et al, 2019; Lewis et al, 2016) in axons of WT neurons, 78% of mitochondria and 45% of lysosomes were stationary, whereas upon loss of *Wdr47*, the fraction of stationary mitochondria and lysosomes increased to 86% and 65%, respectively (Fig. 7J–M). Nonetheless, apart from a mild decrease in the velocity of anterograde moving mitochondria, velocity of the motile organelles was unchanged (Fig. EV4D,E). Similar defects in lysosomal dynamics were observed in fibroblasts derived from M01 patient and *WDR47*-KO HeLa cells (Fig. EV4F,G). As WDR47 has been shown to interact with and stabilize the microtubule (MT) cytoskeleton (Buijs et al, 2021; Chen et al, 2020; Wang et al, 2012), we reasoned that the faulty axonal transport observed in *Wdr47*-deficient neurons (Fig. 7J–M) could stem from an unstable microtubule network. In line with this hypothesis, the term "*regulation of microtubule polymerization/depolymerization*" was among the deregulated GO biological processes (Fig. 7B) and several tubulin members, the major component of MTs, were downregulated upon *Wdr47* deficiency in the RNA-seq and qPCR (Fig. 7A–D). We therefore assessed the levels of stabilized (acetylated) and dynamic (tyrosinated) MTs in primary cultures of WT and *Wdr47*^tm1b/tm1b^ HOM neurons at DIV6 and showed that levels of acetylated tubulin were decreased by more than half (Fig. 7N) while the total alpha tubulin and tyrosinated alpha tubulin levels were not altered (Fig. EV4H). Accordingly, levels of acetylated tubulin were also severely affected in fibroblasts derived from M01 patient (p.(Arg193His)) and *WDR47*-KO HeLa cells (Fig. EV4I,J). Further analysis of MT repolymerization in M01 fibroblast after nocodazole washout revealed impaired recovery of the MT network with a high percentage of cells presenting with

disorganized MT cytoskeleton compared to the controls (Fig. EV4K). Collectively, these data confirm the instability of the MT network upon *Wdr47* depletion. Finally, we tested the ability of Epothilone D (EpoD), a MT stabilizing compound (Brunden et al, 2010), to rescue the survival defects induced by the loss of *Wdr47*. We treated WT and HOM neurons at DIV2 and monitored their survival over 10 days. We demonstrated that while it did not have any effect on the survival of WT neurons (Fig. EV4L), EpoD treatment significantly recued the death of *Wdr47*-deficient neurons (34% and 58% survival in untreated and EpoD treated DIV10 HOM cultures, respectively) (Fig. 7O,P), indicating a major contribution of altered MT dynamics to the rapid degeneration of *Wdr47*-deficient neurons.

Altogether, these results indicate that WDR47 exerts its pro-survival function, at least in part, through the regulation of axonal transport and MT stability.

## Pathogenic WDR47 variants impede neuronal migration and interhemispheric connectivity

Having validated the pathogenicity of the complete loss-of-function p.(Arg193His) variant using depletion of *Wdr47* in mice, and considering the function of WDR47 in regulating neuronal migration and the development of callosal neurons, we hypothesized, that the other variants that do not directly impair WDR47 expression could lead to NAPs through partial loss of function. To test this hypothesis, we first tested the ability of p.(Asp466His), p.(Lys592Arg), p.(Pro650Leu), and p.(His659Pro) variants to restore the migration defects observed upon *Wdr47* deletion. We expressed human wild-type or mutant WDR47 in *Wdr47*-depleted neurons by IUE of NeuroD expression constructs (NeuroD:hWDR47) and NeuroD-Cre-GFP in *Wdr47*^fl/fl embryos at E14.5. While the p.(Pro650Leu) and the p.(His659Pro) variants partially restored the distribution of electroporated neurons at E18.5 (52%, 71%, 61%, 61% GFP+ cells in the uCP for NeuroD-Cre, WT, p.(Pro650Leu) and p.(His659Pro), respectively), expression the p.(Lys592Arg) and the p.(Asp466His) the two variants found in Patient M03, who presents with periventricular nodular heterotopia (Dataset EV1), totally failed to rescue the impaired migration of *Wdr47*-KO neurons (Fig. 8A,B). Of note, when overexpressed under control conditions (Cre in *Wdr47*^fl/+), 3 of the 4 variants induced a slight migration phenotype in the upper cortical plate only (Fig. EV5A,B). Altogether, these results demonstrate that missense WDR47 variants impede, to various extents, the radial migration of projection neurons, with an inverse correlation between the level of rescue and the severity of the migration phenotype observed in patients.

We next tested for restoration, by the various WDR47 variants, of CC anomalies induced by loss of *Wdr47* at P21 once callosal axons achieve their adult-like arborization pattern. We performed IUEs of NeuroD-Cre-GFP and pCAG2-mScarlet plasmids together with either WT or mutant WDR47 constructs in E15.5 *Wdr47*^fl/fl embryos and analyzed the scarlet-positive neurons and their axons. In contrast to the total loss of *Wdr47* that leads to an early onset burst of neuronal death (Fig. 4F), all tested variants rescued the neuronal death phenotype as revealed by the bipolar morphology of electroporated neurons (Fig. 8C) and the absence of pyknotic nuclei and Caspase 3 positive cells. Nonetheless, expression of the p.(Pro650Leu) and p.(His659Pro) variants very slightly rescued

axonal projection as shown by the reduced density of scarlet-positive axons in the midline compared to control (−71%, $p = 0.0003$ and −67%, $p = 0.0011$, respectively) (Fig. 8C,D). Interestingly, the variants found as compound heterozygous in M03 have distinct effect after expression in *Wdr47*-depleted neurons. Indeed, while the expression of the p.(Lys592Arg) variant fully restore the CC thickness, the expression of the p.(Asp466His) variant partially rescued the density of scarlet positive axons in the midline (−45%, $p = 0.0133$) (Fig. 8C,D). Rerouting through alternate commissures was excluded as no aberrant axonal projections were observed and expression of none of the variants in the *Wdr47*^fl/WT animals used as controls caused any CC phenotype (Fig. EV5C,D). Altogether these results indicate that while the variant that leads to a total loss of *WDR47* causes AgCC through an early burst of neuronal death, variants that lead to partial loss of protein function induces thinning of CC through other mechanisms acting possibly on axon generation or extension.

## Discussion

In this study, we demonstrate a causative link between bi-allelic missense variants in human *WDR47* and various neuroanatomical phenotypes (NAPs). In line with previously characterized *Wdr47*-deficient mouse models (Kannan et al, 2017), corpus callosum dysgenesis (CCD) and microcephaly are fully penetrant features observed in all human patients (7/7) (Dataset EV1). Other NAPs are also present but not fully penetrant. Five patients out of 7 showed enlarged ventricles with no other cilia-related pathological symptoms. This partial penetrance of enlarged ventricles has also been observed in *Wdr47*-deficient mouse models (Kannan et al, 2017; Liu et al, 2021). Interestingly, enlarged ventricles in mouse has been correlated with disorders of the corpus callosum (Collins et al, 2019). The absence of ciliopathy phenotypes indicates that the ventricular enlargement in these patients might be a secondary consequence of CCD, rather than a direct consequence of ciliopathy, which contrasts with the previous reports of WDR proteins' involvement in ciliopathies (Accogli et al, 2024; Kim and Kim, 2020). One patient (M03) presented with periventricular heterotopia (PVHT), a neuronal migration disorder reflecting Wdr47's regulatory role in migration (Chen et al, 2020; Kannan et al, 2017). In accordance, the only variants that completely failed to compensate the migration phenotype induced by the loss of *Wdr47* are the two variants found as compound heterozygous in Patient M03 (Fig. 8A,B). On the same line, given the loss of WDR47 protein in M01 and M02 patients, one can speculate that M01 and M02 newborns presented with heterotopia, that could not be detected by brain MRI.

The brain is the most severely affected organ in the patients identified in this study. This is in line with the expression pattern of WDR47 that is particularly enriched in both developing and adult brain compared to other organs in humans and mice (Fig. 1 and EV1). Within the brain, the cortex, which is the most severely affected structure in patients, has the highest level of WDR47 expression. Interestingly, as WDR47 is enriched in both excitatory and inhibitory neurons, it is not surprising that all the patients present with epilepsy (Dataset EV1). Yet, WDR47 is also expressed in the olfactory epithelium and, to a lesser extent, in the thalamus, hypothalamus, septum and hindbrain (Fig. 1M,N), with the latter

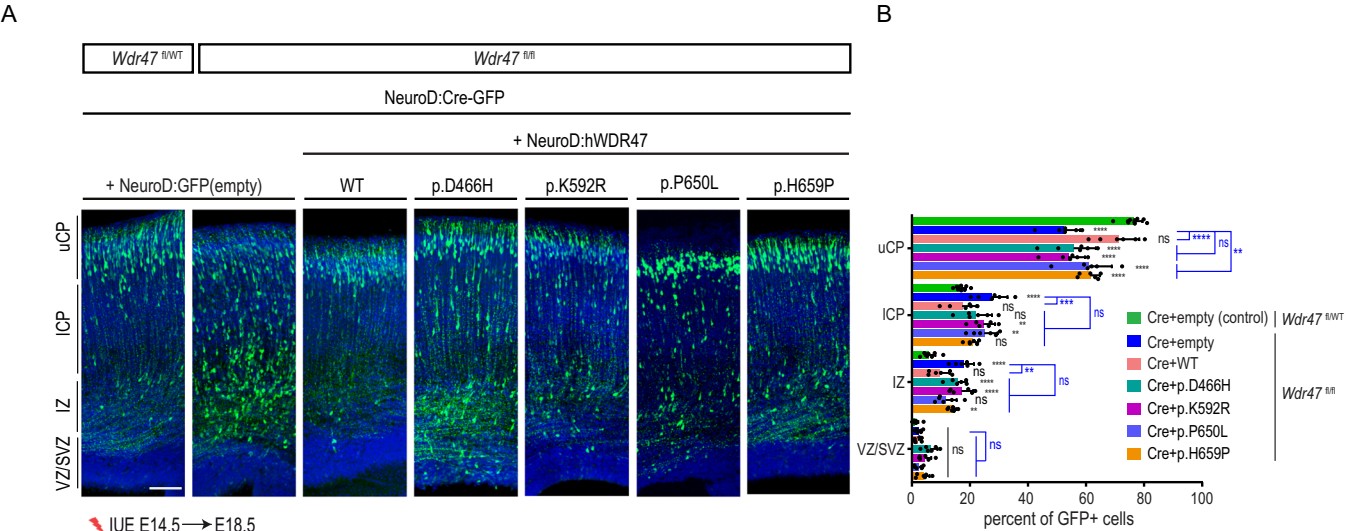

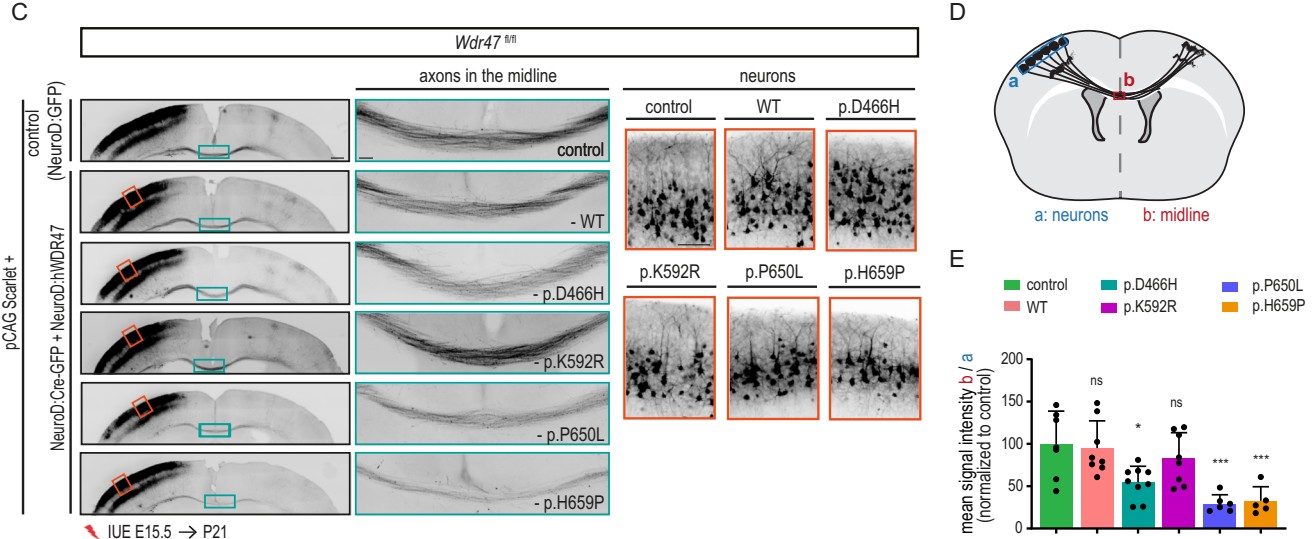

**Figure 8. Different *WDR47* variants impede neuronal migration and interhemispheric connectivity to varying degrees.**

(A) Coronal sections of E18.5 mouse cortices, four days after in utero electroporation with NeuroD-Cre-GFP together with NeuroD:empty vector or FL or mutant NeuroD:hWDR47 constructs. GFP-positive electroporated cells are depicted in green. Nuclei are stained with DAPI. Scale bar: 100 μm. (B) Analysis of the percentage of electroporated GFP-cells in different regions (uCP, lCP, IZ, and VZ/SVZ) showing the ability of the WDR47 constructs carrying one of the different variants found in patients to rescue neuronal migration. Data (means ± s.d.) from at least five embryos from 2 to 4 different litters per condition was analyzed by two-way ANOVA, with Bonferroni's multiple comparisons test. uCP, Upper cortical plate; lCP, Lower cortical plate; IZ, intermediate zone; VZ, ventricular zone; SVZ, subventricular zone. (C) Coronal sections of P21 mouse brains electroporated at E15.5 with pCAG:Scarlet plasmid together with either a control plasmid or a NeuroD:Cre-GFP plasmid and a rescue construct (either a NeuroD: hWDR47-WT or a NeuroD:hWDR47 carrying one of the human mutations). Scarlet positive electroporated neurons are depicted in black. Close-up views of the green boxed area show that different constructs have different effects on rescue of CC phenotypes. Close-up views of the red boxed areas show bipolar morphology of the neurons in the cortical plate. Scale bars: 500 μm, 100 μm (green and red boxed insets). (D) Schematic describing the methods used to quantify (E) the CC thickness (mean intensity of the scarlet signal in the CC (red box) normalized on the mean intensity in the cortical plate (blue box)). (E) CC thickness upon introduction of different WDR47 mutated constructs. Data (means ± s.d.) from at least 5 pups from 2 to 3 different litters per condition were analyzed by one-way ANOVA, with Bonferroni's multiple comparison test, ns, non-significant, *P < 0.05; **P < 0.01; ***P < 0.001; ****P < 0.0001. Exact P values are listed in Dataset EV4. Source data are available online for this figure.

being affected in five patients. Finally, four of the 5 surviving patients present with feeding difficulties, a comorbidity often associated to neurodevelopmental disorders. Given the very low expression of WDR47 in the organs of the digestive system, it is likely that these feeding symptoms arise from indirect mechanisms.

We previously showed that *Wdr47* deficiency in constitutive knock-out models leads to reduced number of progenitors and increased neuronal death, both of which likely contributing to the microcephaly phenotype (Kannan et al, 2017). In this study, we focused on the cellular and molecular origins of CCD. The

development of the CC requires the interplay of different cellular populations, including midline glial structures, meninges, pioneering axons, cortical progenitors and callosal projection neurons (CPNs) (Edwards et al, 2014). Through specific inactivation of *Wdr47* in neurons, we identified a major contribution of postmitotic neurons in the CC phenotype (Figs. 3 and 4; Table 1). This cell autonomous function in cortical projection neurons is in line with the enriched expression of *WDR47* in neurons, including callosal neurons, compared to glia in the cortex throughout life (Fig. 1O). Notably, acute deletion of *Wdr47* in CPNs led to defects in axon extension at P2 and a sudden, massive induction of cell death at P3, with nearly no neurons remaining and the CC being totally lost by P8. This unexpected burst of neuronal death contrasts with previous studies, which typically attribute human CCD syndromes to anomalies in neuronal specification and migration, axon extension and guidance (Edwards et al, 2014).

Our study identified five bi-allelic pathogenic variants in *WDR47*. The nearly complete loss of WDR47 protein in M01 Patient (Fig. 2C–E) suggests that the p.(Arg193His) variant completely abolishes the expression of the protein, consistent with a complete loss-of-function (LOF) mechanism. While the p.(Pro650Leu) variant partially reduces WDR47 expression (Fig. 2C), the other variants tested ((p.(Asp466His), p.(Lys592Arg), p.(His659Pro)) do not affect protein abundance (Fig. 2C,F). However, their recessive mode of inheritance and the absence of phenotypes in heterozygous parents (Fig. 1), support a potential loss-of-function mechanism. In line with partial LOF, none of these variants were able to fully restore the NAPs observed in mice upon *WDR47* KO (Fig. 8). The five missense variants are distributed throughout the WDR47 protein. Notably, the Arginine residue at position 193, mutated in Patients M01 and M02, is located within the WDR47 N-terminal domain (WDR47-NTD), specifically in the cross over region (COR), a subdomain critical for forming the WDR47-NTD intertwined dimer, which serves as an interaction interface with protein partners (Ren et al, 2022). Using the crystal structure of the mouse WDR47-NTD (Ren et al, 2022), we showed that the Arginine 193 residue forms a salt bridge with Glutamate 205 located in the COR of the opposing protomer (Fig. 2B). Substitution of the Arginine 193 with a Histidine disrupts this salt bridge, likely destabilizing the dimer. Supporting this hypothesis, we showed a dramatic reduction in WDR47 protein expression in the p.(Arg193His) variant (Fig. 2C–E). Patients M01 and M02 presented with the most severe phenotype, including neonatal death within 24–48 h after birth and agenesis of the corpus callosum. Given the NTD-mediated dimer formation, it is conceivable that mutations in this domain that interfere with dimerization, are highly deleterious for WDR47 expression and function, potentially leading to lethality in utero or shortly after birth. Although, this is supported by: (1) the lethality of the constitutive *Wdr47* KO mice (Chen et al, 2020; Kannan et al, 2017) and (2) the intolerance to complete loss of function in humans, indicated by the absence of homozygous *WDR47* frameshift variants in the gnomAD (Genome Aggregation Database, v4.0.0) collection. Nevertheless, identification of other cases with neonatal death would be required to sustain a relationship between complete *WDR47* LOF and neonatal lethality in humans.

Interestingly, patients with variants outside the NTD presented with milder CC phenotypes. Using acute depletion of *Wdr47* in neurons and complementation assays, we found that

p.(Pro650Leu), p.(His659Pro), and p.(Asp466His) variants cause CC abnormalities not by affecting neuronal survival but possibly by impairing axon generation and growth (Fig. 8C–E). Additionally, our results demonstrate that loss of one copy of *Wdr47* does not lead to any neuronal survival or axon extension defects (Figs. 4A–F and EV2A,B). Together, these results support a model where a minimal level of WDR47 activity is required for proper corpus callosum development in humans and mice. In this model, maintaining 50% of WDR47 function is sufficient to promote CC development. Below this threshold, the severity and breadth of phenotypes increase with the degree of functional loss. Consistently, total loss of function (p.(Arg193His)) leads to complete loss of the CC, while variants suggesting partial loss of function (p.(Pro650Leu), p.(His659Pro), and p.(Asp466His), seen as either a decrease in overall activities or a loss of specific functions, lead to CC thinning, likely through defects in axon extension.

At the cellular level, we demonstrated that the pro-survival function of WDR47 is independent of its roles in neuronal migration and axon extension. While the introduction of Camsap3 successfully restored the migration and axon extension deficits as previously suggested (Chen et al, 2020), it failed to rescue the neuronal survival defects (Fig. 5). These results indicate that: (1) an early burst of neuronal apoptosis is the main driver of the CC phenotype upon *Wdr47* loss, and (2) WDR47 mediates neuronal survival through different downstream effectors other than Camsap3. Given the presence of WDR47 in subcellular fractions enriched in synaptic vesicles (Even et al, 2019) and the perturbations of several genes related to synapse function/organization in *Wdr47*-deficient neurons (Fig. 7B), a synaptic function of WDR47 is plausible. However, as *Wdr47*-deficient neurons die at P3, before synaptic maturation is fully established, it is unlikely that synaptic dysfunction triggers the neuronal loss in vivo.

Our study provides significant insights into the mechanisms underlying the rapid degeneration of *Wdr47*-deficient neurons, highlighting a role for WDR47 in regulating microtubule (MT) homeostasis and intracellular organelles trafficking. First, *Wdr47*-deficient neurons had remarkably reduced acetylated tubulin, indicative of an unstable MT network (Reed et al, 2006) (Fig. 7N–P). WDR47 may stabilize microtubules through its binding to two known regulators of MT dynamics: (i) the superior cervical ganglion-10 (SCG10) protein, a well-established microtubule-destabilizing protein (Kannan et al, 2017; Riederer et al, 1997); and/or (ii) the MT-associated protein 8 (MAP8), which stabilizes MTs and helps localize WDR47 to them (Ding et al, 2006; Orbán-Németh et al, 2005; Wang et al, 2012). Although Camsap3, known to interact with Wdr47 and stabilize MT minus ends (Buijs et al, 2021; Chen et al, 2020), could contribute to Wdr47's MT regulatory function, it does not seem to be involved in the pro-survival role of WDR47, as it failed to rescue callosal neurons survival in vivo (Fig. 5D). Second, we observed impaired MT-based transport of lysosomes and mitochondria in *Wdr47*-deficient neurons (Figs. 7J–M and EV4D–G). Although these transport defects may be partly due to decreased acetylation and MT instability (Dompierre et al, 2007; Reed et al, 2006) (Fig. 7N–P), unpublished co-immunoprecipitation assays suggest that WDR47 may also interact directly with motor proteins like dynein and kinesin, influencing cargo transport. Accordingly, WDR47 has been found on the surface of motile vesicles isolated from mouse cerebral cortex (Hinckelmann et al, 2016). Moreover, recent studies showed

that WDR47 negatively controls the formation of motile molecular motor complexes, thereby regulating MT-dependent transport of ATG9A-containing vesicles (Guardia et al, 2021) and the intraflagellar transport of cargos along the ciliary axoneme (Song et al, 2022). Finally, treatment with EpoD, a MT-stabilizing compound (Brunden et al, 2010) known to enhance intracellular transport (Zhang et al, 2012), significantly improved the survival of *Wdr47*-deficient neurons (Fig. 7O,P). Although a full rescue with EpoD was not achieved, possibly due to dose or time-dependent factors, this suggests that additional mechanisms besides decreased MT stability contribute to the rapid degeneration of *Wdr47*-deficient neurons.

Our study also unveils a previously unidentified function of WDR47 in maintaining mitochondrial homeostasis. We found that loss of *Wdr47* in neurons leads to mitochondrial fragmentation and increased production of reactive oxygen species (ROS) (Fig. 7E–I). Although we cannot totally rule out the possibility that mitochondrial defects are a consequence of dying neurons, several lines of evidence suggest that it is one of the mechanisms that ultimately trigger neuronal death. First, defects in mitochondria homeostasis likely precedes onset of neuronal death as attested by: (1) time lapse recording of mitochondrial defects in neurons devoid of hallmarks of cell death (Figs. 7E–G) and (2) impaired mitochondrial mass and membrane potential in WDR47 KO HeLa cells that do not show any defects in survival (Fig. EV4B,C). Second, *Wdr47*-deficient neurons exhibited hyperpolarization of mitochondrial membrane (Fig. 7H), a condition known to cause oxidative stress and neuronal death, as seen in neurons derived from patients with Tau mutations (Esteras et al, 2017). Indeed, the increased ROS levels and the decreased expression of the critical antioxidant enzyme *Sod1* in *Wdr47*-deficient neurons suggest a disruption in redox homeostasis (Fig. 7D,I). As neurons are highly vulnerable to oxidative stress owing to their non-dividing nature and high metabolic demands (Bonda et al, 2014), this imbalance may contribute to their degeneration. Third, given WDR47's presence in mitochondria isolated from neurons (Fecher et al, 2019) and the recent analyses of the WDR47 interactome that have revealed its binding to mitochondrial proteins involved in the ATP synthase complex, pyruvate metabolism and the transport of metabolites and ions (Buijs et al, 2021; Chen et al, 2020), it is tempting to speculate that WDR47 has a direct function on mitochondrial homeostasis, in addition to its role in MT-based transport.

The molecular alterations we observed in *Wdr47*-deficient developing neurons resemble pathological mechanisms that underlie neuronal loss in neurodegenerative diseases (Brady and Morfini, 2017; Dubey et al, 2015; Flippo and Strack, 2017; Mattson et al, 2008; Sferra et al, 2020). We identified disruptions in a significant number of Alzheimer's and Parkinson's-related genes in *Wdr47*-deficient primary neuronal cultures, corroborating a recent study that recognized *WDR47* as a hub gene in Alzheimer's (Zhang et al, 2020). In addition, loss of MT homeostasis, impaired cargo transport and mitochondria alterations are hallmarks of a variety of common neurodegenerative diseases, such as Alzheimer's disease (AD), amyotrophic lateral sclerosis (ALS), Parkinson's disease (PD) and Huntington's disease (HD). Interestingly, WDR47 shares similarities with Huntingtin (Htt) and Tau, the proteins whose dysfunctions cause HD and AD/frontotemporal dementia (FTD), respectively. Alike Htt and Tau (Cheng and Bai, 2018; David et al, 2005; Ganapathy et al, 2009; Ismailoglu et al, 2014; Jimenez-

Sanchez et al, 2017; Swaih et al, 2022; Yablonska et al, 2019), WDR47 plays a role in mitochondrial homeostasis and interacts with mitochondrial proteins (Buijs et al, 2021; Chen et al, 2020; Fecher et al, 2019), in addition to its initially recognized MT-associated function. WDR47 has also been identified as a binding partner of Huntingtin (Culver et al, 2012). Given the growing evidence that neurodegeneration in HD stems from alterations occurring during development (Barnat et al, 2020; Braz et al, 2022; Capizzi et al, 2022), it is tempting to speculate that WDR47 safeguards callosal neurons from the early activation of neurodegenerative mechanisms.

Future studies should investigate whether WDR47 variants in different domains interfere with specific functions to varying extends. The MT-stabilizing function could rely on both the N- and C-terminal domains, as the WD40 domain is involved in WDR47's binding to MTs via MAP8 (Wang et al, 2012), while the N-terminal domain is critical for binding to SCG10 (Kannan et al, 2017). Given the strong homology to LIS1, that has an N-terminal dimerization domain and a WD40 C-terminal domain required for its binding to molecular motors (Kim et al, 2004; Sasaki et al, 2000), WDR47's C-terminal domain may play a key role in regulating intracellular transport. The precise mapping of WDR47's functions using sensitive methods is required to unveil the possible pleiotropic roles of its different domains and to better predict the pathogenicity of variants based on their position within the WDR47 protein sequence.

In summary, our data identify *WDR47* as an important gene for neuroanatomical disorders, including corpus callosum dysgenesis (CCD) and microcephaly, in both humans and mice. We show that the degeneration of *Wdr47*-deficient neurons is the main driver of CC agenesis upon *Wdr47* loss. WDR47 safeguards callosal neurons from early-onset neuronal death by contributing to the maintenance of mitochondrial and microtubule homeostasis. We suggest that *WDR47* variants should be considered in unexplained cases of corpus callosum abnormalities, microcephaly, intellectual disability, and epilepsy.

# Methods

## Reagents and tools table

| Reagent/Resource | Reference or Source | Identifier or Catalog Number |
|---|---|---|
| **Experimental models** | | |
| *Wdr47^tma(EUCOMM)Wtsi* (M. musculus) | Welcome Sanger Institute (Cambridge, UK) | N/A |
| *Camsap3^tma(EUCOMM)Wtsi* (M. musculus) | Welcome Sanger Institute (Cambridge, UK) | N/A |
| 129S2/SvPasOrlRj (M. musculus) | Janvier Laboratories | 129S2/SvPasOrlRj |
| C57BL/6NCRL (M. musculus) | Charles River Laboratories | C57BL/6NCRL |
| Nex^Cre (M. musculus) | Provided by Laurent Nguyen (Goebbels et al, 2006) | |
| CaMKIIα^Cre (M. musculus) | Provided by Yann Herault (Mantamadiotis et al, 2002) | |
| Human embryonic kidney (HEK) 293T | ATCC | |

| Reagent/Resource | Reference or Source | Identifier or Catalog Number |
|---|---|---|
| WT HeLa cells | Provided by Juan S, Bonifacino | |
| *WDR47*- KO Hela cells | Provided by Juan S, Bonifacino (Guardia et al, 2021) | |
| Patient fibroblasts | This study | |
| Patient lymphocytes | This study | |
| Mouse neuroblastoma N2A | ATCC | |
| wild-type BY4742 reference strain (MATα leu2Δ0 ura3Δ0 his3Δ0 lys2Δ0) (*S. cerevisiae*) | Euroscarf | Y10000 |
| oxa1Δ mutant strain (MATα leu2Δ0 ura3Δ0 his3Δ0 lys2Δ0 oxa1::KanMX) (*S. cerevisiae*) | Euroscarf | Y16151 |
| **Recombinant DNA** | | |
| human *WDR47* cDNA | DNASU | plasmid ID HsCD00434289 |
| NeuroD-IresGFP | Provided by Laurent Nguyen (Hand and Polleux, 2011) | |
| pcDNA3.1+/C-HA | Provided by Gulayse Ince-Dunn | |
| NeuroD: hWDR47-WT-IresGFP | This study | |
| pcDNA3.1 hWDR47-WT-HA | This study | |
| pcDNA3.1 hWDR47-R193H-HA | This study | |
| NeuroD: hWDR47-D466H-IresGFP | This study | |
| pcDNA3.1 hWDR47-D466H-HA | This study | |
| NeuroD: hWDR47-K592R-IresGFP | This study | |
| pcDNA3.1 hWDR47-K592R-HA | This study | |
| NeuroD: hWDR47-P650L-IresGFP | This study | |
| pcDNA3.1 hWDR47-P650L-HA | This study | |
| NeuroD: hWDR47-H659P-IresGFP | This study | |
| pcDNA3.1 hWDR47-H659P-HA | This study | |
| NeuroD: hWDR47delLISH-IresGFP | This study | |
| pcDNA3.1 hWDR47delLISH-HA | This study | |
| NeuroD: hWDR47delCTLH-IresGFP | This study | |
| pcDNA3.1 hWDR47delCTLH-HA | This study | |

| Reagent/Resource | Reference or Source | Identifier or Catalog Number |
|---|---|---|
| NeuroD: hWDR47delLISH-CTLH-IresGFP | This study | |
| pcDNA3.1 hWDR47delLISH-CTLH-HA | This study | |
| NeuroD: hWDR47delWD40-IresGFP | This study | |
| pcDNA3.1 hWDR47delWD40-HA | This study | |
| pFASBac+GFP-Camsap3 | Addgene | plasmid# 59038 |
| NeuroD: Camsap3-IresGFP | This study | |
| Neuro-D:Cre-GFP | Gift from Laurent Nguyen | |
| pCAG2-mScarlet | Gift from Julien Courchet | |
| pSCV2-Venus | Gift from Julien Courchet (Hand and Polleux, 2011) | |
| pCAG-mitoDsRed | Gift from Julien Courchet (Courchet et al, 2013) | |
| LAMP1-YFP | Addgene | plasmid#1816 |
| pCAGmito-Grx1-roGFP2 | Gift from Julien Courchet (Gutscher et al, 2008) | |
| promCUP1-mCherry-V5-ATG8 | Gift from Fulvio Reggiori | |
| pAG413GAL-ccdB | Addgene (Alberti et al, 2007) | plasmid# 14141 |
| pAG413-promGPD-hWDR47-WT | This study | |
| pAG413-promGPD-WDR47-ΔLISH | This study | |
| pAG413-promGPD-WDR47-ΔCTLH | This study | |
| **Antibodies** | | |
| Rabbit Anti-activated Caspase-3 (1:100/IHC) | R and D System | AF835 |
| Chicken Anti-GFP (1:800/IHC) | Aves lab | GFP-1020 |
| Mouse Anti-LAMP1 (1:1000/IHC) | Developmental Studies – Hybridoma Bank | H4A3 |
| Rabbit Anti-WDR47 (1:200/IHC on human sections – 1:2000/IHC on mouse sections) | Atlas antibodies | HPA027289 |
| Mouse Anti-GAPDH (1:5000/WB) | Sigma-Aldrich | MAB-374 |
| Goat Anti-GFP (1:1000/WB) | Abcam | ab6673 |
| Mouse Anti-α-tubulin (1:2000/WB) | Sigma-Aldrich | T9026 |
| Mouse Anti-acetylated α-tubulin (1:500/WB) | Thermo Fisher | 322700 |
| Rat Anti-tyrosinated α-tubulin (1:2000/WB) | Sigma-Aldrich | MAB1864-I |

| Reagent/Resource | Reference or Source | Identifier or Catalog Number |
|---|---|---|
| Mouse Anti-β actin coupled HRP (1:40,000/WB) | Sigma-Aldrich | A3854 |
| Rat Anti-HA (1:2000/WB) | Roche Life Science products | 11867423001 |
| Rabbit Anti-WDR47 (1:1000/WB) | Abcam | ab121935 |
| Donkey Anti-chicken-488 (1:1000/IHC) | Abcam | ab63507 |
| Donkey Anti-rabbit-647 (1:1000/IHC) | ThermoFisher Sc. | A-31573 |
| Donkey Anti-mouse-555 (1:1000/ICC) | ThermoFisher Sc. | A-31570 |
| Goat Anti-mouse-HRP (1:10,000/WB) | ThermoFisher Sc. | G-21040 |
| Goat Anti-rabbit-HRP (1:10,000/WB) | ThermoFisher Sc. | G-21234 |
| Goat Anti-rat-HRP (1:10,000/WB) | ThermoFisher Sc. | 62-9520 |
| Donkey Anti-goat-HRP (1:5000/WB) | Santa Cruz | Sc2020 |
| **Oligonucleotides and other sequence-based reagents** | | |
| qPCR primers | This study | Appendix Table S2 |
| Genotyping primers | This study | Appendix Table S3 |
| **Chemicals, Enzymes and other reagents** | | |
| Direct PCR Lysis Reagent | Viagen | 101-T |
| Proteinase K | Sigma-Aldrich | P6556 |
| Ketamine | Alcyon France | 6740407 |
| Xylazine (Domitor) | Alcyon France | 8131902 |
| Solvent Blue 38 | Sigma-Aldrich | S3382 |
| Cresyl violet acetate | Sigma-Aldrich | C5042 |
| Bouin solution | ThermoFisher Sc. | 11994074 |
| Paraffin | VWR | 10048502 |
| Histosol | ThermoFisher Sc. | NC1706825 |
| FastGreen FCF | Sigma-Aldrich | F-7258 |
| Fluoromount-G with DAPI | Invitrogen | 00-4959-52 |
| 16% paraformaldehyde | Electron Microscopy Sciences | 15710 -S |
| Low-melting agarose | Bio-Rad | 1613111 |
| Tissue-Tek O.C.T. Compound | Sakura | 4583 |
| Aquapolymount | Polysciences Inc. | 18606-20 |
| DAPI | Sigma-Aldrich | D9542 |
| Papain | Worthington | 3178 |
| DNAse I | Life Technologies | EN0521 |
| Ovomucoïde | Worthington | 3182 |
| Poly-D-lysine | Sigma | P6407 |
| NeuroMag | OZ Bioscience | NM50200 |

| Reagent/Resource | Reference or Source | Identifier or Catalog Number |
|---|---|---|
| QVd-Oph | Sigma-Aldrich | SML0063 |
| Necrostatin | Sigma-Aldrich | N9037 |
| Ferrostatin | Sigma-Aldrich | SML0583 |
| Epothilone D | Abcam | ab143616 |
| Trizol reagent | Invitrogen | 15596026 |
| SuperScript IV Synthesis kit | Invitrogen | 15307696 |
| Light Cycler 480 SYBR Green Master | Roche | 04887352001 |
| Lipofectamine 2000 | Invitrogen | 11668027 |
| MG132 | Sigma-Aldrich | M7449 |
| cOmplete EDTA-free protease inhibitor coctail | Roche | 11873580001 |
| 4X Laemmli sample buffer | Bio-Rad | 1610747 |
| SuperSignal West PicoPLUS Chemilimunescent | ThermoFischerScientific | 34580 |
| SuperSignal™ West Femto Maximum Sensitivity Substrate | ThermoFischerScientific | 34094 |
| Lysotracker Red DND-99 | ThermoFischerScientific | L7528 |
| Tetramethylrhodamine, ester ethylique, perchlorate (TMRE) | Sigma-Aldrich | 87917 |
| MitoTracker Green FM | ThermoFischerScientific | M46750 |
| Nocodazole | Sigma-Aldrich | SML1665 |
| MgSO4 | Sigma-Aldrich | M7506 |
| D-Glucose | Sigma-Aldrich | G7021 |
| yeast nitrogen base w/o ammonium sulfate | MP Biomedicals | 4027012 |
| yeast nitrogen base with ammonium sulfate | MP Biomedicals | 4027512 |
| CSM-His powder | MP Biomedicals | 4510312 |
| **Software** | | |
| The Single Nucleotide Polymorphism Database (dbSNP) | https://ncbi.nlm.nih.gov/snp/ | |
| 1000 genomes | https://www.internationalgenome.org/ | |
| Genome Aggregation Database (v4.0.0, v2.1.1 and v3.1.2) | https://gnomad.broadinstitute.org/ | |
| CADD | https://cadd.gs.washington.edu/ | |
| Mutation Tester | https://www.mutationtaster.org/ | |
| Polyphen2 | http://genetics.bwh.harvard.edu/pph2/ | |
| SIFT | https://sift.bii.a-star.edu.sg/ | |

| Reagent/Resource | Reference or Source | Identifier or Catalog Number |
|---|---|---|
| AlphaFold2 | https://github.com/google-deepmind/alphafold (Jumper et al, 2021) | |
| CZ CELLxGENE | https://cellxgene.cziscience.com/ (Abdulla et al, 2023) | |
| evodevoapp | https://apps.kaessmannlab.org/evodevoapp/ (Cardoso-Moreira et al, 2019) | |
| cutadapt v1.10 | https://cutadapt.readthedocs.io/en/v1.10/installation.html (Martin, 2011) | |
| bowtie v2.2.8 | https://bowtie-bio.sourceforge.net/bowtie2/index.shtml (Langmead and Salzberg, 2012) | |
| STAR 2.5.3a | https://github.com/alexdobin/STAR (Dobin et al, 2013) | |
| htseq-count v0.6.1p1 | https://htseq.readthedocs.io/en/latest/ (Anders et al, 2015) | |
| R v. 3.3.2 | https://cran-archive.r-project.org/bin/windows/base/old/3.3.2/ (Ihaka and Gentleman, 1996) | |
| DESeq2 v1.16.1 | https://bioconductor.org/packages/release/bioc/html/DESeq2.html (Love et al, 2014) | |
| ClusterProfiler v. 4.8.2 (R v. 4.3.1) | https://bioconductor.org/packages/release/bioc/html/clusterProfiler.html (Wu et al, 2021; Yu et al, 2012) | |
| ImageJ | https://imagej.net/software/imagej/ (Schneider et al, 2012) | |
| Fiji | https://imagej.net/software/fiji/ (Schindelin et al, 2012) | |
| Kymo ToolBox v.1.01 | https://github.com/fabricecordelieres/IJ-Plugin_KymoToolBox/releases | |
| FlowJo 10 | https://www.flowjo.com/solutions/flowjo/downloads | |
| **Other** | | |
| Nanozoomer 2.0HT | Hamamatsu | C9600 series |
| Microtome Micom HM 450 | Thermo-Fischer | 387760 |
| Vibrating-blade microtome (VT1000S) | Leica Microsystems | 14 0472 80101 |
| Microinjector FemtoJet 4i | Eppendorf | 5252 000 030 |
| Glass capillaries puller | KOPF modified | 720 |
| Cryostat (Leica CM3050S) | Leica Biosystems | CM3050S |
| ECM-830 BTX square wave electroporator | VWR International | 732-0024 |
| 3-mm platinum tweezers electrodes | Sonidel | CUY650P3 |

| Reagent/Resource | Reference or Source | Identifier or Catalog Number |
|---|---|---|
| 35 mm diameter glass bottom dish | Cellvis | D35-20-1.5h |
| NucleoSpin RNA purification kit | Macherey-Nagel | 740955 |
| Maxiprep-EF DNA purification kit | Macherey-Nagel | 740424 |
| LightCycler 480 System | Roche | |
| TruSeq® Stranded mRNA Library Prep kit | Illumina | 20020595 |
| TruSeq® RNA Single Indexes kits A and B | Illumina | 20020492/20020493 |
| Agilent 2100 Bioanalyzer | Agilent | G2939BA |
| Qubit dsDNA HS Assay Kit | ThermoFischerScientific | Q32854 |
| HiSeq 4000 | Illumina | |
| Membrane Immobilon-P | Merck | IPVH00010 |
| BD LSRFortessa™ Cell Analyzer | BD Biosciences | BD LSRFortessa |
| Bradford protein assay reagent | Bio-Rad | 5000001 |
| Amersham Imager 600 | GE Healthcare Life Sciences | 695204 |

## Methods and protocols

### Exome sequencing

Informed consent was obtained from all subjects and the experiments conformed to the principles set out in the WMA Declaration of Helsinki and the Department of Health and Human Services Belmont Report.

Patients M01 and M02.   Research human subjects were enrolled in an IRB-approved research protocol at King Faisal Specialist Hospital and Research Center (Number: 2080 006) with informed consent. Blood was collected from index and available relatives in EDTA tubes for DNA extraction. All subjects were exome sequenced and the resulting variants were filtered by the candidate autozygome as described previously (Alkuraya, 2013; Anazi et al, 2017).

Patient M03.   DNA was extracted from peripheral blood of the patients and their parents. Patient M03's DNA was captured with the SureSelect Human All Exon V6 Kit (Agilent Technologies, Santa Clara, CA, USA) and sequenced on a NextSeq500 (Illumina, San Diego, CA, USA) with 150-bp paired-end reads. Reads were aligned to reference genome (GRCh38) using BWA-MEM (Version 0.7.17). Duplicated reads were removed by Picard (Version 2.20.3), and local realignment and base quality recalibration were performed by GATK (Version 4.1.9.0). The total aligned bases and mean depth of coverage against RefSeq coding sequences (33.9 Mb) were 2,278,284,739 bp and 67.18. The target RefSeq coding sequences covered by greater than 20 reads were 95%. Variants were identified with the GATK HaplotypeCaller. The final variants were annotated with Annovar (Wang et al, 2010) for predictive value of functional impact of the coding variants and assessing allele frequency using

follow databases: the gnomAD database v3.1.2 (https://gnomad.broadinstitute.org/), the Human genetic variation database (http://www.hgvd.genome.med.kyoto-u.ac.jp/), and the ToMMo 8.3KJPN Allele Frequency Panel (v20200831) (https://jmorp.megabank.tohoku.ac.jp/202102/downloads/legacy/). We focused on rare variants with minor allele frequencies below 1% in the above four databases. The damaging prediction was performed by SIFT, Polyphen-2, MutationTaster, CADD and M-CAP programs. Candidate variants were confirmed by Sanger sequencing using an ABI 3130xl Genetic Analyzer (Applied Biosystems, Foster City, CA). This study was approved by the institutional review board committee of Showa University School of Medicine (G220-N). Consent to publish the patient's picture was obtained.

Patient M04. Genomic DNA was fragmented, and the exons of the known genes in the human genome, as well as the corresponding exon-intron boundaries were enriched using Roche KAPA capture technology (Basel, Switzerland), amplified and sequenced simultaneously by Illumina technology (San Diego, US). The target regions were sequenced with an average coverage of 148.9-fold. For about 99.9% of the regions of interest a 15-fold coverage was obtained. NGS data were aligned to the hg19 genome assembly. Variant calling and annotation were performed by an in-house developed bioinformatics pipeline. Identified SNVs and indels were filtered against external and internal databases focusing on rare variants with a minor allele frequency (MAF) in gnomAD of 1% or less and removing known artifacts and variants in regions with highly homologous regions. The filtering of the exome data targeted X-linked, autosomal dominant and recessive disorders. In parallel, we received parental samples for which exome sequencing was performed in a Trio setting. Classification of variants was conducted based on ACMG guidelines (Richards et al, 2015) considering variant databases including but not limited to ClinVar, bioinformatics prediction tools and literature status. A change of pathogenicity classifications over time cannot be excluded. Variants annotated as common polymorphisms in databases or literature or that were classified as (likely) benign were neglected. Written informed consent to perform clinal and genomic studies for research purpose and to publish the Patient's picture were obtained from patient's legal representatives.

Patients M05 and M06. After extraction, quality control and quantification, DNA was enriched for paired-end sequencing of the entire coding region and the exon/intron boundaries of all the known genes of the human genome (~20,000), for a total of ~33 Mb of human exonic content. Sequencing was performed at an Illumina certified laboratory. The alignment of the reads was done on the GRCh37/hg19 assembly. The mitochondrial DNA sequence was obtained by aligning the off-target reads. Analysis metrics included an average coverage of 135x and target percentages at 10x and 20x were 97.6% and 97.4%, respectively. The percentage of segmental duplications was 8.04%. The data were analyzed with primary, secondary and tertiary bioinformatics analysis, annotating the variants on the main databases, including ClinVar, performing in silico predictions with a validated set of algorithms, and using variant filtering platforms based on the patient's clinical phenotype. GnomAD allele frequencies are from version v.2.1.1 of the database. Some genes or portions of genes may not be analyzable due to the particular conformation of the sequence or the presence of pseudogenes. Variants that are poorly covered (e.g. <10x) or have an unbalanced allele ratio may be filtered and excluded from the analysis. Consent to publish both siblings' picture was obtained. This study was approved by the ethic committee of Centro Zigote Srl, Salerno, Italy.

Patient M07. Exome sequencing was performed at Mendelics Genomic Analysis facilities using Illumina NovaSeq 6000 after obtaining written informed consent to perform clinal and genomic studies for research purpose. Sequencing library was built with Illumina Nextera Flex and for capture of target regions Customized Exome Kit from Twist Biosciences is used. Sequencing of sample results in paired 101 bp sequences that are mapped to hg38 reference using BWA MEM software (http://bio-bwa.sourceforge.net/). 99.8% of the targeted regions had at least 10 reads sequenced, resulting in a total 43,754,430 generated and with a medium vertical coverage of 79-fold. Resulting BAM files were genotyped using Broad Institute best practices with GATK (https://software.broadinstitute.org/gatk/). Resulting VCF files were processed using Mendelics in-house pipeline for annotation and filtering. In silico pathogenicity evaluation was performed using VSA, a Mendelics proprietary machine-learning-based software. Aligned BAM files were also processed by Exome-Depth (an R package, see https://cran.r-project.org/web/packages/ExomeDepth/index.html) in order to identify CNVs. Generated data was analyzed by Mendelics medical team and guided by clinical information provided by the primary physician. For analysis, several genomic tools and variant/phenotype databases were applied, including University of Santa Cruz Genomic Browser (https://genome.ucsc.edu/index.html), gnomAD browser v2.1.1 and v3.1.2 (https://gnomad.broadinstitute.org/), Human Gene Mutation Database – HGMD (http://www.hgmd.cf.ac.uk/ac/index.php), ClinVar (https://www.ncbi.nlm.nih.gov/clinvar/), Mastermind by Genomenom (https://mastermind.genomenon.com/); Leiden Open Variation Database – LOVD (https://www.lovd.nl/), OMIM – Online Mendelian Inheritance in Man (https://www.omim.org/), as well as medical literature by searching Pubmed (https://pubmed.ncbi.nlm.nih.gov/). Multiple parameters were considered during variant analysis—population frequency, gene associated phenotypes and inheritance pattern, presence of previous reports in medical literature and variant databases, functional data, segregation analysis and in silico predictions. Potentially clinically significant variants were visualized using Integrative Genomics Viewer - IGV Browser (https://software.broadinstitute.org/software/igv/).

Whenever parental consanguinity was present and no clinically relevant variants were found in genes with a presently known phenotype, search for rare homozygous variants in genes without an associated phenotype was performed, using MAF < = 1% as a filtering parameter. Genes in which potentially relevant variants were identified, were evaluated for expression pattern, biological function (when known), functional data and animal models that might be reminiscent of the patient's phenotype, when available.

Based on these parameters, the *WDR47* missense variant was considered a possible candidate and thus was validated in the proband and confirmed in the heterozygous state in both of his parents by Sanger sequencing.

### Analysis of human and mouse WDR47's expression pattern

To explore the expression pattern of *WDR47* across organs, developmental stages and cell types, in Figs. 1K and EV1A, RPKM

values for human *WDR47* (ENSG00000085433) and mouse *Wdr47* (ENSMUSG00000040389) were obtained from (Cardoso-Moreira et al, 2019) (https://apps.kaessmannlab.org/evodevoapp/). Ages "youngAdult", "youngMidAge", and "olderMidAge" were reassigned to "adult" and a subset of the most relevant ages was selected for plotting with ggplot2. In Figs. 1L,O and EV1B, CZ CELLxGENE (https://cellxgene.cziscience.com/) datasets were used (Abdulla et al, 2023). Mouse and human datasets with Wdr47 expression were accessed as Seurat Object using the *get_seurat* function. Data was then log-normalized and scaled using Seurat's *NormalizeData* (factor = 10000) and *ScaleData* functions. Age definition was standardized across datasets with the following time brackets; human: "first trimester", "early second trimester", "late second trimester", "third trimester", "newborn", "infant", "child", "adolescent", "adult", "senior"; mouse: "E11.5","E12", "E13", "E14", "E16", "E18", "P0", "adult", "senior". For each dataset, the percentage of cells expressing *WDR47* and their average expression values were extracted and scaled via the function *DotPlot* function, grouped by their age and organs. For the organ-level plots (Figs. 1L and EV1B), values were obtained for all cells combined. For each organ and age, we calculated the mean expression across datasets, weighted by their total number of cells as well as their weighted mean percentage of cell expressing *WDR47*. For the cortex-level plot (Fig. 1O), values were calculated for each individual cell type. Scaled expression values obtained from Seurat from all available datasets (human *n* = 201, mouse *n* = 7) were used. To ensure consistency, datasets with the following criteria were included: (a) cortex-specific data was available, (b) cellular resolution at the neuronal subtype-level was included, (c) age information was specified for the samples. Only clusters with at least 50 cells and of brain-specific identity were further retained. To homogenize across datasets, clusters were renamed to match overlapping cell identities. Expression values and cell percentages for each of these new cell types and ages were obtained by averaging (weighted mean over their number of cells) the scaled expression. Both developmental and adult stages were selected for visualization.

### Ethics human fetuses

All samples were provided by INSERM's HuDeCA Biobank and used in compliance with French regulations. Authorization to use human tissues was granted by the French agency for biomedical research (Agence de la Biomédecine, Saint-Denis La Plaine, France; N° PFS19-012) and the INSERM Ethics Committee (IRB00003888). Non-pathological human fetuses were obtained from pregnancies terminated voluntarily after written informed consent of the parents (Gynecology Department, Jeanne de Flandre Hospital, Lille, France).

### Immunolabeling on slices of fixed-tissues for human fetuses

Fetal tissues were made available in accordance with French bylaws (Good Practice Concerning the Conservation, Transformation, and Transportation of Human Tissue to Be Used Therapeutically, published on December 29, 1998). Gestational week 14 (GW14) intact foetuses were fixed by immersion in 4% paraformaldehyde (PFA) at 4 °C for 5 days. The tissues were cryoprotected in 30% sucrose/PBS at 4 °C for two nights, embedded in Tissue-Tek OCT compound (Sakura Finetek, USA), frozen on dry ice and stored at −80 °C until sectioning. Frozen samples were cut serially at 16 μm using a Leica CM 3050S cryostat (Leica Biosystems Nussloch

GmbH, Germany). For immunolabeling, antigen retrieval was performed by incubating the slices in boiling citrate buffer (0.1 M) for 15–20 min. Slices were then washed with PBS and incubated with primary anti-WDR47 antibody (Appendix Table S1) diluted 1:200 in PBST (PBS + 0.3% Triton) + 1% NDS (Normal Donkey Serum) for 2 nights at 4 °C. After washing with PBS, sections were incubated with secondary antibody (Appendix Table S1) in PBST + 2% NDS for 1 h at room temperature (RT). Finally, sections were coverslipped with Fluoromount-G with DAPI (Invitrogen 00-4959-52), as an antifade mounting medium.

### Mice

All animal studies were conducted in accordance with French regulations (EU Directive 86/609 – French Act Rural Code R 214-87 to 126) and all procedures were approved by the local ethics committee and the Research Ministry (APAFIS#15691-201806271458609 for in utero electroporation, APAFIS #15169 – 2018052111497498 for intracardiac final perfusion, APAFIS #2016010717527861 for neuroanatomical analysis). Mice were bred at the animal facility of the Mouse Clinical Institute (MCI) (IGBMC) or of the INSERM unit 1231 under controlled light/dark cycles and were provided with food (chow diet) and water ad libitum, with cage enrichment.

*Wdr47^tma(EUCOMM)Wtsi* and *Camsap3^tma(EUCOMM)Wtsi* knock-out mice generated by homologous recombination in embryonic stem cells using the knock-out-first allele method (Skarnes et al, 2011) (Appendix Fig. S2A) were obtained through collaboration with the Mouse Genetics Project (MGP) from the Welcome Sanger Institute (Cambridge, UK), a partner of the International Mouse Phenotyping Consortium (IMPC).

To obtain timed-pregnant females: hybrid F1 females *Wdr47*^tm1c/wt (fl/wt) were obtained by mating inbred 129S2/SvPasOrlRj males (Janvier-labs) with *Wdr47*^tm1c/tm1c(fl/fl) females (C57BL/6NCRL background Charles River Laboratories). F1 females were then crossed with *Wdr47*^tm1c/tm1c(fl/fl) males for in utero electroporation with dating of pregnancy by vaginal plug.

Nex^Cre mice (Goebbels et al, 2006) were obtained from Charles River and Nex^Cre mice were crossed with *Wdr47*^fl/fl mice to generate double mutant mice Nex^Cre/wt;*Wdr47*^fl/wt and those double mutant mice were crossed to generate mice with a deletion of *Wdr47* in post-mitotic neurons (Nex^Cre/wt;*Wdr47*^fl/fl and Nex^Cre/Cre;*Wdr47*^fl/fl and their controls (Nex^Cre/wt;*Wdr47*^WT/WT and Nex^Cre/Cre;*Wdr47*^WT/WT).

CaMKIIα^Cre mice (Mantamadiotis et al, 2002) were obtained from the Jacksons Laboratory and crossed with *Wdr47*^fl/fl mice to generate double mutant mice CaMKIIα^Cre; *Wdr47*^fl/WT. Those double mutant mice were then crossed to generate mice with deletion of *Wdr47* in peri-natally born cortical neurons CaMKIIα^Cre; *Wdr47*^fl/fl and their controls (CaMKIIα^Cre;*Wdr47*^WT/WT).

### Genotyping and PCR

Genotyping was done as follows: Genomic DNA was extracted from tail biopsies using PCR reagent (Viagen) supplemented with Proteinase K (1 mg/mL), heated at 55 °C for 2–5 h. Proteinase K was inactivated for 45 min at 85 °C, and cell debris was removed by centrifugation. PCRs for each line were carried as follows and presence of expected products were checked on 1% agarose gel.

PCR for each line was done as follows:

*Wdr47*^tm1a/WT: *Wdr47* forward: 5′-TCCTTTGCTAACTTCCACT ATCC-3′ and *Wdr47* reverse: 5′-TCAGCCTGGTCTACAGAGTT A-3′ for *Wdr47* targeted exon amplification. The presence of the

wild type and knock-out alleles was indicated by 615 bp and 533 bp products, respectively.

*Wdr47*tm1b/WT: *Wdr47* forward: 5′-AGGTTGTCATGCAGTCTG GG-3′ and *Wdr47* reverse: 5′-GGATGACTATAAAGCGGTGCA AG-3′ for *Wdr47* targeted exon amplification; 5′-CTCCTACATAG TTGGCAGTGTTTGGG-3′ for knock-out allele. The presence of the wild type and knock-out alleles was indicated by 642 bp and 362 bp products, respectively.

*Wdr47*tm1c/WT: *Wdr47* forward: 5′-AGGTTGTCATGCAGTCTG GG-3′, *Wdr47* reverse: 5′-GGATGACTATAAAGCGGTGCAAG-3′ for *Wdr47* targeted exon amplification. The presence of the wild type and knock-out alleles was indicated by 642 bp and 848 bp products, respectively.

*Camsap3* tm1a/WT: PCR genotyping was carried out using genomic DNA isolated from mouse tail samples by identifying the inserted cassettes, as described in (White et al, 2013).

NexCre: Nex Cre forward: 5′-GAGTCCTGGAATCAGTCTTTTT C-3′, Nex cre WT reverse: 5′-AGAATGTGGAGTAGGGTGAC-3′, NexCre KI reverse: 5′-CCGCATACCAGTGAAACAG-3′, for Nex targeted exon amplification. The presence of the wild type and knock-in alleles was indicated by 770 bp and 525 bp products, respectively.

CaMKIIαCre: Cre forward 5′-ATCCGAAAAGAAAACGTTG A-3′, Cre reverse 5′-ATCCAGGTTACGGATATAGT-3′. Presence of the cre transgene was detected with a 500 bp product.

### Neuroanatomical characterization of adult murine brains

To obtain the brain samples derived from *Wdr47*tma(EUCOMM)Wtsi and *Camsap3*tma(EUCOMM)Wtsi knock-out mice, adult mice (16 weeks old) were anesthetized using either Ketamine (100 mg/kg, intraperitoneally) and Xylazine (10 mg/kg, i.p.) and then the brains were dissected out and drop fixed in 10% neutral buffered formalin. Neuroanatomical studies were carried out using 3 homozygous *Camsap3*−/− and 24 matched baseline WT mice as well as 3 homozygous *Wdr47*tm1a/tm1a and 3 littermate WT mice, all on a C57BL/6NTac pure genetic background at 16-weeks of age. To facilitate data integration, CaMKIIαCre; *Wdr47*fl/fl conditional mice were studied at 16 weeks of age on the same genetic background. We calculated statistical power using a recognized model (Gpower) validated for comparison of small numbers of mice ($n = 3$) to evaluate neuroanatomical phenotypes with an effect size of 10% or more with a detection power of 80% (Collins et al, 2019). Paraffin embedded brain samples were cut at 5μm thickness using a sliding microtome (Micom HM 450) to obtain brain sections at Bregma −1.34 mm (*Camsap3*) or at Lateral +0.72 mm (*Wdr47*), according to the Allen Mouse Brain Atlas (Sunkin et al, 2013). Sections were stained with 0.1% Luxol Fast Blue (Solvent Blue 38; Sigma-Aldrich) and 0.1% Cresyl violet acetate (Sigma-Aldrich) and scanned using the high-resolution digital slide scanner (Nanozoomer 2.0HT, C9600 series) at 20× resolution. For *Camsap3*, three measurements were taken on the coronal plane and included the soma of the corpus callosum connecting the lateral part (lat) and the medial part (med) of the cerebral cortex, as well as the area of the dorsal hippocampal commissure (dhc). For *Wdr47*, we used a previously described procedure involving 40 measurements (Collins et al, 2018).

### Neuroanatomical characterization of embryonic murine brains

For male and female embryos at E18.5 ($n = 5$ WT vs $n = 4$ *Wdr47*tm1a/tm1a; $n = 6$ WT vs $n = 6$ NexCre; *Wdr47*fl/fl), the animals were kept cold on ice and killed by making a cut between the brain and spinal cord junction, before fixation in Bouin solution for 48 h. The embryonic brains were then harvested, transferred to 70% ethanol, and manually embedded in paraffin using the following steps: three incubation baths in 70% ethanol for 30 min each, two baths in 95% ethanol for 30 min each, two baths in 100% ethanol for 45 min each, three baths in Histosol Plus for 1 h each, and five baths in warm paraffin (60 °C) for 30 min each, followed by incubation in warm paraffin overnight before casting in a mold. Brains were cut at a thickness of 5 μm on a microtome (HM 450, Microm Microtech, France), such that we obtain sections matching planes at precise embryonic coronal sections, as explained in (Nguyen et al, 2022). The sections were stained with 0.1% Cresyl violet acetate (Sigma-Aldrich) and scanned using Nanozommer 2.0HT, C9600 series at 20× resolution.

Each image was quality controlled to assess whether (i) the section is at the correct position, (ii) the section is symmetrical, (iii) the staining is of good quality, and (iv) the image is good quality. Only images that fulfilled all of the quality control checks were fully processed. These quality control steps are essential for the detection of small to moderate neuroanatomical phenotypes and without which the large majority of neuro-anatomical phenotypes would be missed. This is explained in great details elsewhere (Nguyen et al, 2022). 54 brain morphological parameters made of area and length measurements were taken for each sample at E18.5 (Dataset EV2). Assessed embryonic brain regions included the total brain area, the cortices (motor, insular, somatosensory, retrosplenial granular and motor), the hippocampus, the genu of the corpus callosum, the internal capsule, the caudate putamen, the fimbria of the hippocampus, the anterior commissure and the ventricles (lateral and third). Every aspect of the procedure was managed through a relational database using the FileMaker (FM) Pro database management system (detailed elsewhere (Mikhaleva et al, 2016)). A list of histological parameters is also provided in Fig. 3. Data were analyzed using two-tailed Student's *t*-tests of equal variances.

### Plasmids and cloning

Wild-type (WT) human *WDR47* cDNA corresponding to Uni-protKB O94967-3 isoform of WDR47 was obtained from DNASU and subcloned by restriction-ligation into the NeuroD-IresGFP (Hand and Polleux, 2011) and the pcDNA3.1+/C-HA vectors. Constructs carrying the human *WDR47* variants p.(Arg193His), p.(Asp466His), p.(Lys592Arg), p.(Pro650Leu), and p.(His659Pro) were created from the WT CDS by Sequence and Ligation Independent Cloning (SLIC). Truncated constructs were generated from the NeuroD-WT-hWDR47-IresGFP and the pcDNA3.1-hWDR47-C-HA vectors by deletion of the LISH (amino acids 10–42), CTLH (amino acid 45–102) and the WD40 (amino acids 658–920) domains by SLIC. pFASBac+GFP-Camsap3 plasmid was obtained from Addgene. GFP fused to the N-terminal of Camsap3 was removed and subcloned by restriction-ligation into the NeuroD-Ires-GFP vector. Neuro-D:Cre-GFP was a gift from L. Nguyen (GIGA, Liege, Belgium) and pCAG2-mScarlet vector was provided by J. Courchet (INMG, Lyon, France). Plasmid DNAs used in this study were prepared using the EndoFree plasmid purification kit (Macherey Nagel).

### In utero electroporation

The uterine horns of time pregnant (E14.5 and E15.5) mice were exposed after anesthetizing the females with isoflurane (2 L per min of

oxygen, 4% isoflurane in the induction phase and 2% isoflurane during surgery operation; Tem Sega) and one ventricle of each embryo was injected with 0.5–1 μL of Endofree plasmid DNA solution mixed with 0.05% Fast Green (Sigma-Aldrich) using pulled glass capillaries (Harvard Apparatus, 1.0OD*0.58ID*100mmL) and a micro injector (Eppendorf Femto Jet). For migration analysis 3 μg/μL of NeuroD:Cre-GFP vector together with 1 μg/μL of either empty NeuroD-IRES-GFP or a rescue construct under the NeuroD promoter was used at E14.5. For CC analysis, the same plasmid mixes together with 0.8 μg/μl of pCAG2-mScarlet were used at E15.5. Plasmids were then further electroporated into the neuronal progenitors adjacent to the ventricle by discharging five electrical pulses of 40 V and 50 ms at 950 ms intervals using 3-mm platinum tweezers electrodes (Sonidel CUY650P3) and ECM-830 BTX square wave electroporator (VWR International). After electroporation, embryos were placed back in the abdominal cavity and the abdomen was sutured using surgical needle and thread. Males and females embryos were indifferently used for the study.

### Mouse brain fixation and cutting

For migration and CC analysis after IUE E18.5 to P8 animals were sacrificed by head sectioning and P21 pups were sacrificed by terminal perfusion by 0.9% NaCl followed by 4% PFA. For WDR47 expression pattern, E18.5 animals were sacrificed by head sectioning. All brains were then fixed in 4% paraformaldehyde (PFA, Electron Microscopy Sciences) in Phosphate buffered saline (PBS, HyClone) overnight at 4 °C. For IUE analysis, after fixation, brains were rinsed and embedded in a solution of 4% low-melting agarose (Bio-Rad) and cut into coronal sections (60 μm-thick for E18.5 and P2 mice, 100-μm thick for P3 to P21 mice) using a vibrating-blade microtome (Leica VT1000S, Leica Microsystems). Sections were maintained in PBS-azide 0.05% for short-term storage and in an Antifreeze solution (30% Ethyleneglycol, 20% Glycerol, 30% $H_2O$, 20% PO4 Buffer) for long-term storage.

For WDR47 immunostaining, after fixation, the brains were rinsed, equilibrated overnight at 4 °C in 20% sucrose in PBS, embedded in Tissue-Tek O.C.T. (Sakura), frozen on dry ice and cut into 18-μm-thick coronal sections using a cryostat (Leica CM3050S). Sections were maintained at −80 °C.

### Immunolabeling of mouse sections

Vibratome and cryo-sections were rinsed in PBS, permeabilized and blocked in 5% Normal Donkey Serum/PBST (1X PBS, 0.5% Triton X-100) and incubated with primary antibodies (Appendix Table S1) diluted in blocking solution at 4 °C overnight. After washing with PBST, the sections were incubated for 1 h at room temperature with fluorescence-conjugated secondary antibodies coupled to Alexa-488 or Alexa-647 (Thermo Fischer Scientific). Nuclei were counterstained with Dapi (1 μg/mL Sigma-Aldrich) and sections were mounted using Aquapolymount mounting solution (Polysciences Inc.). The slides were stored in the dark at 4 °C. The primary and secondary antibodies used are listed in Appendix Table S1.

### Image acquisition and analyses after IUE

All experiments were done in at least three independent replicates and analysis were performed blinded to condition. Cell counting and CC analyses were done in at least three different brain slices of at least three different embryos or pups for each condition. After histological examination, only brains with comparative electroporated regions and efficiencies were conserved for quantification.

Counting for neuronal migration. Images for migration analysis were acquired in $1024 \times 1024$ mode using confocal microscope (TCS SP8X; Leica) at 20× magnification with a z stack of 1.5 μm and analyzed using ImageJ software. For analyses, upper cortical plate, lower cortical plate, intermediate zone and subventricular zone/ventricular zone were identified according to cell density (nuclei staining with DAPI). The total number of GFP-positive cells in the embryonic brain sections was quantified by counting positive cells within a box of fixed size and the percentage of positive cells in each cortical area was calculated.

In vivo CC analyses. Images were acquired using a Zeiss Axio Observer 7 microscope equipped with a 10x/0.45 plan Apochromat objective controlled by zen 3.3 software, with a CMOS sensor camera (Hamamatsu Orca Flash 4.0 LT Plus) and analyzed using ImageJ software. For projecting neuron analyses, the mean fluorescence intensity within a box of fixed width covering the whole axonal thickness at the immediate start of the CC was measured and this value was divided by the mean fluorescence intensity in the cortical plate (Figs. 4C and 5E). For axon extension analysis, CC length was divided into 10 equal bins from the immediate start of the CC till the midline and fluorescence intensity in each bin was measured. Intensity in bin1 was considered 100% and intensity in each bin was normalized to bin1 (Figs. 4D and 5F). For axonal midline crossing analyses, the fluorescence within a box of fixed size placed at the contralateral side of the midline of the brain section was measured and divided by the fluorescence at the ipsilateral side of the midline area (Fig. 4E). For all analysis background fluorescence intensity was subtracted from the measured intensities.

### Primary neuronal culture

Cortices from either male or female E15.5 $Wdr47^{tm1b/tm1b}$ or WT mouse embryos were dissected in cold PBS supplemented with BSA (3 mg/mL), MgSO4 (1 mM, Sigma), and D-glucose (30 mM, Sigma). They were then dissociated in Neurobasal media containing papain (20 U/mL, Worthington) and DNase I (100 μg/mL, Sigma) for 20 min at 37 °C with brief vortexing every 6–7 min. Cells were then incubated for 7 min with Neurobasal media containing Ovomucoïde (15 mg/mL, Worthington), and manually triturated 10 times with a 1000 ml pipette in OptiMeM supplemented with D-Glucose (20 mM). Cells were then plated at $2 \times 10^5$ cells per 24-well plate or $3.5 \times 10^5$ cells per 35 mm diameter glass bottom dish (Cellvis, D35-20-1.5 h) coated with poly-D-lysine (1 mg/mL, Sigma) overnight at 4 °C and cultured for up to 10 days in Neurobasal medium supplemented with B27 (1×), L-glutamine (2 mM), and penicillin (5 U/mL)-streptomycin (50 mg/mL). Half of the media was changed with fresh supplemented Neurobasal media every 3 days.

### In vitro cell death experiments and drug treatments

Cultured neurons were magnetofected with 50 ng of Scarlet plasmid at DIV1 using NeuroMag (OZ Bioscience) according to the manufacturer's protocol and treated at DIV2 with either DMSO, QVd-Oph (Sigma - SML0063) (50 μM), Necrostatin (Sigma - N9037) (2 μM), Ferrostatin (Sigma - SML0583) (5 μM), or EpoD (Abcam - ab143616) (10 nM). Half of the media was changed every

3 days and neurons were imaged once every day from DIV3 to DIV10 using a Zeiss Axio Observer 7 microscope equipped with a 10x/0.45 plan Apochromat objective controlled by Zen 3.3 software, with a CMOS sensor camera (Hamamatsu Orca Flash 4.0 LT Plus) and an incubation chamber (37 °C, 5% $CO_2$).

### RNA extraction, cDNA synthesis, and RT‒qPCR

Total RNA was extracted from DIV4 or DIV6 primary neuronal cultures using the NucleoSpin RNA purification kit (Macherey-Nagel) and from human fibroblasts using Trizol reagent (Invitrogen). For RT-qPCR, 1 µg of total RNA was reverse transcribed with SuperScript IV (Invitrogen) using random hexamers (2 µM) and RT-qPCR was performed with the SYBR Green I Master (Life-Science) and specific primers (Appendix Table S2) according to the manufacturer's instructions using the Roche 480 LightCycler. *Actin* or *36B4* was used as the housekeeping gene and the relative amount of target mRNAs was determined using standard curve method.

### RNA sequencing

RNA was isolated from DIV6 primary neuronal cultures obtained from embryos of 3 different pregnant mothers, using the NucleoSpin RNA purification kit (Macherey-Nagel) according to the manufacturer's instructions. RNA-seq libraries were prepared from 600 ng of total RNA using the TruSeq® Stranded mRNA Library Prep kit and the TruSeq® RNA Single Indexes kits A and B from Illumina. The library quality and quantity were checked using an Agilent 2100 Bioanalyzer and a Qubit dsDNA HS Assay Kit. The sample concentration was adjusted to 2.8 nM before sequencing (50 bp single end) on a HiSeq 4000 (Illumina) using HiSeq 3000/4000 SR Cluster Kit and HiSeq 3000/4000 SBS Kit (50 cycles) according to the manufacturer's instructions.

Read alignment and quality assessment. Reads were preprocessed using cutadapt (Martin, 2011) version 1.10 in order to remove adapter, polyA and low-quality sequences (Phred quality score below 20); reads shorter than 40 bases were discarded for further analysis. Reads mapping to rRNA were also discarded (this mapping was performed using bowtie (Langmead and Salzberg, 2012) version 2.2.8). Reads were then mapped onto the mm10 assembly of mouse genome using STAR (Dobin et al, 2013) version 2.5.3a. Gene expression was quantified using htseq-count (Anders et al, 2015) version 0.6.1p1 with annotations from Ensembl release 102. Samples coming from one of the pregnant mothers were discarded from analysis as a higher plating density which in follow-up experiments was shown to effect timing of cell death, was noted for that culture.

Differential expression and pathway analysis. Differentially expressed genes (DEGs) between mutant and control samples were identified using R v. 3.3.2 (Ihaka and Gentleman, 1996) and DESeq2 (Love et al, 2014) version 1.16.1. Wald test was implemented in DESeq2 version 1.16.1 Bioconductor library with *p*-values adjusted for multiple testing using the Benjamini and Hochberg (BH) method (Benjamini and Hochberg, 1995). Genes with an adjusted *p*-value smaller than 0.05 were considered to be DEGs. Results of the differential expression analysis can be found in Dataset EV3. Pathway enrichment analysis of up- and down-regulated genes was performed separately using ClusterProfiler v. 4.8.2 (R v. 4.3.1) (Wu et al, 2021; Yu et al, 2012) and all genes found to be expressed in the dataset were used as gene universe. Only pathways with minimum

size of 10 and maximum size of 250 genes were included in the analysis. Ontologies with a BH-adjusted *p*-value < 0.05 were considered significant. Highly similar GO terms were replaced by the most significant one using GoSemSim implemented in ClusterProfiler. In case of ties, pathways with identical gene contribution and *p*-value were manually collapsed in the most representative one. For disease ontology analysis mouse genes were converted to human orthologs and a gene set enrichment analysis (GSEA) was run using Cluster Profiler by ranking the genes by log2FC and *p*-values were BH-corrected.

### Cell culture and transfections

All cells provided by the cell culture platform of the IGBMC (Strasbourg) are mycoplasma free (PCR test Venorgem) and have not been authenticated. Human fibroblasts, Human embryonic kidney (HEK) 293T, WT and *WDR47*-KO (Guardia et al, 2021) HeLa cells were cultured in DMEM (1 g/L glucose) (GIBCO) supplemented with 10% Fetal Calf Serum (FCS), penicillin 100 UI/mL, streptomycin 100 µg/mL. Human LCLs were cultured in RPMI 1640 (Gibco) supplemented with 10% Fetal Bovine Serum (FBS), penicillin 100 UI/mL, streptomycin 100 µg/mL. Mouse neuroblastoma N2A (ATCC) cells were cultured in DMEM (1 g/L glucose) (GIBCO) supplemented with 5% FCS and Gentamycin 40 µg/mL. All cells were kept in a humidified atmosphere containing 5% $CO_2$ at 37 °C.

For expression analysis of truncated *WDR47* constructs HEK cells were transfected with different HA tagged Wdr47 truncated constructs using Lipofectamine 2000 (Invitrogen) according to the manufacturer's protocol. Expression of transfected constructs was analyzed 48 h after transfection by immunoblotting. To study the effect of human variants on WDR47 levels, N2A cells were transfected with different HA tagged WDR47 variants using Lipofectamine 2000 (Invitrogen) according to the manufacturer's protocol. Expression of transfected constructs was analyzed 24 h after transfection by immunoblotting. For MG132 experiments, transfected cells were treated with 10 µM MG132 8 h prior to collection.

### Migration assay using primary human fibroblasts

Primary human fibroblasts were seeded in 96-well plates at 50,000 cells per well. At confluency, the scratch assay was performed by creating a linear wound in the wells. Cells were incubated 2 h with media containing mitomycin (10 µg/mL, Sigma) to inhibit proliferation and Hoechst 33342 (2 µg/mL). After medium replacement, images were acquired every 3 h for 36 h on Cellomics Arrayscan using 5× objective and live cell chamber. Image J was used to measure wound area over time using 4-field Photoshop assembly.

### Protein extraction and western blot

Proteins from mouse cortices (E18.5), primary neuronal cultures (DIV4 and DIV6), transfected HEK 293T cells and N2A cells, HeLa cells, human fibroblasts and human lymphocytes were extracted as follows: cortices and cells were lysed with RIPA buffer (0.01 M Hepes, 0.15 M NaCl, 0.01 M EDTA, 2.5 mM EGTA, 0.1% Triton X-100, 0.1% SDS, 1% Na deoxycholate, 2% NaF)) supplemented with EDTA-free protease inhibitors (cOmplete™, Roche) for 30 min. Cell debris was removed by high speed centrifugation at 4 °C for 25 min and protein concentration was determined using Bio-Rad Bradford

protein assay reagent. Samples were denatured at 95 °C for 10 min in Laemmli buffer (Bio-Rad) with 2% β-mercaptoethanol and the indicated amount of proteins were resolved by SDS–PAGE and transferred onto PVDF membrane (Immobilon-P). Using neuronal extracts, 500 ng protein was loaded to detect actin, alpha tubulin, tyrosinated tubulin and 2 μg of protein was loaded to detect acetylated tubulin. Using extracts from human fibroblasts and HeLa cells, 5 μg of proteins were loaded to detect GAPDH, alpha tubulin, tyrosinated tubulin and 30 μg proteins were loaded to detect acetylated tubulin. 10 μg and 5 μg of protein extracts from HEK 293T and N2A cells transfected with different WDR47-HA constructs were used, respectively, to detect HA and GFP in those cells. 20 μg of proteins from mouse cortices, human fibroblasts and lymphocytes were used to detect endogenous WDR47 levels. Membranes were blocked in 5% milk in PBS buffer with 0.1% Tween (PBS-T) and incubated overnight at 4 °C with the appropriate primary antibodies (Appendix Table S1) in blocking solution. Membranes were washed 3 times in PBS-T, incubated at room temperature for 1 h with HRP-coupled secondary antibodies (Invitrogen) (Appendix Table S1) at 1:10,000 dilution in PBS-T, followed by 3 times PBS-T washes. The revelation was done by chemiluminescence using SuperSignal West PicoPLUS or Super-Signal™ West Femto Chemiluminescent Substrates (Thermo Fischer) and gels were imaged using the Amersham Imager 600. Relative protein expression was quantified using ImageJ software.

### Live mitochondria imaging and analysis

At DIV4, neurons were transfected with a Venus plasmid (50 ng) and mitoDs-Red (250 ng) by magnetofection. At DIV6, 10 z-stacks of 0.2 μm step size was imaged from the medial part of the axon (~500–800 μm from soma) using a confocal microscope Spinning Disk CSU-X1 "Nikon", equipped with a 100× oil-immersion objective (N.A. 1.4) controlled by the Metamorph 7.10 software, with a CMOS sensor camera (Photometrics Prime 95B) and an incubation chamber (37 °C, 5% $CO_2$). A home-made ImageJ macro that selects the mitochondria based on red channel intensity was used to quantify the number and morphologic characteristics of the mitochondria.

### Lysosome and mitochondria motility

At DIV4 neuronal cultures were magnetofected with either LAMP1-YFP (500 ng) or mito-Dsred (250 ng) to label lysosomes and mitochondria, respectively. Sparsely labeled cultures were selected to allow selection of single axons and determine their orientation unambiguously. The medial part of the axon (~500–800 μm from soma) was chosen and images were acquired every 750 ms for 7 min for lysosomes and every 1 s for 5 min for mitochondria. For analyzing lysosome motility, primary human fibroblasts were labeled with 100 nM Lysotracker Red DND-99 (Thermo Fisher) for 45 min and images were acquired every 1 s for 2 min in trailing processes of fibroblasts. All images were taken using a Leica spinning disk microscope (CSU-W1) equipped with an Adaptative Focus Control (AFC), a 63× oil-immersion objective (NA 1.4), an Orca Flash 4.0 camera and an incubation chamber (37 °C, 5% $CO_2$). All acquisition settings were set to keep the signal in a dynamic range and laser powers were kept at minimum to prevent photobleaching. Kymographs were obtained from the time-lapse recordings using a custom written ImageJ plugin Kymo ToolBox v.1.01 (https://github.com/fabricecordelieres/IJ-Plugin_KymoToolBox/releases). Mitochondria and lysosomes that

have displacement <6 μm and <10 μm, respectively, in neurons and <3 μm in fibroblasts was considered stationary to avoid bias due to stage drift. All dynamic parameters of intracellular transport are from data that were obtained for each condition in 10–15 axons from at least three independent neuronal cultures and 20 cells from 2 independent fibroblasts cultures.

### Measurement of mitochondrial membrane potential in neurons

DIV6 neurons were co-labeled with 20 nM potentiometric fluorescent probe TMRE (tetramethylrhodamine, ester ethylique, perchlorate) (Sigma) and 50 nM MitoTracker Green FM dye (ThermoFisher) at 37 °C 5% $CO_2$ for 30 min, maintained with 5 nM TMRE for 10 min to achieve equilibration and kept in the same media during subsequent live imaging for a maximum of 20 min. Images with a z-stack of 0.3 μm were acquired using a Leica spinning disk microscope (CSU-W1), equipped with a 63× oil-immersion objective (NA 1.4), an Orca Flash 4.0 camera and an incubation chamber (37 °C, 5% $CO_2$). To select mitochondria in the neurites, images were segmented based on MitoTracker Green labeling using the Imaris program and TMRE intensity was analyzed for 500–1000 mitochondria from 3 to 5 independent experiments.

### Measurement of mitochondrial redox potential in neurons

Neurons were transfected with 250 ng of mitochondria targeted roGFP2 plasmid (Grx1-roGFP2) by magnetofection at DIV1 and live imaging of mitochondria in neurites was performed using a confocal microscope (TCS SP8X; Leica) equipped with 63× oil immersion objective and an incubation chamber (37 °C, 5% $CO_2$) at DIV6. Confocal settings were pixel size 70.81 (1304 × 1304 pixels), scan speed 400 Hz, laser power 405 nm 20%, laser power 488 nm 30%, emission bandwidth 498–560, line accumulation of 3 and 2X zoom. For each condition, 405:488 nm ratio of each mitochondrion was calculated in 10–15 axons from at least 3 independent neuronal cultures. Increased 405:488 ratios toward a more oxidized state suggest increased ROS production.

### Flow cytometry analysis to determine mitochondrial mass and membrane potential in HeLa cells

Mitochondrial mass was quantified using a flow cytometry-based assay modified from existing protocols (Puleston, 2015). A single cell suspension was prepared by trypsinization of HeLa cells and $0.5 \times 10^6$ cells were transferred to a round bottom 96-well plate. Cells were pelleted by centrifugation and supernatant was discarded. To measure mitochondrial mass, cells were resuspended in 100 μL PBS + 5% FCS containing 150 nM MitoTracker Green FM (Life Technologies M-751) and incubated for 25 min at 37 °C. To measure mitochondrial membrane potential, cells were resuspended in PBS containing 100 nM TMRE (Sigma) and incubated for 15 min at 37 °C. Cells were then washed, centrifuged and resuspended in PBS + 2% FSC. Flow cytometry was carried out using the BD LSR FortessaTM cell analyzer, equipped with FACSDIVA (BD Biosciences) software and data were analyzed using FlowJo 10 software (TreeStar). For each experiment, cells were carefully gated to include only viable population in the analysis and 30,000 were analyzed in each condition of 9 independent experiments.

### Lysosome distribution in HeLa cells

HeLa cells were cultured on glass coverslips in 24-well plates, transfected with GFP expressing vector to label the cytoplasm using

Lipofectamine 2000 (Invitrogen) according to the manufacturer's protocol, fixed after 24 h in 4% PFA in PBS for 15 min, washed 3 × 5 min with PBS, permeabilized for 10 min in 1X PBS + 0.3% Triton X-100, blocked for 1 h in 1X PBS + 5%NDS and incubated with LAMP1 and GFP antibodies (Appendix Table S1) diluted in blocking solution at 4 °C overnight. After washing with PBS, coverslips were incubated for 1 h at room temperature with fluorescence-conjugated secondary antibodies (Appendix Table S1). Nuclei were counterstained with Dapi (1 µg/mL Sigma-Aldrich) and coverslips were mounted using Aquapolymount mounting solution (Polysciences Inc.). Single focal plane images were acquired using the LiveSR mode on Spinning Disk CSU-X1 Nikon equipped with a 100× oil-immersion objective (N.A. 1.4) controlled by the Metamorph 7.10 software, with a CMOS sensor camera (Photometrics Prime 95B). Image analysis was done as described in (Schmied et al, 2024). Cell segmentation was done manually in ImageJ using the GFP channel. Nuclei segmentation and organelle detection was done using the ImageJ plugin available via the update site (https://sites.imagej.net/Cellular-Imaging/). "Calibrated distance of each organelle" and "ferret diameter" was extracted using the plugin and "normalized detection distance of each organelle" was calculated by dividing the "calibrated distance of each organelle" by "ferret diameter of each cell" as described in (Schmied et al, 2024). The data was analyzed using the nested mixed effect model that allows to take into consideration that each lysosome is nested within one cell.

### Microtubule repolymerization assay

To depolymerize microtubules, human fibroblasts seeded on glass coverslips in P24 plates were incubated on ice for 30 min with cold culture media without serum (DMEM (1 g/L glucose) (GIBCO) supplemented with penicillin 100 UI/mL, streptomycin 100 µg/mL) containing 1.5 µg/ml Nocodazole (Sigma) in the cold room. Cells were washed once with warm media to remove the nocodazole and either fixed directly to check for depolymerization or fixed after 2 min of incubation with warm media to allow microtubule repolymerization. Cells were fixed with cold methanol for 7 min at −20 °C, washed 2 times for 5 min with PHEM buffer (60 mM PIPES, 25 mM Hepes, 10 mM EGTA, 2 mM MgCl₂, pH 6.8), blocked for 1 h with 5% NDS and 0.3% Triton-X in PHEM buffer and incubated with tyrosinated tubulin antibody (Appendix Table S1) at 4 °C overnight. After washing with PHEM buffer, the coverslips were incubated for 1 h h at room temperature with fluorescence-conjugated secondary antibodies coupled to Alexa-561 (Appendix Table S1). Nuclei were counterstained with Dapi (1 µg/mL Sigma-Aldrich) and sections were mounted using Aquapolymount mounting solution (Polysciences Inc.).

### Plasmids, strains, media, and methods for yeast cells

The human WDR47 cDNA either full-length (FL), or lacking the CTLH (ΔCTLH) and LISH (ΔLISH) domain were cloned by the Gateway® (Invitrogen) method into pDONR221 entry vector and then recombined into a yeast low-copy number CEN destination vector (Addgene, (Alberti et al, 2007)) to obtain pAG413-promGPD-WDR47 (pSF634), pAG413-promGPD-WDR47-ΔCTLH (pSF636) or pAG413-promGPD-WDR47-ΔLISH (pSF637) plasmids bearing the HIS3 auxotrophic marker for selection of the transformants on SC-His medium. Plasmid sequences were verified (GATC Biotech). The promCUP1-mCherry-V5-ATG8 (pFL78, LEU2 selection marker) plasmid was a kind gift from Fulvio Reggiori (Aarhus Institute of

Advanced Studies, Aarhus University, Denmark). The *Saccharomyces cerevisiae* wild-type BY4742 (*MATα leu2Δ0 ura3Δ0 his3Δ0 lys2Δ0*) reference strain and the *oxa1Δ* (*MATα leu2Δ0 ura3Δ0 his3Δ0 lys2Δ0 oxa1::KanMX*) mutant strain were used. The indicated yeast strains were grown at 30 °C in rich medium YPD: 1% yeast extract, 2% peptone, 2% glucose, in synthetic complete medium to maintain the plasmid SC-His: 0.67% yeast nitrogen base (YNB, MP Biomedicals) without amino acids, 2% glucose and the appropriate -His dropout mix (CSM-His, MP Biomedicals) or in autophagy induction medium SD-N: 0.17% YNB without ammonium sulfate without amino acids, 2% glucose. Yeast cells were transformed using the modified lithium acetate method (Gietz et al, 1992).

For western-blot analysis, total yeast protein extracts were obtained by NaOH lysis of 1.5 unit at OD-600 nm of yeast cells, followed by trichloroacetic acid (TCA) precipitation and the pellet was resuspended in 50 µL of 2X Laemmli buffer plus Tris Base. Samples were incubated 5 min at 37 °C prior anti-WDR47 western-blot analysis using standard procedures. TCE (2,2,2-Trichloroethanol) staining was used for total protein detection and loading control (Ladner et al, 2004). Images were acquired with the ChemiDoc Touch Imaging System (Bio-Rad).

Drop test growth assays were done on the indicated yeast cells grown from SC-His precultures to exponential phase in YPD for 4 h at 30 °C, prior plating on YPD-2% agar medium as 7 µL drops of serial dilutions at OD-600 nm of 0.5, 0.1, 0.01, and 0.005. The plates were incubated at 30 °C.

Autophagy was analyzed on yeast cells bearing the empty pAG413 plasmid (control) or expressing Wdr47 (FL, ΔCTLH or ΔLISH) constructs transformed with the mCherry-ATG8 expression vector (pFL78). Cells were grown in SC-His-Leu +CuSO₄ (1 mM) medium to induce the expression of mCherry-V5-ATG8, at OD-600 nm 0.5–1 cells were collected before washing with autophagy SD-N medium, and incubation in SD-N medium at 30 °C for 4 h prior observation by fluorescent microscopy. Observation was performed with 100×/1.45 oil objective (Zeiss) on a fluorescence Axio Observer D1 microscope (Zeiss) using DsRED filter and DIC optics. Images were captured with a CoolSnap HQ2 photometrix camera (Roper Scientific) and treated by ImageJ (Rasband W.S., ImageJ, U. S. National Institutes of Health, Bethesda, Maryland, USA, http://imagej.nih.gov/ij/).

### Statistics

All statistics were calculated using GraphPad Prism 6 (GraphPad) and are represented as mean +/− s.d. or mean +/− SEM with the exception of Fig. 3 which shows the average between data points for simplicity purposes. All statistical tests used and n numbers have been mentioned in Figures legends along with the respective data and statistical details are reported in Dataset EV4. Analysis of IUE experiments and neuroanatomical characterization of mouse brains were performed blinded. Normality was checked using Shapiro–Wilk or KS normality test depending on the sample size, when the data was big enough. In case normality was violated, non-parametric tests were used. Graphs were generated using GraphPad and images were assembled with Adobe Photoshop 13.0.1 (Adobe Systems).

## Data availability

All other relevant data included in the article are available from the authors upon request. The following databases and in silico

## The paper explained

### Problem

Brain development is a complex process where different structures of the brain grow and make connections that help us learn, remember, and control our movements. One key structure involved in these functions is the corpus callosum (CC), a bundle of nerve fibers that connects the two halves of the brain. Defective development of CC can lead to a range of disorders known as corpus callosum dysgenesis (CCD). CCD can occur on its own or along with other issues, such as smaller brain size, which can increase the risk of cognitive and mental health problems. There are over 300 genetic conditions linked to CCD, but many of these are still not well understood or diagnosed.

### Results

In this study, we used exome sequencing to find a new genetic locus linked to CCD and discovered five variants in the *WDR47* gene. This gene is important for regulating various brain functions, including maintaining healthy mitochondria, managing microtubule dynamics, and ensuring proper axonal transport. We studied seven patients, three of whom share the same variant, and assessed variant pathogenicity using various mouse models. Our findings show that a certain level of WDR47 activity is needed for the proper development of the corpus callosum. When WDR47 activity drops below 50%, it leads to more severe problems: a complete loss of WDR47 causes early-onset neuronal degeneration and the absence of the corpus callosum, while partial loss results in a thinner corpus callosum, likely due to defects in nerve fiber growth.

### Impact

This study reveals how important the *WDR47* gene is for brain development and how changes in this gene can lead to brain disorders, including corpus callosum dysgenesis. Our research highlights the need to consider *WDR47* variants when diagnosing unexplained brain developmental disorders.

software were used in the study: Human Gene Mutation Databases (http://www.hgmd.cf.ac.uk/ac/introduction.php?lang=english), the single Nucleotide Polymorphism database (http://ftp.ncbi.nih.gov/snp/), genome aggregation database (gnomAD browser v2.1.1 and v3.1.2 (https://gnomad.broadinstitute.org/)), 1000 genomes (https://www.internationalgenome.org/), University of Santa Cruz Genomic Browser (https://genome.ucsc.edu/index.html), ClinVar (https://www.ncbi.nlm.nih.gov/clinvar/), Mastermind by Genomenom (https://mastermind.genomenon.com/), Leiden Open Variation Database – LOVD (https://www.lovd.nl/), OMIM – Online Mendelian Inheritance in Man (https://www.omim.org/), Integrative Genomics Viewer - IGV Browser (https://software.broadinstitute.org/software/igv/), Polyphen-2 (http://genetics.bwh.harvard.edu/pph2/), Mutation Taster (http://www.mutationtaster.org/), Sorting Intolerant from Tolerant (SIFT, https://sift.bii.a-star.edu.sg/), and Combined Annotation Dependent Depletion (CADD, https://cadd.gs.washington.edu/). The hWDR47 variants have been deposited in LOVD (Leiden Open Variation Database) v3.0 under the accession numbers #0000953686 (p.(Pro650Leu); https://databases.lovd.nl/shared/variants/0000953686#00025892), #0000953780 ((p.(Lys592Arg); https://databases.lovd.nl/shared/variants/0000953780#00025892) and #0000953781 (p.(Asp466His); https://databases.lovd.nl/shared/variants/0000953781#00025892). RNA-seq data are available in the NCBI GEO database under the record GSE247160. All images used

for this study are accessible through the BioImage repository (Accession number: S-BIAD1389).

The source data of this paper are collected in the following database record: biostudies:S-SCDT-10_1038-S44321-024-00178-z.

## Peer review information

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

## Acknowledgements

This work was supported by grants from INSERM (ATIP-Avenir program, JDG), the Fyssen Foundation (JDG), the French state funds through the Agence Nationale de la Recherche (JCJC CREDO ANR-14-CE13-0008-01 to JDG; JCJC WDR ANR-18-CE12-0009 to BY; ANR-10-IDEX-0002-02 and ANR-10-LABX-0030-INRT to JDG), SFRI-STRAT'US project [ANR 20-SFRI-0012]; and EUR IMCBio [ANR-17-EURE-0023]), INSERM/CNRS (JDG and SF), University of Strasbourg (JDG and SF) and Intramural Research Program of the National Institute of Environmental Health Sciences (project ZIA ES103370-01 to CMG). The authors also acknowledge the support of the Inserm Cross-Cutting Scientific Program (HuDeCA to PG). EB was supported by INSERM, Fondation Jerome Lejeune and ANR. JRA was supported through the IGBMC PhD program and Fondation pour la recherche médicale. LT was supported by INSERM through a training grant. SCC is a senior lecturer at the University of Bourgogne; LT is a technician; and MeK a PhD student supported through the Agence Nationale de la Recherche (ANR-11-PDOC-0029-01 to BY). RL are supported by Ministère de l'Enseignement Supérieur de la Recherche et de l'Innovation. PT is research assistant at the University of Strasbourg. SF is a CNRS investigator. BY and JDG are INSERM investigators. HS, MN, and MiK were funded by the Japan Society for the Promotion of Science, Grant-in-Aids for Scientific

Research (JP23K27566 to HS; JP21K06819 to MN and JP24K11055 to MiK), Japan Agency for Medical Research and Development (AMED; grant numbers: JP23ek0109549, JP23ek0109674, JP23ek0109637 to HS and JP24ek0109591 to MiK), and Takeda Science Foundation Specific Research Grants (HS and MN), HUSM Grant-in-Aid from Hamamatsu University School of Medicine (HS and MN), and the Ministry of Health, Labor, and Welfare Research Program on Rare and Intractable Diseases under grant number JPMH23FC0201 (MiK). We thank the Imaging Center of IGBMC (https://ici.igbmc.fr/) part of the IBiSA labeled PIQ-QuESt (https://piq.unistra.fr) and of the national infrastructure France Bioimaging (https://france-bioimaging.org) supported by the French National Research Agency (ANR-10-INBS-04), in particular Elvire Guiot and Erwan Grandgirard, for their assistance in the imaging experiments. We are grateful to the staff of the molecular biology service (in particular, Thierry Lerouge and Paola Rossolillo), of the mouse facilities of the Institut Clinique de la Souris (ICS) (in particular Sophie Brignon), of the GenomeEast platform (in particular Celine Keime), of the flow cytometry service (in particular Claudine Ebel) of the IGBMC PluriCell East platform and and of the ImaFlow core facility (Biologie Santé Dijon BioSanD US58, 21079, Dijon, France) supported by Burgundy Regional Council (in particular Amandine Bataille and Audrey Geissler) for their involvement in the project. We also thank Oktay Cakil for the qPCR with human fibroblasts and Marie-Christine Fischer, Nina Pigeonneau, Perrine F Kretz and Christel Wagner for their helps with histology, image analysis and genotyping for the Nex and CaMKII neuroanatomy studies. We warmly thank Dr. Julien Courchet and Dr. Marine Lanfranchi (Institut NeuroMyoGene, Lyon, France) for sharing plasmids and their helpful comments and advice for mitochondrial assays, Dr. Clement Charenton (IGBMC, Strasbourg, France) for his help on modeling the WDR47 variants based on WDR47 structure and Dr. Fulvio Reggiori (Aarhus University, Aarhus, Denmark) for sharing the mCherry-Atg8 plasmid. We also would like to thank Dr. Frederic Saudou, Dr. Chiara Scaramuzzino, Dr. Sandrine Humbert and Mariacristina Capizzi (Grenoble Institute of Neurosciences, Grenoble, France) and Dr Juan Bonifacino and Dr Raffaella de Pace (Eunice Kennedy Shriver National Institute of Child Health and Human Development (NICHD), Bethesda, USA) for helpful comments on experimental design. We are also grateful to Amélie Piton (IGBMC) and to members of BY's and JDG's laboratory for discussion and technical assistance and in particular to Aline Dubos for critical reading of the manuscript.

## Author contributions

**Efil Bayam**: Conceptualization; Data curation; Formal analysis; Supervision; Validation; Investigation; Visualization; Writing—original draft; Writing—review and editing. **Peggy Tilly**: Conceptualization; Data curation; Formal analysis; Investigation; Methodology; Writing—review and editing. **Stephan C Collins**: Conceptualization; Data curation; Formal analysis; Investigation; Methodology. **José Rivera Alvarez**: Conceptualization; Data curation; Formal analysis; Investigation. **Meghna Kannan**: Conceptualization; Data curation; Formal analysis; Investigation. **Lucile Tonneau**: Investigation; Methodology. **Elena Brivio**: Data curation; Formal analysis; Methodology. **Bruno Rinaldi**: Formal analysis; Investigation; Methodology. **Romain Lecat**: Formal analysis; Investigation. **Noémie Schwaller**: Formal analysis; Investigation. **Ludovica Cotellessa**: Formal analysis; Investigation. **Sateesh Maddirevula**: Formal analysis; Investigation. **Fabiola Monteiro**: Formal analysis; Investigation. **Carlos M Guardia**: Resources. **João Paulo Kitajima**: Formal analysis; Investigation. **Fernando Kok**: Formal analysis; Investigation. **Mitsuhiro Kato**: Formal analysis; Investigation. **Ahlam A A Hamed**: Formal analysis; Investigation. **Mustafa A Salih**: Formal analysis; Investigation. **Saeed Al Tala**: Formal analysis; Investigation. **Mais O Hashem**: Formal analysis; Investigation. **Hiroko Tada**: Formal analysis; Investigation. **Hirotomo Saitsu**: Formal analysis; Investigation. **Mariano Stabile**: Formal analysis; Investigation. **Paolo Giacobini**: Conceptualization; Resources; Formal analysis; Supervision; Methodology. **Sylvie Friant**: Conceptualization; Formal analysis; Supervision; Methodology; Writing—review and editing. **Zafer Yüksel**: Formal analysis; Investigation; Writing—review and editing. **Mitsuko Nakashima**: Formal analysis; Investigation. **Fowzan S Alkuraya**: Resources; Formal analysis; Investigation; Writing—review and editing. **Binnaz Yalcin**: Conceptualization; Resources; Data curation; Formal analysis; Supervision; Funding acquisition; Investigation; Writing—original draft; Project administration; Writing—review and editing. **Juliette D Godin**: Conceptualization; Resources; Data curation; Formal analysis; Supervision; Funding acquisition; Investigation; Writing—original draft; Project administration; Writing—review and editing.

In addition to the CRediT author contributions listed above, the contributions in detail are: EB conceived and performed the experiments, coordinated and supervised the study, analyzed the data, performed statistical analysis and wrote the manuscript with contributions from all other authors. PT conceived and performed in utero electroporation, collected the mouse samples, processed the tissues and did immunostainings, performed the analysis of migration, did genotyping and took care of the mouse colonies and coordinated the in vivo studies. SCC, MeK, and LT did neuroanatomical characterization of NexCre and CaMKIICre KO mice. JRA performed lysosome motility and mitochondrial morphology experiments. EBr helped with the analysis of RNA-seq data, re-analyzed WDR47 expression in mouse and human tissue from available datasets and prepared the Figures. BR did the clonings, western blot, transformation and growth test assay for the yeast experiments. RL did the qPCRs and WBs in neuronal cultures. NS did WBs in N2A and patients cells. LC and PG performed immunostaining on human samples. SM, FM, JPK, FK, MiK, AAAH, MAS, SAT, MOH, HT, HS, MS, ZY, MN, and FSA followed-up the patients and families, provided the clinical and imaging data and contributed to the generation of exome sequencing, bioinformatics tools and analysis of sequencing data. CMG generated the WDR47 KO HeLa cells. SF performed analysis of yeast data and contribute to the writing of the manuscript. BY and JDG conceived, coordinated and supervised the study and wrote the manuscript with contributions from all other authors. Source data underlying figure panels in this paper may have individual authorship assigned. Where available, figure panel/source data authorship is listed in the following database record: biostudies:S-SCDT-10_1038-S44321-024-00178-z.

## Disclosure and competing interests statement

The authors declare no competing interests.

# Expanded View Figures

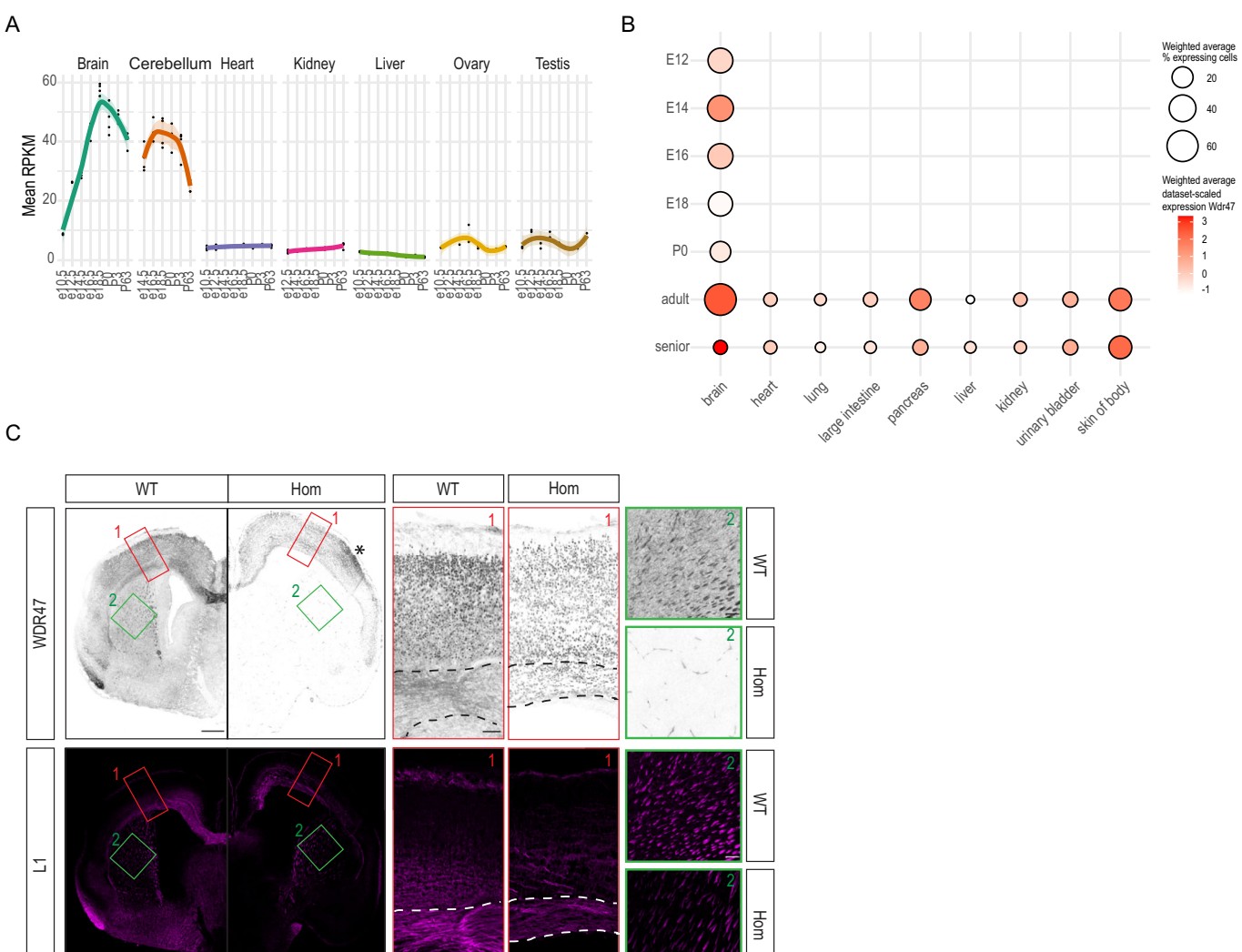

**Figure EV1.** **Pattern of expression of mouse *Wdr47*.**

(**A**) Expression values of *Wdr47* as RPKM (Reads per kilo base per million mapped reads) throughout life for different organs in mouse. Dots represent average values for each replicate. Shaded regions represent standard deviation of distribution. Data from (Cardoso-Moreira et al, 2019). (**B**) Expression pattern of *Wdr47* in mice across organs (x axis) and ages (y axis) showing prominent expression in the brain. Data obtained by multiple published dataset through https://cellxgene.cziscience.com (Abdulla et al, 2023). Color scale represents the weighted average expression across datasets and dot size represents the weighted average percentage of cells expressing *Wdr47*. (**C**) Coronal sections of E18.5 WT and HOM (*Wdr47^tm1b/tm1b*) mouse brains immunostained with WDR47 and L1 antibodies show expression of WDR47 in the cortex with enrichment in fiber tracks. Asterisk depicted unspecific staining. Scale bars: 360 μm and 60 μm (insets).

A

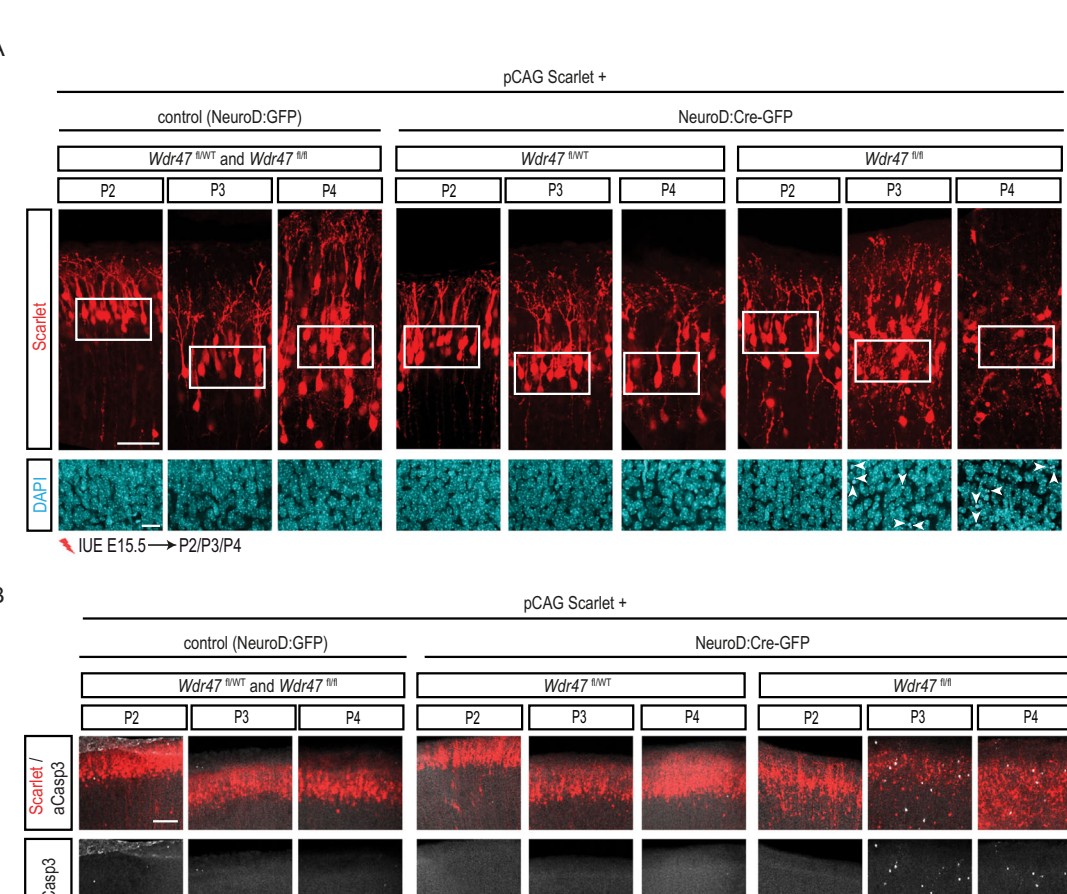

B

C

D

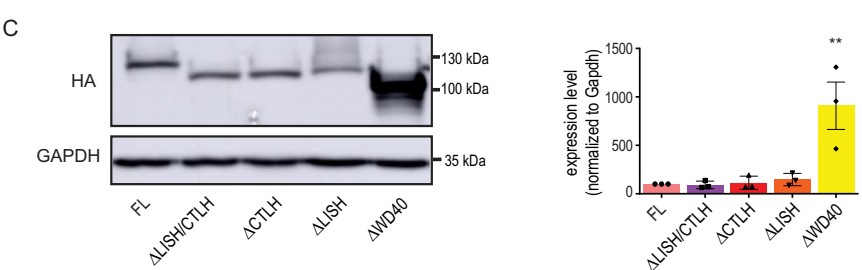

◀ **Figure EV2. Loss of *Wdr47* induces massive neuronal death at early postnatal stages.**

(A, B) Coronal sections of P2, P3, and P4 mouse brains electroporated at E15.5 with pCAG:Scarlet plasmid together with either a control (NeuroD:Ires:GFP) or a NeuroD:Cre-GFP vector. Scarlet positive electroporated neurons are depicted in red. (A) While *Wdr47*$^{fl/WT}$ neurons and neurons in control conditions keep a proper morphology, *Wdr47*$^{fl/fl}$ neurons lose their bipolar morphology from P3 on. In close-up views of the white boxed area, nucleus is counterstained with DAPI and arrowheads point to pyknotic nuclei. Scale bars: 50 μm and 20 μm (insets). (B) Coronal sections are immunolabelled for activated Caspase3 (aCasp3) (white). Several aCasp3+ cells appear at P3 and P4 in *Wdr47*$^{fl/fl}$ condition. Scale bar: 100 μm. (C) Western blot analysis of extracts from HEK cells transfected with the indicated HA tagged hWDR47 truncated constructs. Gapdh is used as the loading control. Data (means ± s.d.) from at least 3 independent experiments were analyzed by one-way ANOVA, with Bonferroni's multiple comparisons test. **$P < 0.01$. Note that ΔWD40 construct is expressed about 10 times more than the other constructs. (D) Effect of different rescue constructs on CC and neuronal survival in control conditions. Coronal sections of P4 *Wdr47*$^{fl/WT}$ mouse brains electroporated at E15.5 with pCAG:Scarlet and NeuroD:Cre-GFP plasmids together with a truncated WDR47 construct. Scarlet positive electroporated neurons are depicted in black. Close-up views of the green and orange boxed area show no effect of the construct on the CC and neuronal survival. Data from at least 3 independent experiments. Scale bars: 500 μm, 200 μm (green boxed inset) and 50 μm (red boxed inset). Exact *P* values are listed in Dataset EV4.

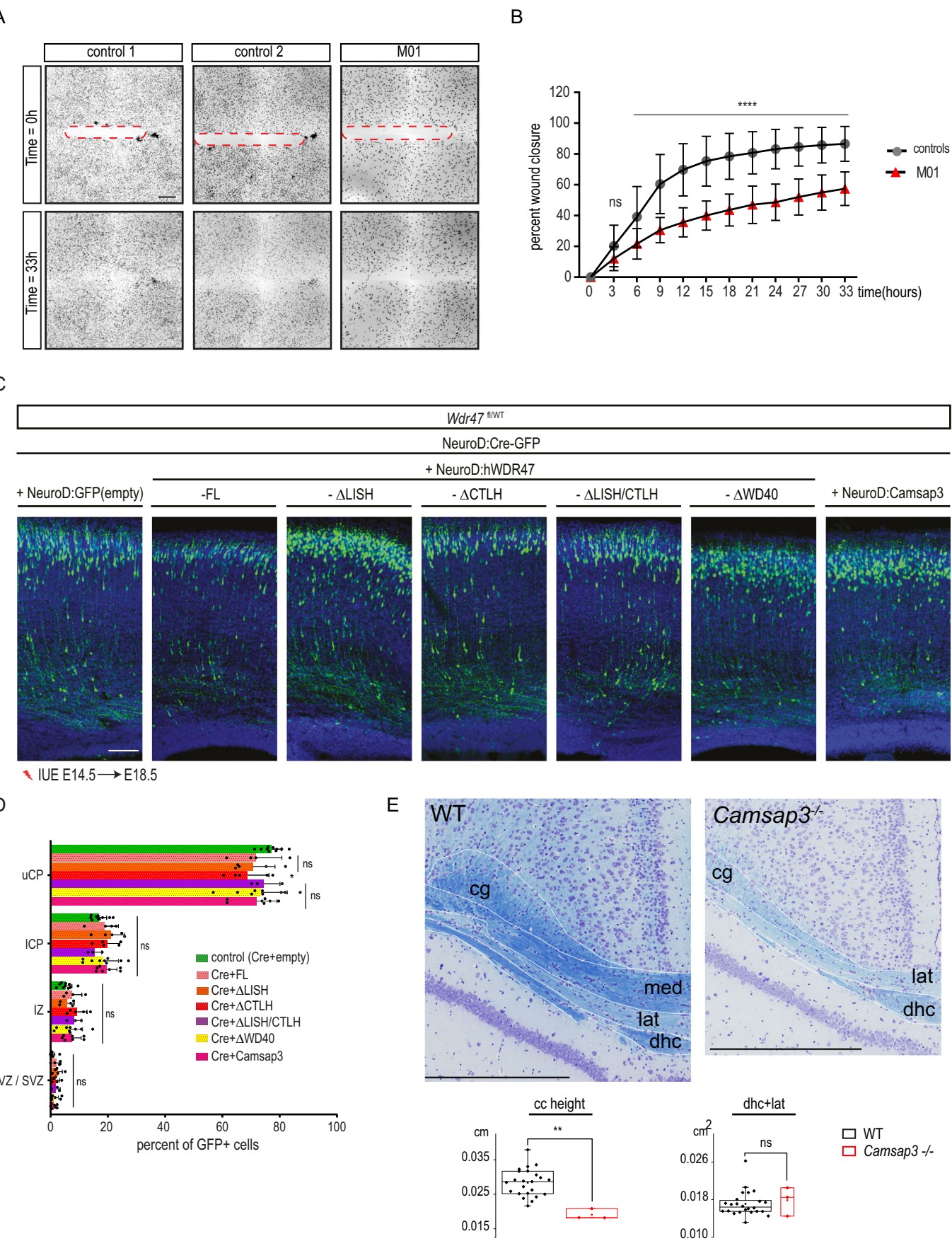

◀ **Figure EV3. Effect of different rescue constructs on neuronal migration in control conditions.**

(A) Transmission light micrographs of an in vitro 36-h neuronal migration assay performed on Mitomycin-treated fibroblast lines obtained from two healthy subjects and patient M01. The dashed red lines show the edge of the wound. Scale bar: 500 μm. (B) Percentage of wound closure is shown over time in M01-derived fibroblasts compared to control lines. Data (means ± s.d.) from at least 18 wells per condition was analyzed by two-way ANOVA, with Bonferroni's multiple comparison test, ns, non-significant, ****$P < 0.0001$. (C) Coronal sections of E18.5 $Wdr47^{fl/WT}$ mouse cortices 4 days after in utero electroporation with NeuroD-Cre-GFP together with a different NeuroD construct used for rescue experiments. GFP-positive electroporated cells are depicted in green. Nuclei are stained with DAPI. The representative image of NeuroD:GFP (empty) electroporation in $Wdr47^{fl/WT}$ embryo is identical to the one shown in Fig. 5A. Scale bar: 100 μm. (D) Analysis of the percentage of electroporated GFP-cells in different regions (uCP, lCP, IZ, and VZ/SVZ) show that apart from the mild effect of ΔWD40 domain construct, none of the constructs have an effect on neuronal migration in control conditions. Data (means ± s.d.) from at least three embryos per condition were analyzed by two-way ANOVA, with Bonferroni's multiple comparisons test, ns, non-significant; *$P < 0.05$. uCP, Upper cortical plate; lCP, Lower cortical plate; IZ, intermediate zone; VZ, ventricular zone; SVZ, subventricular zone. (E) Top: Representative brain image stained with Nissl-luxol of adult male WT and *Camsap3* KO mice showing the soma of the corpus callosum at Bregma −1.34 mm. Bottom: Box plot showing the combined size of the lateral fibers (lat) of the corpus callosum with the hippocampal commissure in 3 male $Camsap3^{-/-}$ and 24 matched baseline WT mice of 16 weeks of age bred on a pure genetic background C57BL/6N. The line in the middle represents the median, the upper limit of the box corresponds to Q3, the lower limit to Q1, and the whiskers extend to 1.5× interquartile range. Scale bar: 0.05 cm. Data (means ± s.d.) was analyzed by two-tailed Student's *t*-tests of equal variances. ns, non-significant; **$P < 0.01$. cg, cingulate bundle; dhc, dorsal hippocampal commissure; lat, lateral fibers; med, medial fibers. Exact *P* values are listed in Dataset EV4.

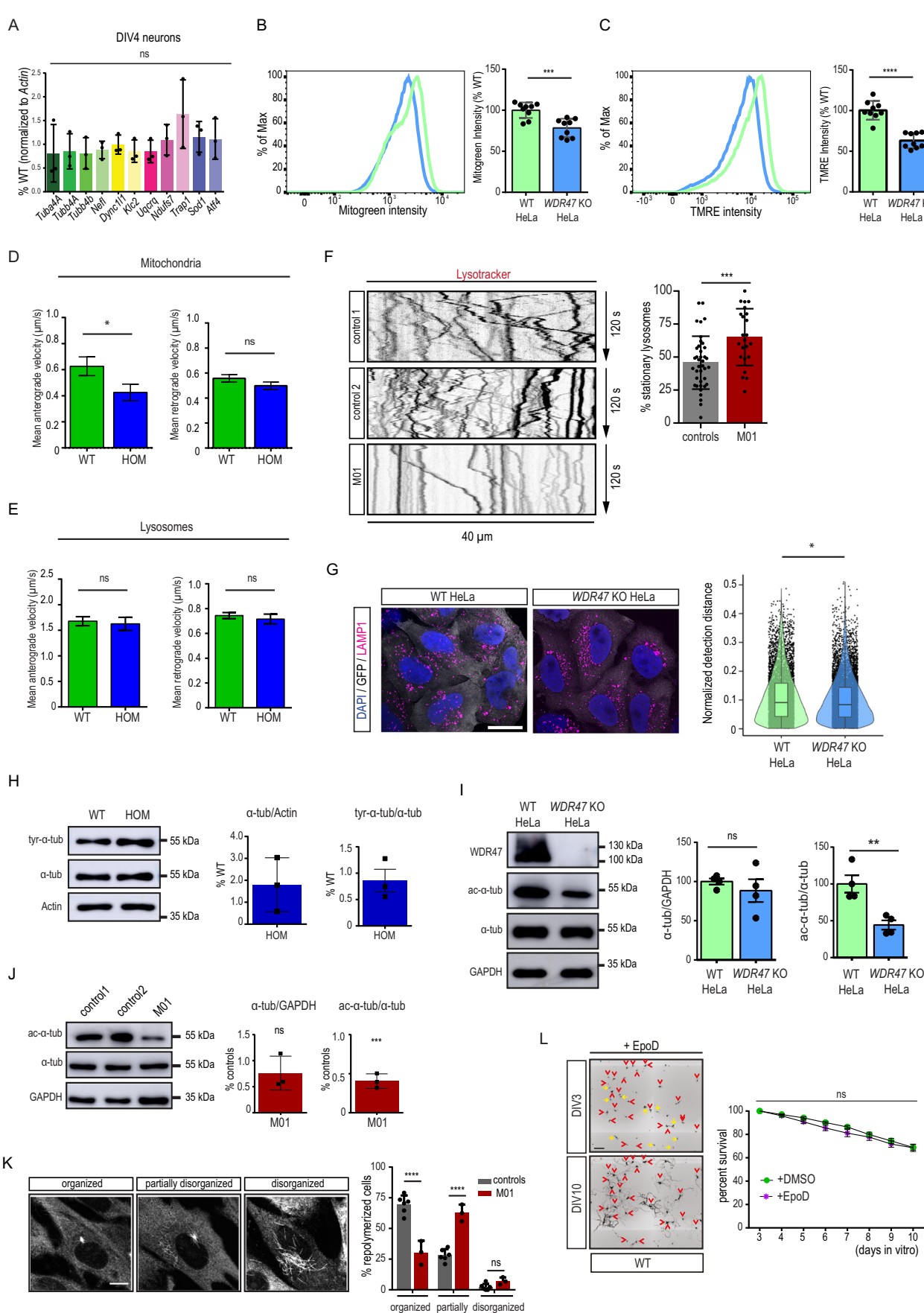

◄ **Figure EV4.  Loss of *Wdr47* in neurons does not impair intracellular transport or pool of dynamic microtubules.**

(**A**) RT-qPCR using extracts from DIV4 primary cortical neurons, for the 11 genes that were validated to be differently expressed at DIV6. Data (means ± s.d.) from at least 3 independent cultures per condition was analyzed by unpaired t-test. ns, non-significant. (**B**) Mitochondrial mass and (**C**) mitochondrial membrane potential quantified using flow cytometry in WT and *WDR47* KO HeLa cells. Data (means ± s.d.) from 9 independent experiments was analyzed by unpaired t-test, ***$P < 0.001$, ****$P < 0.0001$. (**D, E**) Histograms represent mean anterograde and retrograde velocities of (**D**) mitochondria (anterograde velocity: $n = 77$ for WT and $n = 36$ for HOM; retrograde velocity: $n = 130$ for WT and $n = 88$ for HOM) and (**E**) lysosomes (anterograde velocity: $n = 216$ for WT and $n = 116$ for HOM; retrograde velocity: $n = 408$ for WT and $n = 173$ for HOM). Data (means ± SEM) from at least 3 independent cultures per condition was analyzed by unpaired t-test with Welch correction, ns, non-significant, *$P < 0.05$. (**F**) Kymographs illustrating the motility of lysosomes (Lysotracker) in control and mutant fibroblast in time (y, sec) and space (x, μm). Histograms representing the percentage of stationary lysosomes. Lysosomes from $n = 39$ and $n = 23$ cells were analyzed for control lines and fibroblasts derived from M01, respectively, and data (means ± s.d.) was analyzed by unpaired t-test, ***$P < 0.001$. (**G**) LAMP1 immunostainings showing increased clustering of lysosomes around the nucleus in *WDR47*-KO HeLa cells compared to WT control HeLa cells. Cells were transfected with GFP to label their cytoplasm. Scale bar: 10 μm. $n = 7138$ and $n = 8447$ lysosomes from 94 WT and *WDR47*-KO HeLa cells was analyzed, respectively, by nested mixed effect model. Box plot: center line: median; box limits: 1st and 3rd quartiles; whiskers: $+/-1.5\times$ interquartile range. *$P < 0.05$. (**H**) Western blot analysis showing unchanged levels of alpha tubulin (α-tub) and tyrosinated alpha tubulin (tyr-α-tub) in DIV6 HOM (*Wdr47*[tm1b/tm1b]) primary neurons compared to WT. Actin was used as loading control. Data (means ± s.d.) from 3 independent cultures was analyzed by unpaired t-test. (**I, J**) Western blot analysis showing decreased levels of acetylated tubulin (ac-α-tub) and unchanged levels of alpha tubulin (α-tub) in (**I**) *WDR47* KO HeLa cells compared to WT HeLa cells and (**J**) fibroblasts derived from M01 compared to fibroblasts derived from healthy individuals. GAPDH was used as loading control. Data (means ± s.d.) from 3 to 4 independent cultures was analyzed by unpaired t-test, ns, non-significant, **$P < 0.01$; ***$P < 0.001$. (**K**) Representative images of human primary fibroblasts categorized as organized, partially disorganized and disorganized depending on the nucleation pattern of microtubules after 30 min of depolymerization followed by 2 min of repolymerization. Percentage of cells in each category is shown in M01-derived fibroblasts compared to control lines. >50 cells were analyzed for each culture and data (means ± s.d.) from 3 cultures was analyzed by two-way ANOVA, with Bonferroni's multiple comparison test, ns, non-significant; ****$P < 0.0001$. (**L**) Effect of EpoD on WT cultures. (Right) Representative fields, at DIV3 and DIV10, of WT neuronal cultures treated with EpoD at DIV2. Scarlet positive electroporated neurons are depicted in black. Yellow arrows correspond to neurons that died, red arrowheads correspond to neurons that are alive and could be followed from DIV3 to DIV10. (Left) Survival of WT neurons from DIV3 to DIV10 upon treatment with DMSO and EpoD. Data (means ± s.d.) from at least 3 cultures per condition was analyzed by two-way ANOVA, with Bonferroni's multiple comparison test. ns, non-significant. Exact *P* values are listed in Dataset EV4.

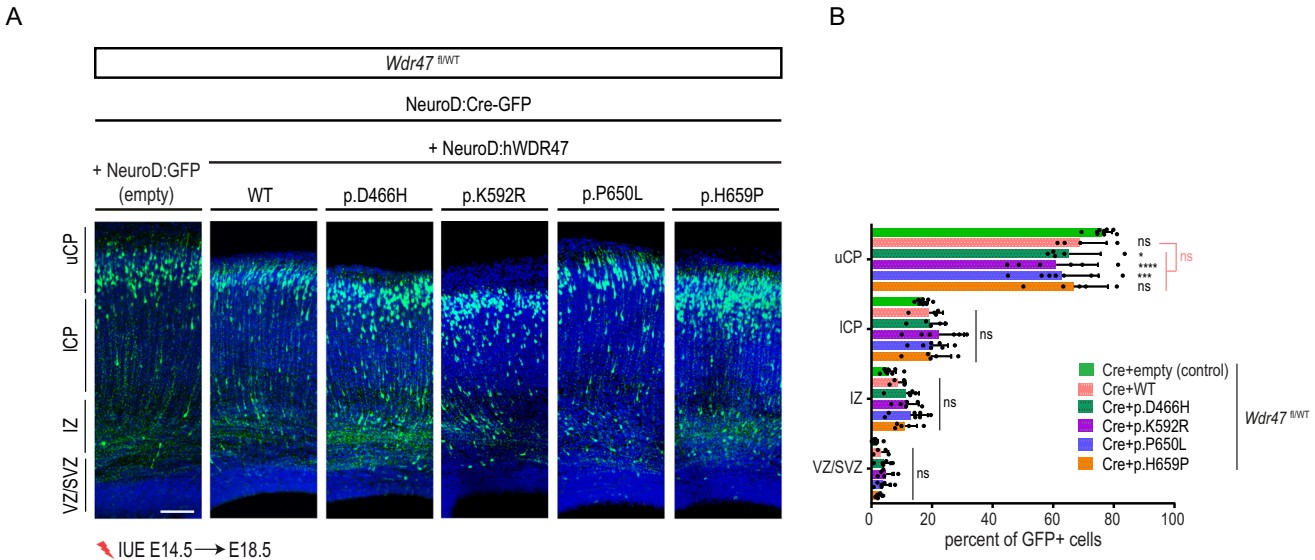

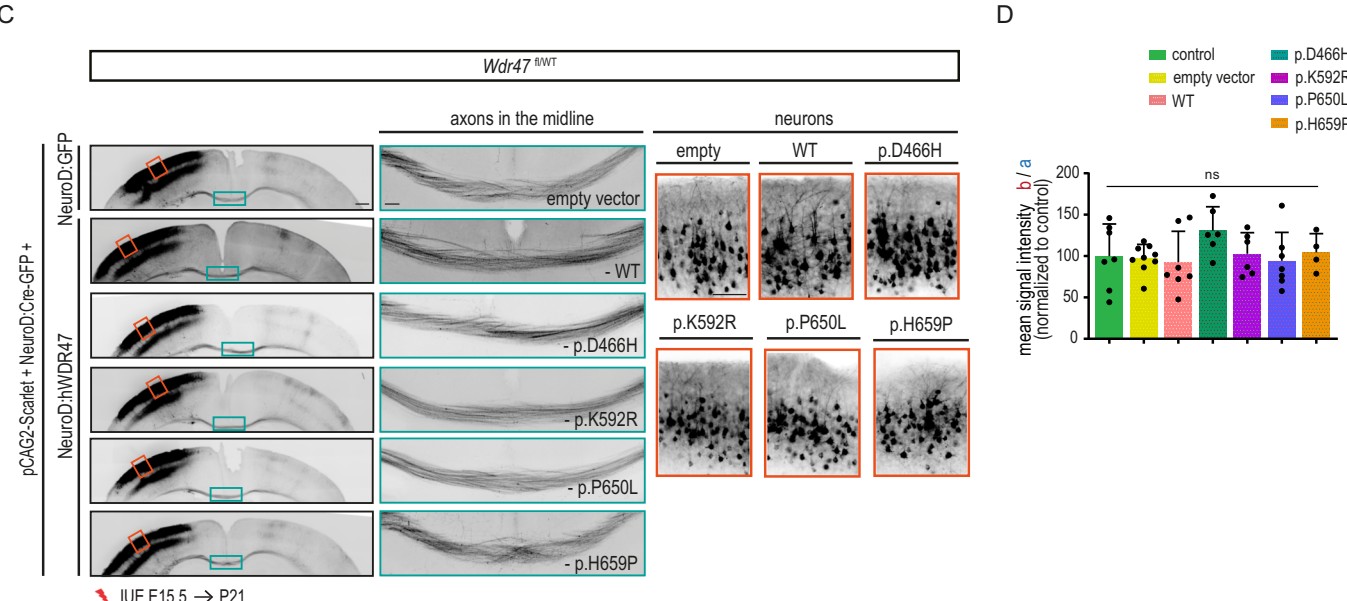

**Figure EV5.  Effect of wild-type and mutant hWDR47 constructs on CC and neuronal survival in control conditions.**

(A) Coronal sections of E18.5 *Wdr47*[fl/WT] mouse cortices, 4 days after in utero electroporation with NeuroD-Cre-GFP together with an empty (NeuroD:GFP) or wild type (WT) or mutant NeuroD:WDR47 constructs. GFP-positive electroporated cells are depicted in green. Nuclei are stained with DAPI. Scale bar: 100 μm. (B) Analysis of the percentage of electroporated GFP-cells in different regions (uCP, lCP, IZ, and VZ/SVZ) showing a mild effect of overexpression of some mutant WDR47 in the upper cortical plate (uCP). Data (means ± s.d.) from at least five embryos from 2 to 4 different litters per condition was analyzed by two-way ANOVA, with Bonferroni's multiple comparisons test. uCP, Upper cortical plate; lCP, Lower cortical plate; IZ, intermediate zone; VZ, ventricular zone; SVZ, subventricular zone. (C) Coronal sections of P21 *Wdr47*[fl/WT] mouse brains electroporated at E15.5 with pCAG:Scarlet and NeuroD:Cre-GFP plasmids and together with either an empty (NeuroD:GFP) or a NeuroD:WDR47 construct with or without the human mutation. Scarlet positive electroporated neurons are depicted in black. Close-up views of the green boxed and red boxed areas show that none of the constructs have an effect on CC or neuronal morphology. Scale bars: 500 μm, 100 μm (green and red boxed insets). (D) CC thickness upon introduction of different hWDR47 constructs. Data (means ± s.d.) from at least 5 pups per condition were analyzed by one-way ANOVA, with Bonferroni's multiple comparison test. ns, non-significant; *$P < 0.05$; ***$P < 0.001$; ****$P < 0.0001$. Exact $P$ values are listed in Dataset EV4.

