## [Peer Review File · EMBO Molecular Medicine]

Bi-allelic variants in WDR47 cause a complex neurodevelopmental syndrome

Efil Bayam, Peggy Tilly, Stephan Collins, José Rivera Alvarez, Meghna Kannan, Lucile Tonneau, Elena Brivio, Bruno Rinaldi, Romain Lecat, Noémie Schwaller, Ludovica Cotellessa, Sateesh Maddirevula, Fabiola Monteiro, Carlos Guardia, João Paulo Kitajima, Fernando Kok, Mitsuhiro Kato, Ahlam Hamed, Mustafa Salih, Saeed Al Tala, Mais Hashem, Hiroko Tada, Hirotomo Saitsu, Mariano Stabile, Paolo Giacobini, Sylvie Friant, Zafer Yüksel, Mitsuko Nakashima, Fowzan AlKuraya, Binnaz Yalcin, and Juliette Godin

Corresponding authors: Juliette Godin (godin@igbmc.fr) , Binnaz Yalcin (binnaz.yalcin@inserm.fr), Efil Bayam (bayame@igbmc.fr)

Review Timeline:

Submission Date:	9th Jan 24
Editorial Decision:	12th Feb 24
Revision Received:	29th Sep 24
Editorial Decision:	5th Nov 24
Revision Received:	13th Nov 24
Accepted:	13th Nov 24

Editor: Zeljko Durdevic

Transaction Report:

12th Feb 2024

Dear Dr. Godin,

Thank you for the submission of your manuscript to EMBO Molecular Medicine. We have now received feedback from the three reviewers who agreed to evaluate your manuscript. All three referees recognize potential interest of the study but also raise serious and partially overlapping concerns that should be addressed in a major revision. If you would like to discuss further the points raised by the referees, I am available to do so via email or video. Let me know if you are interested in this option.

We would welcome the submission of a revised version within three months for further consideration. Please let us know if you require longer to complete the revision.

I look forward to receiving your revised manuscript.

Yours sincerely,

Zeljko Durdevic

We require:

- 1) A .docx formatted version of the manuscript text (including legends for main figures, EV figures and tables). Please make sure that the changes are highlighted to be clearly visible.
- 2) Individual production quality figure files as .eps, .tif, .jpg (one file per figure). For guidance, download the 'Figure Guide PDF': (<https://www.embopress.org/page/journal/17574684/authorguide#figureformat>).
- 3) A .docx formatted letter INCLUDING the reviewers' reports and your detailed point-by-point responses to their comments. As part of the EMBO Press transparent editorial process, the point-by-point response is part of the Review Process File (RPF), which will be published alongside your paper.
- 4) A complete author checklist, which you can download from our author guidelines (<https://www.embopress.org/page/journal/17574684/authorguide#submissionofrevisions>). Please insert information in the checklist that is also reflected in the manuscript. The completed author checklist will also be part of the RPF.
- 5) Please note that all corresponding authors are required to supply an ORCID ID for their name upon submission of a revised manuscript.
- 6) It is mandatory to include a 'Data Availability' section after the Materials and Methods. Before submitting your revision, primary datasets produced in this study need to be deposited in an appropriate public database, and the accession numbers and

database listed under 'Data Availability'. Please remember to provide a reviewer password if the datasets are not yet public (see <https://www.embopress.org/page/journal/17574684/authorguide#dataavailability>).

13) Author contributions: You will be asked to provide CRediT (Contributor Role Taxonomy) terms in the submission system. These replace a narrative author contribution section in the manuscript.

14) A Conflict of Interest statement should be provided in the main text.

15) Every published paper now includes a 'Synopsis' to further enhance discoverability. Synopses are displayed on the journal

webpage and are freely accessible to all readers. They include a short stand first (maximum of 300 characters, including space) as well as 2-5 one-sentences bullet points that summarizes the paper. Please write the bullet points to summarize the key NEW findings. They should be designed to be complementary to the abstract - i.e. not repeat the same text. We encourage inclusion of key acronyms and quantitative information (maximum of 30 words / bullet point). Please use the passive voice. Please attach these in a separate file or send them by email, we will incorporate them accordingly.

Please also suggest a striking image or visual abstract to illustrate your article as a PNG file 550 px wide x 300-800 px high.

**** Reviewer's comments ****

Referee #1 (Comments on Novelty/Model System for Author):

The authors report novel information that is of great relevance to medical genetics /genomics, neurodevelopment and neuroscience. Mutations in WDR47 have not previously been shown to be disease-causing.

Corpus callosum agenesis/dysgenesis (CCD) is fairly common - as an isolated finding on MRI -or may be associated with other brain or neurological abnormalities, including known genetic syndromes. Apparently predisposing genetic factors have included deletions and duplications of numerous small chromosome regions, identified as CNVs by chromosomal microarray studies.. Knowledge is limited about single gene mutations that when inherited as recessive Mendelian traits cause CCD. Therefore, the discovery of another causative gene, WDR47, acting as a Mendelian recessive, represents a significant contribution.

The five cases in 4 families reported here have been identified from different parts of the world via the Gene Matcher platform, demonstrating the power of this approach to solving the causation of rare genetic disorders. In 3 consanguineous families, the affected individuals were homozygous for the missense variants. In the non-consanguineous family, the affected individual was a compound heterozygote. All 5 amino acid substitutions are conserved in evolution and are predicted to be deleterious.

The major part of the manuscript describes a large set of in vitro and in vivo studies of mouse models for WDR47 deficiency. The mouse models do not represent the mutations reported in the patients. Partial or complete knockout mouse models of WDR 47 were derived from a publicly available knock-out line previously used by the authors and others. Only the complementation assays in Fig. 7 use constructs representing the mutations identified in the patients.

Referee #1 (Remarks for Author):

The authors report novel information that is of great relevance to medical genetics /genomics, neurodevelopment and neuroscience. Mutations in WDR47 have not previously been shown to be disease-causing.

Corpus callosum agenesis/dysgenesis (CCD) is fairly common - as an isolated finding on MRI -or may be associated with other brain or neurological abnormalities, including known genetic syndromes. Apparently predisposing genetic factors have included deletions and duplications of numerous small chromosome regions, identified as CNVs by chromosomal microarray studies.. Knowledge is limited about single gene mutations that when inherited as recessive Mendelian traits cause CCD. Therefore, the discovery of another causative gene, WDR47, acting as a Mendelian recessive, represents a significant contribution.

The five cases in 4 families reported here have been identified from different parts of the world via the Gene Matcher platform, demonstrating the power of this approach to solving the causation of rare genetic disorders. In 3 consanguineous families, the affected individuals were homozygous for the missense variants. In the non-consanguineous family, the affected individual was a compound heterozygote. All 5 amino acid substitutions are conserved in evolution and are predicted to be deleterious.

The major part of the manuscript describes a large set of in vitro and in vivo studies of mouse models for WDR47 deficiency. The mouse models do not represent the mutations reported in the patients. Partial or complete knockout mouse models of WDR 47 were derived from a publicly available knock-out line previously used by the authors and others. Only the complementation assays in Fig. 7 use constructs representing the mutations identified in the patients.

Comments to consider for revision:

1. The clinical data are summarized in Table 1 which is illegible on the size of a page.
2. The Japanese patient (M03- designation missing on Table 1) had "multiple congenital anomalies" - these should be listed

/documented completely.

3. In family 1, the two affected males were delivered by elective C-section and died within 24-48 hrs. A female sibling had died at 7 days, with no clinical information. The brain MRI of one male shows severe CNS abnormalities. Were autopsies done on these children? Were there any other organs involved? What about the lungs? Were the boys able to breathe or kept on ventilators? Note that Wdr47-deficient mice did not breathe after birth (Chen 2020).

4. Family 1, with two R193H homozygous males that died at birth, is interpreted as the LOF equivalent of the mouse complete knock-out model. While transcript levels were normal, the transfected R193H protein shows decreased stability (Fig. 1K), apparently due to proteasomal degradation (Fig. 1L).

Before concluding that complete loss-of-function variants cause neonatal mortality as part of this disorder, the authors should exclude alternative explanations. The neonatal lethality could potentially be caused by another recessive condition (with variant not reported by the exome sequencing lab), given the high level of autozygosity. Table 1 states: "no pathogenic variants in autozygome - misspelled aurozygome". If re-examination of the autozygome results on family 1 is not informative, whole genome sequencing should be done to exclude the possibility of another pathogenic variant elsewhere in the genome.

5. Given that WDR47 is an important component of microtubules in motile cilia, did any of the patients have signs or symptoms of ciliary dyskinesia syndrome (as reported by Liu 2021 for a WDR47-deficient mouse model)?

6. The surviving patients all had CNS abnormalities beyond CCD, such as seizures, and (where known) facial dysmorphism, and severe developmental delay. This is a complex neurodevelopmental disorder and the emphasis placed on CCD in this paper seems excessive/unbalanced.

7. Detailed neuroanatomical studies on a variety of mouse models in vivo and in neuronal cultures generated novel insights into the mechanism of neuronal death in WDR47 deficiency. Data show that the pro-survival function is independent of WDR47's roles in migration and axonal growth; instead it involves the regulation of axonal transport and MT stability. On Page 21, the authors also claim that WDR47 plays a previously undetected role in mitochondrial homeostasis, because they observed that LOF WDR47 alleles lead to fragmentation of mitochondria and increased production of reactive oxygen species. Could this simply be a secondary process happening in dying cells?

8. The paper is very well written, except for the Discussion that includes several grammatical errors.

9. Incorrect Terminology on Pages 11 and 19: "Additionally, our results demonstrate that Wdr47 haploinsufficiency does not lead to any survival or axon extension defects". When the authors describe a lack of phenotype in heterozygous mice, they should not call them "haploinsufficient", but rather "heterozygous". Note the definition: "Haploinsufficiency is defined as a dominant phenotype in diploid organisms that are heterozygous for a loss-of-function allele".

10. Fig. 1N: why are there two bands for WDR47 on Western blot of control and mutant lymphoblastoid cell lines?

11. Whereas the 5 novel WDR47 missense mutations reported here were not found in databases, a R661H variant was reported in a agCC cohort, (polyphen: probably damaging) (Kannan et al PNAS 2017 Dataset S12). Have the authors tried to obtain more information on the three R661H homozygotes reported in ExAC?

12. The authors suggest that WDR47 screening should be considered in "unexplained syndromic cases involving corpus callosum dysgenesis, microcephaly and other neuroanatomical phenotypes." This recommendation sounds reasonable, but the field of human molecular genetic diagnostics is moving beyond single gene or gene panel testing to more global approaches like whole exome and genome sequencing. The information in this paper will be crucial in the interpretation of WDR47 variants identified by these global tests.

Referee #2 (Comments on Novelty/Model System for Author):

The authors have completed a thorough examination of the role of loss of function WDR47. The conditional animals demonstrate the importance of WDR47 in neurodevelopmental disease and are necessary given the lethality of knock-out models. There are no concerns on the use of animals. While WDR47 related disease is likely to be rare; understanding the role of WDR47 in neurodevelopmental disease is important.

Referee #2 (Remarks for Author):

The authors have done a substantial amount of work to correlate loss of function in WDR47 with corpus callosum dysgenesis.

Overall the manuscript is well written, and I have only minor suggestions/clarifications.

The use of whole should be removed from whole exome sequencing; there is nothing complete/whole in exome sequencing.

While there are no homozygous individuals in population databases, the MAF from gnomadV4 should be included in Table 1.

Lastly, while the data of p.R193H clearly supports loss of function, the other missense variants are only suggestive of diminished function, particularly p.K592R. The discussion should more clearly state the limitations/suggestive nature of the findings. Concordantly, the introduction states four bi-allelic loss-of-function variants, and this statement could also be softened.

Referee #3 (Comments on Novelty/Model System for Author):

This is the first time that human WDR47 variants are suggested to associate with human corpus callosum dysgenesis. This association has been previously suggested based on WDR47 KO mouse model (Kannan et al. 2017) revealing that WDR47 plays roles in brain development, corpus callosum dysgenesis, and autophagy in brain.

I found this manuscript very interesting with novel genotype-phenotype data on WDR47. However, I think that in its current form the MS doesn't reach the level acquired to be published in EMBO due the following aspects (please, see below the suggestions for a major revision). I conclude that the WDR47 mouse work is interesting on its own in studying the significance of WDR47 in brain development, but comparison to human data would require more evidence, even though proteins in different species are over 95% homologous.

Referee #3 (Remarks for Author):

This is the first time that human WDR47 variants are suggested to associate with human corpus callosum dysgenesis. This association has been previously suggested based on WDR47 KO mouse model (Kannan et al. 2017) revealing that WDR47 plays roles in brain development, corpus callosum dysgenesis, and autophagy in brain.

I found this manuscript very interesting with novel genotype-phenotype data on WDR47. However, I think that in its current form the MS doesn't reach the level acquired to be published in EMBO due the following aspects (please, see below the suggestions for a major revision). I conclude that the WDR47 mouse work is interesting on its own in studying the significance of WDR47 in brain development, but comparison to human data would require more evidence, even though proteins in different species are over 95% homologous.

A. Major revision: summary of data presented in the current MS version and suggestions for additional experiments to verify the human WDR47 genotype-phenotype association and to increase the novelty value of the manuscript:

A1. Human data on WDR47 genotype-phenotypes: The current title of the manuscript is misleading in focusing on WDR47-related human phenotype even though most of the data in the manuscript is based on the WDR47 KO mouse model except for brain MRIs from 5 patients with WDR47 variants, a summary table on clinical features and functional studies on fibroblasts from 2 out of 5 patients. The amount of human data was much less compared to mouse data, and the connection of human data to mice data was mostly based on the CC phenotype.

A2. Functional studies on WDR47 variants: All five WDR47 variants identified in patients were investigated as plasmid constructs to study their stability. The variant and wild type constructs were transfected into mouse N2A neuroblastoma cells. These protein constructs also had a HIS tag attached to the protein, which allowed the proteins to be identified from WB membranes. As a result, three variants were more stable and two less stable. One of the less stable variants could be saved by a protease inhibitor, whereas the other was broken down in the proteasome. These findings were then confirmed in the patient-derived fibroblasts of two patients.

- o In a patient with a homozygous variant, the WDR47 protein became more unstable in N2A cell model; the same result (=reduced amount of protein) was observed in fibroblasts confirming the finding.

- o In a patient with heterozygous variants, both variants were normally expressed/stable based on cellular studies and appeared to have a normal amount of protein confirming the finding.

- o Based on these results, it was suggested that the human CC phenotype may be due to a reduced amount of protein in at least some patients, while other variants may have some kind of loss of function effect.

- o As this is the first time when the WDR47 variants are associated with neurodevelopmental phenotype including dysgenesis of CC, more human specific data to confirm this association should be provided. As there could be several potential options for in vitro models by using iPSC techniques up to even brain organoids, why did the authors restrict the functional studies only on patient-derived fibroblasts instead of proceeding to human tissue and/or cell type specific experiments or brain organoids? (See for instance Ku RY, and Torii M. New Molecular Players in the Development of Callosal Projections? Cells 2021, and Kirihara T

et al. A Human Induced Pluripotent Stem Cell-Derived Tissue Model of a Cerebral Tract Connecting Two Cortical Regions, *iScience* 2019). Based on the WDR47 KO mouse data provided by the authors, these functional studies on patient-derived cell lines should also concern mitochondrial function (excessive ROS, mtDNA deletions, autophagosomes) and microtubule homeostasis.

A3. WES data on patients: The authors performed WES analyses, but how well have they ruled out other possible disease-causing variants in other genes? What other candidate gene variants were found in WES data that is not provided in detail in supplementary material (in the Table it was stated that there were no other candidate gene variants, but this seems quite unlikely)?

A4. Clinical features and neuroradiological findings in patients with WDR47 variants: It would have been necessary to describe in more detail the clinical and neuroradiological findings in patients associated with the WDR47 variants. As two of the patients have deceased, could any tissue samples from autopsy be available? What was the cause of the death in two patients? What were the macroscopic and histopathological autopsy findings in different organs? In the MS text, clinical features of the patients focus mainly on neuroradiological findings. More detailed clinical data is presented only in Table 1 that was written by using a very small font size. The clinical features in patients in different organs should be compared to the data on WDR47 tissue expression in humans.

A5. WDR47 expression in human tissues and cell types versus clinical and neuroradiological findings. The clinical manifestations of the patients in organ/tissue level should be compared to the current knowledge on RNA and protein expression of WDR47 in different tissues and cell types. Brain MRI data and especially anatomic location should be compared to the currently available data on WDR47 expression in different anatomical regions/cell types of the brain (including comparison between human and mice data). Could WDR47 expression on protein level be studied in different human tissues in which the patients present their symptoms and signs; for example, autopsy material or commercially available tissue homogenates (including different human brain regions) and cell lines.

A6. WDR47 mouse modelling: In this manuscript, there is a lot of emphasis on mouse model work. However, the difference here compared to the previous study by Kannan et al. (2017) is that while the previous work had been done on knockout mice, this work was performed on tissue-specific knockouts. In addition, in utero electroporation had been carried out, in which the significance of protein had been studied by targeting smaller populations of neurons. Despite WDR47 KO model data, the effect of identified human variants on WDR47 protein function has not been properly verified.

A7. The drug screening model was used to study the susceptibility of CC neurons to die from apoptosis in the first place and the mechanism behind them. WDR47 variants were more susceptible to death, while wild types survived drug treatment, comparing different routes leading to apoptosis.

B. Minor revision, suggestions for some minor changes (please, see below):

B1. Table 1 includes grammatic errors and wrong terminology (for example "hypotonic reduced head contrôle", "repeated flexin pattern", and "hypserythic EEG pattern").

B2. Discussion: the discussion part could be strengthened by adding a chapter on phenotypic spectrum of WDR-related ciliopathies including references (for example the most recent article by Accogli A et al. Variants in the WDR44 WD40-repeat domain cause a spectrum of ciliopathy by impairing ciliogenesis initiation. *Nat Commun* 2024).

B3. Discussion: The WDR47 expression in different tissues (including different brain regions) should be discussed regarding clinical phenotypes and neuroradiological findings of the patients.

We thank the reviewers for their advice and suggestions. We provide below detailed answers to the reviewers' concerns. As we have added a substantial amount of data, we have updated most of the figures and have added a new main figure (Figure 2), a supplementary table (Table EV1) and an expanded view figure (Figure EV1) in the revised manuscript. We hope that, with these substantial revisions, our manuscript will fit with the reviewers' requirements and be suitable for publication in *EMBO Molecular medicine*.

Referee #1 (Remarks for Author):

The authors report novel information that is of great relevance to medical genetics /genomics, neurodevelopment and neuroscience. Mutations in WDR47 have not previously been shown to be disease-causing. Corpus callosum agenesis/dysgenesis (CCD) is fairly common - as an isolated finding on MRI -or may be associated with other brain or neurological abnormalities, including known genetic syndromes. Apparently predisposing genetic factors have included deletions and duplications of numerous small chromosome regions, identified as CNVs by chromosomal microarray studies. Knowledge is limited about single gene mutations that when inherited as recessive Mendelian traits cause CCD. Therefore, the discovery of another causative gene, WDR47, acting as a Mendelian recessive, represents a significant contribution.

We thank the reviewer for acknowledging the novelty of our study.

The five cases in 4 families reported here have been identified from different parts of the world via the Gene Matcher platform, demonstrating the power of this approach to solving the causation of rare genetic disorders. In 3 consanguineous families, the affected individuals were homozygous for the missense variants. In the non-consanguineous family, the affected individual was a compound heterozygote. All 5 amino acid substitutions are conserved in evolution and are predicted to be deleterious.

Since the first submission, we have identified two novel cases from a fifth unrelated family. Those two siblings who carry the same variant than the one identified in Patient M04 (p.(Pro650Leu)) are now included in the revised version of the manuscript (new Patient M05 and new Patient M06). Figure 1 and Table 1 have been updated accordingly.

The major part of the manuscript describes a large set of in vitro and in vivo studies of mouse models for WDR47 deficiency. The mouse models do not represent the mutations reported in the patients. Partial or complete knockout mouse models of WDR 47 were derived from a publicly available knock-out line previously used by the authors and others. Only the complementation assays in Fig. 7 use constructs representing the mutations identified in the patients.

We believe that the knockout mouse model accurately represents the variant identified in newborns M01 and M02 (p.(Arg193His)), for which we were able to show a decrease of nearly 90% of the protein levels in patient cells (**New Figure 2E**). In addition, we were able to recapitulate the microtubules and lysosomal defects observed in KO neurons (**New Figure 7**) in cells derived from Patient M01 (**New Figure EV4F, J, K**), comforting our initial results. We validated those findings in WDR47 KO HeLa cells (**New Figure EV4G, I**) and further demonstrated impaired mitochondrial homeostasis (mitochondria membrane potential and mass) in this KO cell line compared to the control (**New Figure EV4A,C**).

Given the recessive mode of inheritance of all the identified variants and the segregation studies showing absence of phenotypes in the heterozygous parents, we are confident that the other identified variants (Patients M03-07) are likely acting through loss-of-function mechanisms. Finally, in addition to the data that were included in the initial manuscript showing that the human WDR47 variants fail to fully rescue the corpus callosum thickness (**New Figure 8**), we are now presenting data

showing that the human pathogenic WDR47 variants either failed to rescue or partially restore neuronal migration phenotype induced by the loss of *Wdr47* (**New Figure 8A,B**). This is now addressed in the Discussion section on pages 20-21 of the main manuscript file.

Comments to consider for revision:

1. The clinical data are summarized in Table 1 which is illegible on the size of a page.

We are now providing a more readable version of Table 1.

2. The Japanese patient (M03- designation missing on Table 1) had "multiple congenital anomalies" - these should be listed /documented completely.

We have revised **Table 1** and replaced "multiple congenital anomalies" by "growth failure accompanied by recurrent vomiting was observed since neonatal period". We are also now providing a complete description of the clinical features of the patients with *WDR47* variants in the **Appendix Note 1**.

3. In family 1, the two affected males were delivered by elective C-section and died within 24-48 hrs. A female sibling had died at 7 days, with no clinical information. The brain MRI of one male shows severe CNS abnormalities. Were autopsies done on these children? Where there any other organs involved? What about the lungs? Were the boys able to breathe or kept on ventilators? Note that *Wdr47*-deficient mice did not breathe after birth (Chen 2020).

Unfortunately, we don't have access to the brother's medical records. No autopsy was done on either baby. At birth, no other organs were reported affected. No mechanical ventilation was needed. Oxygen saturation was maintained by oxygen mask. No abnormal breathing was documented. All these information have been added to the **Appendix Note 1** that describes in detail the clinical features of the patients with *WDR47* variants.

4. Family 1, with two R193H homozygous males that died at birth, is interpreted as the LOF equivalent of the mouse complete knock-out model. While transcript levels were normal, the transfected R193H protein shows decreased stability (Fig. 1K), apparently due to proteasomal degradation (Fig. 1L).

Before concluding that complete loss-of-function variants cause neonatal mortality as part of this disorder, the authors should exclude alternative explanations. The neonatal lethality could potentially be caused by another recessive condition (with variant not reported by the exome sequencing lab), given the high level of autozygosity. Table 1 states: "no pathogenic variants in autozygome - misspelled aurozygome". If re-examination of the autozygome results on family 1 is not informative, whole genome sequencing should be done to exclude the possibility of another pathogenic variant elsewhere in the genome.

We are now specifying in **Table 1** the other variants found by exome sequencing for all the patients. We are also providing the list of those variants with more detailed information along with their exclusion criteria in **New Table EV1**. Concerning Family 1, please find a summary of the findings in the table below.

Gene	Variant and zygosity	Coding Effect	Frequency GnomAD v4.1.0	Comments
WDR47	Chr1(GRCh37):g.109554090C>T NM_001142551.2:c.578G>A; p.(Arg193His) Homozygous	CADD (v1.6): Phred: 29.0 REVEL (v2021-05-03): Score: 0.811 PolyPhen2 (v): Not computed SIFT (v6.2.0): DELETERIOUS MutationTaster (v2021): Deleterious.	0.00000248	Variant is proposed based on pathogenic nature of the variant (in silico predictions).
PON1	Chr7(GRCh37):g.94947671C>T NM_000446.7:c.109G>A; p.(Val37Ile) Homozygous	CADD (v1.6): Phred: 6.025 REVEL (v2021-05-03): Score: 0.046. PolyPhen2 (v): HDivPred: benign (score: 0) SIFT (v6.2.0): TOLERATED MutationTaster (v2021): Benign.	0.0000130	Excluded based on benign nature of the variant.
LMTK2	Chr7(GRCh37):g.97833300A>C NM_014916.4:c.4285A>C; p.(Met1429Leu) Homozygous	CADD (v1.6): Phred: 11.88, REVEL (v2021-05-03): Score: 0.249. PolyPhen2 (v): HDivPred: benign (score: 0). SIFT (v6.2.0): TOLERATED MutationTaster (v2021): Benign	0.00001115	Excluded based on the benign nature of the variant
PIEZO2	Chr18(GRCh37):g.10741080C>T NM_001378183.1:c.4657G>A; p.(Ala1553Thr) Homozygous	CADD (v1.6): Phred: 21.5 REVEL (v2021-05-03): Score: 0.175. PolyPhen2 (v): No result. SIFT (v6.2.0): TOLERATED	0.00001106	Excluded based on the benign nature of the variant and no match with the clinical features observed in Homozygous patient.
ANKRD30B	Chr18(GRCh37):g.14763726G>A NM_001367607.2:c.862G>A; p.(Ala288Thr) Homozygous	CADD (v1.6): Phred: 12.17, REVEL (v2021-05-03): Score: 0.030. PolyPhen2 (v): Not computed. SIFT (v6.2.0): TOLERATED	0.0000155	Excluded based on the benign nature of the variant and no match with the clinical features observed in Homozygous patient.

In accordance with the American College of Medical Genetics and Genomics (ACMG) guidelines, we used exome sequencing as a first-line diagnostic tool focusing on the coding regions and cost-effectiveness. Through this approach, we identified a likely pathogenic variant in the *WDR47* gene in Family 1 and have re-classified this gene as pathogenic based on functional studies and the clinical recurrence of neuroanatomical anomalies associated with it. We acknowledge the reviewer's valid point regarding the lack of recurrence of neonatal death in the other cases from our study. We have adjusted our discussion to moderate the suggested relationship between *WDR47* loss-of-function and neonatal lethality in humans (see in the Discussion section at the end of page 21 of the main manuscript file). Although our previously published murine data suggest that the absence of *WDR47* lead to lethality, the current evidence is not fully sufficient to support this conclusion in humans. Since exome sequencing provided a precise genetic diagnosis for Family 1's congenital anomalies, we did not explore whole genome sequencing or re-analysis of our existing exome sequencing data in this study. Future studies could consider these options to further investigate the potential link between pathogenic variants, including *WDR47*, and lethality.

5. Given that *WDR47* is an important component of microtubules in motile cilia, did any of the patients have signs or symptoms of ciliary dyskinesia syndrome (as reported by Liu 2021 for a *WDR47*- deficient mouse model)?

We have updated the clinical **Table 1** in the revised manuscript to include a column indicating the presence or the absence of ciliary-related phenotypes in each of the patient (where the data was available). None of the patients showed ciliary dyskinesia syndrome based on the assessment of medical doctors. However, five patients out of 7 showed enlarged ventricles with no other cilia-related pathological symptoms. Of note, our previous study (Collins et al. 2019 Nature Communications) also showed that enlarged ventricles can be correlated with disorders of the corpus callosum.

Observations in our human clinical cohort are consistent with our previous murine data (Kannan *et al.* 2017 PNAS) where we also observed partially penetrant enlarged ventricles in *Wdr47* mutant mice. More specifically, three out of seven homozygous knockout mice (*Wdr47*^{tm1a/tm1a} allele) displayed this phenotype, as documented in Supplementary Dataset 9 of our 2017 PNAS publication. In Figure 1 below, we provide an example of a *Wdr47*^{tm1a/tm1a} mouse with enlarged ventricles.

This is now discussed on page 19 of the main manuscript file.

Figure 1 Adult *Wdr47*-deficient male mouse model showing enlarged ventricles. The raw neuroanatomical measurements are available in Supplementary Dataset 9 of Kannan et al. 2017 PNAS. **A)** Representative coronal section, double-stained with Nissl-Luxol, at position Bregma +0.98 mm. **B)** Coronal section, double stained with Nissl-Luxol, at position Bregma -1.34 mm. The asterisks (*) indicate the enlarged size of the ventricles at both positions.

The Liu et al. 2021 Nature Communications article used the same *Wdr47* mutant mouse model as our study (*Wdr47*^{tm1a(EUCOMM)Wtsi}) obtained from the Wellcome Trust Sanger Institute. However, details about the strain genetic background and phenotypic penetrance in their study are not provided, limiting our ability to fully compare the models. Nevertheless, upon reviewing Figure 3b,e of the Liu *et al.* paper, we noticed the absence of the corpus callosum, indicating agenesis of the corpus callosum, a finding not discussed in their paper. Both models exhibit enlarged ventricles, though the penetrance in our study was only partial. As for the excess mucus in the nasal cavities reported by Liu *et al.*, we did not assess this phenotype and cannot comment on it.

6. The surviving patients all had CNS abnormalities beyond CCD, such as seizures, and (where known) facial dysmorphia, and severe developmental delay. This is a complex neurodevelopmental disorder and the emphasis placed on CCD in this paper seems excessive/unbalanced.

We thank the reviewer for his/her valuable comment. We have made adjustments throughout the text, in particular in the introduction and abstract, to present a more balanced overview of the disorder presented in this study. We have also revised the title to: “Bi-allelic variants in WDR47 cause a complex neurodevelopmental syndrome”, to better reflect the complexity of the clinical features observed in the patients. We have also added a discussion point on the complexity of this new syndrome focusing on its comorbidities (see pages 19-20 in the Discussion section of the main text).

7. Detailed neuroanatomical studies on a variety of mouse models in vivo and in neuronal cultures generated novel insights into the mechanism of neuronal death in WDR47 deficiency. Data show that

the pro-survival function is independent of WDR47's roles in migration and axonal growth; instead it involves the regulation of axonal transport and MT stability. On Page 21, the authors also claim that WDR47 plays a previously undetected role in mitochondrial homeostasis, because they observed that LOF WDR47 alleles lead to fragmentation of mitochondria and increased production of reactive oxygen species. Could this simply be a secondary process happening in dying cells?

We have observed that the defects in mitochondria homeostasis start at 6 days *in vitro*, so when cells are healthy given that cell death occurs at DIV 8. For most of the analyses (number, size and transport of mitochondria (**New Figure 7**), we performed time lapse recording in neurons and could attest that none of the recorded neurons presented with hallmarks of cell death (retracted or fragmented neurites, vacuoles, blebbing or budding). In addition, we now show that *WDR47* KO HeLa that do not show any defects in survival present with impaired mitochondria homeostasis, suggesting that mitochondrial defects are unlikely triggered by cell death (**New Figure EV4B,C**). Although we are confident the mitochondrial defects are not a consequence of dying neurons, we cannot totally rule out this possibility and accordingly propose to moderate our conclusion in the revised version of the manuscript (see page 23).

8. The paper is very well written, except for the Discussion that includes several grammatical errors.

We thank the reviewer for her/his positive feedback on the overall writing quality. We have carefully reviewed and revised the Discussion section to correct any grammatical errors and improve clarity.

9. Incorrect Terminology on Pages 11 and 19: "Additionally, our results demonstrate that *Wdr47* haploinsufficiency does not lead to any survival or axon extension defects". When the authors describe a lack of phenotype in heterozygous mice, they should not call them "haploinsufficient", but rather "heterozygous". Note the definition: "Haploinsufficiency is defined as a dominant phenotype in diploid organisms that are heterozygous for a loss-of-function allele".

Thanks for spotting this mistake. We have removed the recurrence of "haploinsufficient" and made the appropriate corrections (pages 10, 12 and 21).

10. Fig. 1N: why are there two bands for WDR47 on Western blot of control and mutant lymphoblastoid cell lines?

We do observe two bands when we run Western blot on either fibroblasts (M01) or Lymphoblastoides (M03) cell lines, although the lower unspecific band (as seen in figure 2 below) is less pronounced in fibroblasts.

Figure 2: Western Blot analysis of protein extracts from WT and *WDR47* KO HeLa cells as well as control LCLs confirming the specificity of the upper band seen in LCLs (former figure 1N, see now figure 1P now showing only the specific band)

11. Whereas the 5 novel WDR47 missense mutations reported here were not found in databases, a R661H variant was reported in a agCC cohort, (polyphen: probably damaging) (Kannan et al PNAS

2017 Dataset S12). Have the authors tried to obtain more information on the three R661H homozygotes reported in ExAC?

According to the latest gnomAD dataset (version 4.1.0), the R661H variant (NM_001142550.1: c.1982G>A) has been identified in 31 homozygous occurrences out of 801,913 exomes/genomes (transcript of reference in gnomAD = NM_001142551.2; p.Arg653His). This frequency suggests that the variant is unlikely to be pathogenic. This is further confirmed by structural modeling using an Alphafold2-derived atomic model suggesting that the p.Arg653His substitution does not significantly affect the protein's structure or folding.

12. The authors suggest that WDR47 screening should be considered in "unexplained syndromic cases involving corpus callosum dysgenesis, microcephaly and other neuroanatomical phenotypes." This recommendation sounds reasonable, but the field of human molecular genetic diagnostics is moving beyond single gene or gene panel testing to more global approaches like whole exome and genome sequencing. The information in this paper will be crucial in the interpretation of WDR47 variants identified by these global tests.

We agree with the reviewer and accordingly changed the last sentence of the introduction (page 5) and discussion (page 25).

Referee #2 (Comments on Novelty/Model System for Author):

The authors have completed a thorough examination of the role of loss of function WDR47. The conditional animals demonstrate the importance of WDR47 in neurodevelopmental disease and are necessary given the lethality of knock-out models. There are no concerns on the use of animals. While WDR47 related disease is likely to be rare; understanding the role of WDR47 in neurodevelopmental disease is important.

We thank the reviewer for his/her positive comments and advice to improve our manuscript.

Referee #2 (Remarks for Author):

The authors have done a substantial amount of work to correlate loss of function in WDR47 with corpus callosum dysgenesis. Overall the manuscript is well written, and I have only minor suggestions/clarifications.

The use of whole should be removed from whole exome sequencing; there is nothing complete/whole in exome sequencing.

Thanks for spotting this, we have changed all the WES occurrences by exome sequencing.

While there are no homozygous individuals in population databases, the MAF from gnomadV4 should be included in Table 1.

We have included these metrics in the new version of the **Table 1**.

Lastly, while the data of p.R193H clearly supports loss of function, the other missense variants are only suggestive of diminished function, particularly p.K592R. The discussion should more clearly state the limitations/suggestive nature of the findings. Concordantly, the introduction states four bi-allelic loss-of-function variants, and this statement could also be softened.

We agree with this comment and have adjusted the text accordingly throughout the manuscript and are now discussing the extend of LOF of the variants on pages 20-21 in the Discussion section.

Referee #3 (Comments on Novelty/Model System for Author):

This is the first time that human WDR47 variants are suggested to associate with human corpus callosum dysgenesis. This association has been previously suggested based on WDR47 KO mouse model (Kannan et al. 2017) revealing that WDR47 plays roles in brain development, corpus callosum dysgenesis, and autophagy in brain.

I found this manuscript very interesting with novel genotype-phenotype data on WDR47. However, I think that in its current form the MS doesn't reach the level acquired to be published in EMBO due the following aspects (please, see below the suggestions for a major revision). I conclude that the WDR47 mouse work is interesting on its own in studying the significance of WDR47 in brain development, but comparison to human data would require more evidence, even though proteins in different species are over 95% homologous.

Referee #3 (Remarks for Author):

This is the first time that human WDR47 variants are suggested to associate with human corpus callosum dysgenesis. This association has been previously suggested based on WDR47 KO mouse model (Kannan et al. 2017) revealing that WDR47 plays roles in brain development, corpus callosum dysgenesis, and autophagy in brain.

I found this manuscript very interesting with novel genotype-phenotype data on WDR47. However, I think that in its current form the MS doesn't reach the level acquired to be published in EMBO due the following aspects (please, see below the suggestions for a major revision). I conclude that the WDR47 mouse work is interesting on its own in studying the significance of WDR47 in brain development, but comparison to human data would require more evidence, even though proteins in different species are over 95% homologous.

We thank the reviewer for acknowledging the interest and novelty of our study. As suggested, we are now providing more evidence confirming the function of the human WDR47 protein. These include: 1) analysis of human and mouse transcriptomic data (after extraction from <https://apps.kaessmannlab.org/evodevoapp/> and from <https://cellxgene.cziscience.com/gene-expression> databases) and immunostainings on human fetal tissues (**New Figure 1K,N and Fig EV1**) that show similar spatial and temporal expression pattern of the human and mouse transcripts and proteins, and 2) functional analysis in human cells (patient-derived fibroblast cell lines, WDR47 KO HeLa cells) that demonstrate shared function in regulating microtubules (**New Figure EV4I-K**), organelles (lysosomes and mitochondria) distribution (**New Figure EV4F,G**) and function (**New Figure EV4B,C**). In addition, in the initial manuscript, we presented *in vivo* complementation assay (**New Figure 8**) that consist in expressing the **human wild type** WDR47 proteins in mouse KO neurons using in utero electroporation. The complete rescue of both the migration (**New Figure 8A-B**) and connectivity (**New Figure 8C-E**) phenotypes confirms the critical role of human WDR47 in the developing cortex.

A. Major revision: summary of data presented in the current MS version and suggestions for additional experiments to verify the human WDR47 genotype-phenotype association and to increase the novelty value of the manuscript:

A1. Human data on WDR47 genotype-phenotypes: The current title of the manuscript is misleading in focusing on WDR47-related human phenotype even though most of the data in the manuscript is based on the WDR47 KO mouse model except for brain MRIs from 5 patients with WDR47 variants, a

summary table on clinical features and functional studies on fibroblasts from 2 out of 5 patients. The amount of human data was much less compared to mouse data, and the connection of human data to mice data was mostly based on the CC phenotype.

The novelty of this study resides in describing a new *WDR47*-related neurodevelopmental syndrome and the associated mechanisms. Notably, since the first submission, we have identified two novel cases from a fifth unrelated family. Those two siblings who carry the same variant than the one identified in Patient M04 (p.(Pro650Leu)) are now included in the revised version of the manuscript (M05 and M06). Figure 1 and Table 1 have been updated accordingly. This reinforces the genetic link between *WDR47* variants and human developmental phenotypes. As discussed above, we are now providing more evidence showing the roles of the human protein in the developing cortex and at the cellular levels. We agree that the patients have CNS abnormalities beyond CC phenotype. We have therefore changed the title to: “Bi-allelic variants in *WDR47* cause a complex neurodevelopmental syndrome”. We have also removed the word “human” from the title to make it more inclusive as we agree with the Reviewer’s feedback regarding the inclusion of both human and mouse data.

A2. Functional studies on *WDR47* variants: All five *WDR47* variants identified in patients were investigated as plasmid constructs to study their stability. The variant and wild type constructs were transfected into mouse N2A neuroblastoma cells. These protein constructs also had a HIS tag attached to the protein, which allowed the proteins to be identified from WB membranes. As a result, three variants were more stable and two less stable. One of the less stable variants could be saved by a protease inhibitor, whereas the other was broken down in the proteasome. These findings were then confirmed in the patient-derived fibroblasts of two patients.

- o In a patient with a homozygous variant, the *WDR47* protein became more unstable in N2A cell model; the same result (=reduced amount of protein) was observed in fibroblasts confirming the finding.

- o In a patient with heterozygous variants, both variants were normally expressed/stable based on cellular studies and appeared to have a normal amount of protein confirming the finding.

- o Based on these results, it was suggested that the human CC phenotype may be due to a reduced amount of protein in at least some patients, while other variants may have some kind of loss of function effect.

- o As this is the first time when the *WDR47* variants are associated with neurodevelopmental phenotype including dysgenesis of CC, more human specific data to confirm this association should be provided. As there could be several potential options for *in vitro* models by using iPSC techniques up to even brain organoids, why did the authors restrict the functional studies only on patient-derived fibroblasts instead of proceeding to human tissue and/or cell type specific experiments or brain organoids? (See for instance Ku RY, and Torii M. New Molecular Players in the Development of Callosal Projections? *Cells* 2021, and Kirihaara T et al. A Human Induced Pluripotent Stem Cell-Derived Tissue Model of a Cerebral Tract Connecting Two Cortical Regions, *iScience* 2019). Based on the *WDR47* KO mouse data provided by the authors, these functional studies on patient-derived cell lines should also concern mitochondrial function (excessive ROS, mtDNA deletions, autophagosomes) and microtubule homeostasis.

We thank the reviewer for this suggestion. However, generating a human iPSC model within the scope of the current revision is not feasible. Indeed, the human cells available (fibroblasts from the patient) are not suitable for *in vitro* reprogramming due to high passage numbers, and getting more samples is not possible as some patients are dead, or live in secluded region or do not consent in giving biological samples. While creating a human iPSC model through genome editing of WT iPSC cells would be an interesting approach, it would require more than a year to complete.

Instead, as mentioned by the reviewer, we are providing a better characterization of the patient fibroblasts available (M01) and using *WDR47* KO HeLa cells to confirm the function of the human

protein in regulating microtubule integrity (**New Figure EV4I-K**), lysosomal transport and distribution (**New Figure EV4F-G**) and mitochondria homeostasis (**New Figure EV4B-C**). Of note, the newly provided fibroblast data were obtained a couple of years ago and were not initially included in the initial manuscript. We also tried to perform similar experiments with the lymphoblastoid cell line obtained from Patient M03, but these cell lines are now highly passaged with very low doubling times, which limits their utility for further functional analysis and rescue experiments. In addition, we no longer have access to new batches of these two cell lines.

A3. WES data on patients: The authors performed WES analyses, but how well have they ruled out other possible disease-causing variants in other genes? What other candidate gene variants were found in WES data that is not provided in detail in supplementary material (in the Table it was stated that there were no other candidate gene variants, but this seems quite unlikely)?

We are now specifying in **Table 1** the other variants found by exome sequencing for all the patients. We are also providing the list of those variants with more detailed information along with their exclusion criteria in **Appendix Note 1**.

A4. Clinical features and neuroradiological findings in patients with WDR47 variants: It would have been necessary to describe in more detail the clinical and neuroradiological findings in patients associated with the WDR47 variants. As two of the patients have deceased, could any tissue samples from autopsy be available? What was the cause of the death in two patients? What were the macroscopic and histopathological autopsy findings in different organs? In the MS text, clinical features of the patients focus mainly on neuroradiological findings. More detailed clinical data is presented only in Table 1 that was written by using a very small font size. The clinical features in patients in different organs should be compared to the data on WDR47 tissue expression in humans.

We are now providing 1) a more readable and revised version of **Table 1**, and 2) a complete description of the clinical features of the patients with *WDR47* variants as **Appendix Note 1**. No autopsy was done on either baby in Family 1. At birth, no other organs were reported affected. No mechanical ventilation was needed. Oxygen saturation was maintained by oxygen mask. No abnormal breathing was documented. All these information have been added to the **Appendix Note 1**.

The brain is the most affected organ. This is in line with the expression pattern of WDR47 that is particularly enriched in both developing and adult brain compared to other organs in human and mice (re-analysis of data extracted from <https://apps.kaessmannlab.org/evodevoapp/> and from <https://cellxgene.cziscience.com/gene-expression> databases) (**New Figure 1M,L,O**). Four of the five surviving patients present with feeding difficulties and one patient has a misplaced right testis. Yet the expression of WDR47 is very low in both the testis and the organs of the digestive system (**New Figure 1M,L,O**). We are now discussing the clinical features and their comorbidities in regard to the expression profile of WDR47 on pages 19-20 in the Discussion section of the main text.

A5. WDR47 expression in human tissues and cell types versus clinical and neuroradiological findings. The clinical manifestations of the patients in organ/tissue level should be compared to the current knowledge on RNA and protein expression of WDR47 in different tissues and cell types. Brain MRI data and especially anatomic location should be compared to the currently available data on WDR47 expression in different anatomical regions/cell types of the brain (including comparison between human and mice data). Could WDR47 expression on protein level be studied in different human tissues in which the patients present their symptoms and signs; for example, autopsy material or commercially available tissue homogenates (including different human brain regions) and cell lines.

We have reanalyzed several transcriptomics datasets (from <https://apps.kaessmannlab.org/evodevoapp/> and from <https://cellxgene.cziscience.com/gene-expression>) and found that : 1) WDR47 is particularly enriched in the brain from development to adult stages (**New Figure 1K,L**); and 2) within the cortex, WDR47 shows a higher expression in neurons than in glial cells. Of note, Wdr47 is expressed in all cortical layers including in layer2/3 that contain callosal neurons (**New Figure 1O**). Immunostainings on human fetal tissues at GW14 (**New Figure 1M,N**) revealed a high expression in both ventral and dorsal cortex compared to other brain structures, with expression in germinal zones, cortical plate and intermediate zone that is enriched in projecting axons.

A6. WDR47 mouse modelling: In this manuscript, there is a lot of emphasis on mouse model work. However, the difference here compared to the previous study by Kannan et al. (2017) is that while the previous work had been done on knockout mice, this work was performed on tissue-specific knockouts. In addition, in utero electroporation had been carried out, in which the significance of protein had been studied by targeting smaller populations of neurons. Despite WDR47 KO model data, the effect of identified human variants on WDR47 protein function has not been properly verified.

Although we agree that the knockout mouse model does not model all the variants, we believe that it models the variant identified in newborns M01 and M02 (p.(Arg193His)), for which we were able to show a decrease of nearly 90% of the protein levels in patient cells (**New Figure 2E**). In addition, we were able to recapitulate the microtubules and lysosomal defects observed in KO neurons (**New Figure 7**) in cells from Patient M01 (**New Figure EV4F, J, K**), comforting our initial results. We validated those findings in *WDR47* KO HeLa cells (**New Figure EV4G, I**) and further demonstrated impaired mitochondrial homeostasis (mitochondria membrane potential and mass) in this KO cell line compared to the control (**New Figure EV4A,C**).

Given the recessive mode of inheritance of all the identified variants and the segregation studies showing absence of phenotypes in the heterozygous parents, we are confident that the other identified variants (Patients M03-07) are likely acting through loss of function mechanisms. Finally, in addition to the data that were included in the initial manuscript showing that the human WDR47 variants fail to fully rescue the corpus callosum thickness (**New Figure 8**), we are now presenting data showing that the human pathogenic WDR47 variants either failed to rescue or partially restore neuronal migration phenotype induced by the loss of *Wdr47* (**New Figure 8A,B**). This is now discussed page 21. Importantly, our multiple attempts to better understand the dysfunction of human WDR47 variants at the molecular (lysosomes, mitochondria, microtubule homeostasis) level using complementation assays in *WDR47* KO HeLa cells were unsuccessful, due to issues to efficiently introduce the WDR47 constructs in a proper dosage that do not interfere with the overall cellular homeostasis.

A7. The drug screening model was used to study the susceptibility of CC neurons to die from apoptosis in the first place and the mechanism behind them. WDR47 variants were more susceptible to death, while wild types survived drug treatment, comparing different routes leading to apoptosis.

Those experiments allowed us to show that apoptosis is the main route of neuronal death and that microtubules dysfunction was partially contributing to the phenotype.

B. Minor revision, suggestions for some minor changes (please, see below):

B1. Table 1 includes grammatic errors and wrong terminology (for example "hypotonic reduced head contrôle", "repeated flexin pattern", and "hypserythic EEG pattern").

Thanks for spotting these mistakes. We are using the correct terminology:

hypotonic reduced head control
repeated flexion pattern
hypsarrythmic EEG pattern

B2. Discussion: the discussion part could be strengthened by adding a chapter on phenotypic spectrum of WDR-related ciliopathies including references (for example the most recent article by Accogli A et al. Variants in the WDR44 WD40-repeat domain cause a spectrum of ciliopathy by impairing ciliogenesis initiation. Nat Commun 2024).

We have reviewed the clinical data of each patient and have updated the clinical **Table 1** in the revised manuscript to include a column indicating the presence or the absence of ciliary-related phenotypes in each of the patient (where the data was available). We found five patients out of 7 showing enlarged ventricles with no other cilia-related pathological symptoms. All the details have been added in the brain MRI section of **Table 1** and in the **Appendix Note 1**. These findings are now discussed as suggested (see page 19).

Of note, our previous study (Collins *et al.* 2019 Nature Communications) also showed that enlarged ventricles can be correlated with disorders of the corpus callosum.

Observations in our human clinical cohort are consistent with our previous murine data (Kannan *et al.* 2017 PNAS) where we also observed partially penetrant enlarged ventricles in *Wdr47* mutant mice. More specifically, three out of seven homozygous knockout mice (*Wdr47*^{tm1a/tm1a} allele) displayed this phenotype, as documented in Supplementary Dataset 9 of our 2017 PNAS publication. In Figure 1 below, we provide an example of a *Wdr47*^{tm1a/tm1a} mouse with enlarged ventricles.

Figure 1 Adult *Wdr47*-deficient male mouse model showing enlarged ventricles. The raw neuroanatomical measurements are available in Supplementary Dataset 9 of Kannan *et al.* 2017 PNAS. **A)** Representative coronal section, double-stained with Nissl-Luxol, at position Bregma +0.98 mm. **B)** Coronal section, double stained with Nissl-Luxol, at position Bregma -1.34 mm. The asterisks (*) indicate the enlarged size of the ventricles at both positions.

B3. Discussion: The WDR47 expression in different tissues (including different brain regions) should be discussed regarding clinical phenotypes and neuroradiological findings of the patients.

We are now discussing the phenotypes in regard to WDR47 pattern of expression (see pages 19-20).

5th Nov 2024

Dear Dr. Godin,

Thank you for the submission of your revised manuscript to EMBO Molecular Medicine. I am pleased to inform you that we will be able to accept your manuscript pending the following final amendments:

- 1) Please address minor comments raised by the referee #2.
- 2) Figures: We note that some images/panels are reused. Figure 5A (+ NeuroD:GFP(empty)) is reused in Figure EV3C (+ NeuroD:GFP(empty)). Also, Figure 6B (+DMSO, HOM DIV3 and DIV5) is reused in Figure 7O ((+DMSO, HOM DIV3 and DIV5). Please cite in the respective figure legend every reused image/panel.
- 3) Author Checklist: Please add corr. author name, journal and manuscript number.
- 4) In the main manuscript file, please do the following:
 - Please address all comments suggested by our data editors listed below:
 - o Please note that the specific URLs for Leiden Open Variation Database (0000953686, 0000445644, 0000445645), BiImage repository (accession number TMP_1727364961311) datasets should be provided in the data availability statement.
 - o Figure legends:
 1. Please note that the exact p values are not provided in the legends of figures 2c-e; 4c-d; 5b-c, e-f; 6c; 7d, f-i, k, m-n, p; 8b, e; EV 2c; EV 3b, d-e; EV 4c-d, f-g, i-k; EV 5b.
 2. Please indicate the statistical test used for data analysis in the legend of figure 7a.
 3. Please note that in figure EV 4b; there is a mismatch between the annotated p values in the figure legend and the annotated p values in the figure file that should be corrected.
 4. Please note that the box plots need to be defined in terms of minima, maxima, centre, bounds of box and whiskers, and percentile in the legends of figures EV 3e; EV 4g.
 5. Please note that information related to n is missing in the legends of figures 7a; EV 3b.
 6. Please note that the scale bar is missing for figure EV 5a.
 - Add up to 5 keywords.
 - Please confirm that a signed statement of informed consent to publish any identifiable format (video, recording, photograph, image) has been obtained from each person (parents or legal guardians for minors) who appears in the study. Please make sure that the depictions of the individual concerned and MRI images (in Figure 1F patient information are visible) are anonymized and only those details essential for understanding and interpreting the results of the study should be presented. If the eyes are not essential for the interpretation of the results please place a black bar over patient eyes. Please check "Author Guidelines" for more information: <https://www.embopress.org/page/journal/17574684/authorguide#humansubjects>
 - Indicate in legends exact n and exact p values, not a range, along with the statistical test used. To keep the figures "clear" some authors found providing an Appendix table Sx with all exact p-values preferable. You are welcome to do this if you want to.
 - Please include structured Methods section that includes a Reagents and Tools Table (should be uploaded as a separate file) followed by a Methods and Protocols section. More information on how to adhere to this format as well as downloadable templates (.docx) for the Reagents and Tools Table can be found in our author guidelines: <https://www.embopress.org/page/journal/17574684/authorguide#structuredmethods>

An example of a paper with Structured Methods can be found here:
<https://www.embopress.org/doi/full/10.1038/s44320-024-00037-6#sec-4>

 - Author contributions: Please remove it from the manuscript and specify author contributions in our submission system. CRediT has replaced the traditional author contributions section because it offers a systematic machine-readable author contributions format that allows for more effective research assessment. You are encouraged to use the free text boxes beneath each contributing author's name to add specific details on the author's contribution. More information is available in our guide to authors: <https://www.embopress.org/page/journal/17574684/authorguide#authorshipguidelines>
 - Data availability: Please make sure that all deposited data are freely accessible. The journal encourages authors to provide access to genotype and clinical data with as few restrictions as possible while respecting ethical obligations to the patients and relevant medical and legal issues. A signed statement of informed consent to publish any human clinical, large-scale, and genomic datasets must be obtained from each person (parents or legal guardians for minors) who appears in a study. Please check "Author Guidelines" for more information <https://www.embopress.org/page/journal/17574684/authorguide#datadeposition>
 - Please place Figure legends and EV Figure legends after references.

- 5) Appendix: Please add page numbers to the table of content. Rename Appendix Note 1 to Appendix Supplementary Information and update the callout in the main manuscript text.
- 6) Tables: Please rename Table 1 to Dataset EV4 (or renumber the submitted datasets according to their callouts in the text) and remove it legends from the main manuscript file and place it in a separate sheet/tab in the Excel file. Table 2 and its callouts should be updated accordingly.
- 7) Funding: Please make sure that information about all sources of funding are complete in both our submission system and in the manuscript. Currently, SFRI-STRAT'US project [ANR 20-SFRI-0012]; EUR IMCBio [ANR-17-EURE-0023], CNRS; University of Strasbourg, Intramural Research Program of the NIH, National Institute of Environmental Health Sciences (project ZIA ES103370-01), Agence Nationale de la Recherche (ANR-11-PDOC-0029-0, Ministère de l'Enseignement Supérieur de la

Recherche et de l'Innovation; Japan Society for the Promotion of Science, Grant-in-Aids for Scientific Research (JP23K27566, JP21K06819 and JP24K11055), Japan Agency for Medical Research and Development (AMED; grant numbers: JP23ek0109549, JP23ek0109674, JP23ek0109637 and 24ek0109591), Takeda Science Foundation Specific Research Grants, HUSM Grant-in-Aid from Hamamatsu University School of Medicine, the Ministry of Health, Labor, and Welfare Research Program on Rare and Intractable Diseases JPMH23FC0201 are missing in our submission system.

8) Synopsis:

- Synopsis image: Please upload the image as a high-resolution jpeg, TIFF or png file 550 pixels wide x 200-600 pixels high.
- Please check your synopsis text and image before submission with your revised manuscript. Please be aware that in the proof stage minor corrections only are allowed (e.g., typos).

9) Source data: Please upload source data as one zip file per figure. Please note that dataset deposited in BiolImage repository (accession number TMP_1727364961311) is not accessible.

10) As part of the EMBO Publications transparent editorial process initiative (see our Editorial at <http://embomolmed.embopress.org/content/2/9/329>), EMBO Molecular Medicine will publish online a Review Process File (RPF) to accompany accepted manuscripts. This file will be published in conjunction with your paper and will include the anonymous referee reports, your point-by-point response and all pertinent correspondence relating to the manuscript. Let us know whether you agree with the publication of the RPF and as here, if you want to remove or not any figures from it prior to publication. Please note that the Authors checklist will be published at the end of the RPF.

11) Please provide a point-by-point letter INCLUDING my comments as well as the reviewer's reports and your detailed responses (as Word file).

I look forward to reading a new revised version of your manuscript as soon as possible.

Yours sincerely,

Zeljko Durdevic

*** Instructions to submit your revised manuscript ***

1) a .docx formatted version of the manuscript text (including Figure legends and tables)

2) Separate figure files*

3) supplemental information as Expanded View and/or Appendix. Please carefully check the authors guidelines for formatting Expanded view and Appendix figures and tables at <https://www.embopress.org/page/journal/17574684/authorguide#expandedview>

4) a letter INCLUDING the reviewer's reports and your detailed responses to their comments (as Word file).

5) The paper explained: EMBO Molecular Medicine articles are accompanied by a summary of the articles to emphasize the

major findings in the paper and their medical implications for the non-specialist reader. Please provide a draft summary of your article highlighting

6) Author contributions: the contribution of every author must be detailed in a separate section.

7) EMBO Molecular Medicine now requires a complete author checklist

(<https://www.embopress.org/page/journal/17574684/authorguide>) to be submitted with all revised manuscripts. Please use the checklist as guideline for the sort of information we need WITHIN the manuscript. The checklist should only be filled with page numbers where the information can be found. This is particularly important for animal reporting, antibody dilutions (missing) and exact values and n that should be indicated instead of a range.

8) Every published paper now includes a 'Synopsis' to further enhance discoverability. Synopses are displayed on the journal webpage and are freely accessible to all readers. They include a short stand first (maximum of 300 characters, including space) as well as 2-5 one sentence bullet points that summarise the paper. Please write the bullet points to summarise the key NEW findings. They should be designed to be complementary to the abstract - i.e. not repeat the same text. We encourage inclusion of key acronyms and quantitative information (maximum of 30 words / bullet point). Please use the passive voice. Please attach these in a separate file or send them by email, we will incorporate them accordingly.

You are also welcome to suggest a striking image or visual abstract to illustrate your article. If you do please provide a jpeg file 550 px-wide x 300-600px high.

9) A Conflict of Interest statement should be provided in the main text

10) Please note that we now mandate that all corresponding authors list an ORCID digital identifier. This takes <90 seconds to complete. We encourage all authors to supply an ORCID identifier, which will be linked to their name for unambiguous name identification.

Currently, our records indicate that the ORCID for your account is 0000-0001-6559-1065.

Link Not Available

11) Include a Reagents and Tools Table as part of the Methods section, which can be downloaded from our author guidelines (<https://www.embopress.org/page/journal/17574684/authorguide#structuredmethods>)

Photos 400-800 DPI

*Additional important information regarding figures and illustrations can be found at

<https://bit.ly/EMBOPressFigurePreparationGuideline>. See also figure legend preparation guidelines:

<https://www.embopress.org/page/journal/17574684/authorguide#figureformat>

***** Reviewer's comments *****

Referee #1 (Comments on Novelty/Model System for Author):

This substantially revised manuscript is greatly improved. The authors provide detailed responses to previous criticisms and have made corrections. They added two more patients and listed the other genetic variants found in this mostly consanguineous series of cases. The new experimental work presented is convincing and well documented.

Referee #1 (Remarks for Author):

This substantially revised manuscript is greatly improved. The authors provide detailed responses to previous criticisms and have made corrections. They added two more patients and listed the other genetic variants found in this mostly consanguineous series of cases. The new experimental work presented is convincing and well documented.

Referee #2 (Comments on Novelty/Model System for Author):

The authors have completed a thorough revision of the original manuscript of the examination of the role of loss of function WDR47. The conditional animals demonstrate the importance of WDR47 in neurodevelopmental disease and are necessary given the lethality of knock-out models. There are no concerns on the use of animals. Although an additional family was identified, WDR47 disease is likely to remain rare. Further, given the variants are missense, a clinical diagnosis will be difficult in the absence of functional studies.

Referee #2 (Remarks for Author):

The authors have completed a thorough revision of the manuscript. I have only minor comments to consider.

- 1) At the request of another reviewer additional variants identified in the analysis were included in the clinical table. However, the list of c. is not informative. For example, why were the bi-allelic HEATR5B variants in M03 not considered? This is also associated with a neurodevelopmental disorder (PMID: 33824466)
- 2) the pedigree drawings in Figure 1 should be corrected. In Family 1 a consanguineous mother in generation III is shown, but with no relationship within the pedigree. Either this should be addressed, or generation I and II could be removed for clarity. In all pedigrees if a parent was tested and heterozygous this is typically shown as either a dot or partial shading.

The authors addressed the remaining editorial issues.

13th Nov 2024

Dear Dr. Godin,

We are pleased to inform you that your manuscript is accepted for publication and is now being sent to our publisher to be included in the next available issue of EMBO Molecular Medicine.
